

# Magnetic flux response of non-Hermitian topological phases

**Marco Michael Denner[1] and Frank Schindler[2,3]**

**1** Department of Physics, University of Zürich, Winterthurerstrasse 190,
8057, Zürich, Switzerland
**2** Princeton Center for Theoretical Science, Princeton University,
Princeton, NJ 08544, USA
**3** Blackett Laboratory, Imperial College London, London SW7 2AZ, United Kingdom

## Abstract

We derive the response of non-Hermitian topological phases with intrinsic point gap topology to localized magnetic flux insertions. In two spatial dimensions, we identify the necessary and sufficient conditions for a *flux skin effect* that localizes an extensive number of in-gap modes at a flux core. In three dimensions, we furthermore establish the existence of: a *flux spectral jump*, where flux tube insertion fills up the entire point gap only at a single parallel crystal momentum; a *higher-order flux skin effect*, which occurs at the ends of flux tubes in presence of pseudo-inversion symmetry; and a *flux Majorana mode* that represents a spectrally isolated mid-gap state in the complex energy plane. We uniquely associate each non-Hermitian symmetry class with intrinsic point gap topology with one of these cases or a trivial flux response, and discuss possible experimental realizations.

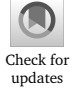

## 1  Introduction

Gapped Hermitian topological phases can be differentiated from trivial phases, as well as between each other, by a quantized response to local magnetic fluxes [1–9]. Such fluxes take the form of zero-dimensional (0D) flux cores (vortices) in two dimensions (2D), or one-dimensional (1D) flux tubes in three dimensions (3D). In particular, in presence of a symmetry that quantizes the magnetic flux $\phi$ through a plaquette (stack of plaquettes) of a 2D (3D) lattice – most commonly to values $\phi = 0, \pi$ – topological insulators generically host flux-localized bound states [10]. In 2D, these states may be constrained to occur at zero energy by a spectral symmetry such as the particle-hole symmetry of (mean-field) superconductors. For example, magnetic vortices in a 2D $p + \mathrm{i}p$ superconductor are known to host unpaired Majorana zero-modes [11, 12]. Without spectral symmetry, flux bound states in 2D systems can be moved out of the gap, but may still contribute to a filling anomaly of the ground state that cannot be trivialized without breaking a symmetry or closing a gap [13]. In 3D, flux bound states fall into two categories: in the first case, they form a gapless state localized along the 1D flux tube, as is the case for $\pi$-flux tubes in 3D time-reversal symmetric topological insulators [3–9]. In the recently discovered second case, the flux tube *ends* bind 0D states that give rise to a filling anomaly or are pinned to zero energy by a spectral symmetry [14]. This latter case is realized in higher-order topological phases protected by crystalline symmetries.

The goal of our present work is to generalize the theory of flux bound states to non-Hermitian (NH) topological phases that break energy conservation. Such phases have generated increased interest recently due to their unconventional bulk-boundary correspondence [15–25]. Fundamentally, there exist two kinds of NH systems [15–18]: Hamiltonians $H$ with a line gap in their complex energy spectrum can be adiabatically (without closing the line gap) deformed to purely Hermitian ($H^\dagger = H$) or anti-Hermitian ($H^\dagger = -H$) Hamiltonians. On the other hand, Hamiltonians without any line gap may still host a point gap – a region of complex energy that is devoid of, but surrounded by, eigenstates. Point-gapped systems without a line gap cannot be adiabatically deformed to any (anti-)Hermitian limit, and therefore realize intrinsically NH topology. A prime example is the 1D Hatano-Nelson chain [26–28] in NH symmetry class A (no symmetries) whose bulk winding number invariant $W \in \mathbb{Z}$ in periodic boundary conditions (PBC) results in the NH skin effect under open boundary conditions (OBC): when $W \neq 0$, an extensive number of (almost all) OBC eigenstates accumulate at only one edge of the system [29–37]. While this observation is reminiscent of a bulk-boundary correspondence, it is unclear how the OBC spectrum may crisply differentiate between systems with different nonzero $W$. To alleviate this issue, we here employ the NH *pseudospectrum* that consists of the collection of spectra associated with all $\mathcal{O}(\epsilon)$-deformed Hamiltonians (see Sec. 2.1 for details) [38–40]. The pseudospectrum reduces to the spectrum in the Hermitian

case, but represents the more physically adequate quantity in the NH case: if a state fails to be an eigenstate only by $\mathcal{O}(\epsilon)$ terms, then it behaves as an eigenstate with respect to realistic measurements. Moreover, the pseudospectrum restores the full bulk-boundary correspondence of NH point-gapped systems: for instance, a Hamiltonian with nonzero bulk winding number $W$ has a pseudospectrum that fills the point gap and is $W$-fold degenerate in presence of boundaries [39]. In the thermodynamic limit, the pseudospectrum is equivalent to the spectrum in semi-infinite boundary conditions (SIBC) [38–40]. Correspondingly, we characterize the flux response of NH systems by their SIBC spectrum in presence of flux defects. We exclusively focus on NH phases with *intrinsic point gap topology*, which were classified in all 38 NH symmetry classes in Refs. [16, 39]: in addition to stability under manipulations preserving point gap and symmetry class, these phases cannot be trivialized even when coupling with arbitrary NH line-gapped phases is allowed.

We find that, depending on the NH symmetry class, the flux response of intrinsically NH point-gapped phases falls into one of 5 classes:

(1) *No flux response:* Some NH phases have a trivial flux response even when their bulk is topologically nontrivial.

(2) *Flux skin effect:* Inserting a $\pi$-flux core in a 2D NH system can induce a skin effect response where a macroscopic number of OBC eigenstates localizes at the flux core [39]. Concomitantly, the SIBC spectrum in presence of flux fills up the entire point gap in the complex energy plane. This effect is similar to the dislocation skin effect of Refs. [41, 42], with the important distinction that it does not rely on (discrete) translational symmetry and therefore probes strong instead of weak NH topology [14].

(3) *Flux spectral jump:* Threading a 1D $\pi$-flux tube through a 3D NH system with nontrivial point gap topology can result in a spectral jump as the momentum coordinate $k_\parallel$ along the flux tube is varied. In particular, the SIBC spectrum in presence of flux remains gapped at generic values of $k_\parallel$, but completely fills up with an extensive number of states at one of the two special points $k_\parallel = 0, \pi$. Such a gapless NH dispersion cannot be realized in any purely 1D lattice system, and therefore represents an intrinsically NH anomalous 1D state. We note that a purely 1D system cannot realize a filled point gap at a single momentum, as there is only a finite Hilbert space available at any given momentum. Instead, the discontinuity in the 1D flux tube dispersion described here capitalizes on a topologically nontrivial 2D bulk.

(4) *Higher-order flux skin effect:* Even when the SIBC spectrum of a 3D NH phase in presence of a 1D flux tube is fully gapped (does not exhibit a spectral jump), preserving crystalline *pseudo-inversion* symmetry [43–51] may result in a nontrivial higher-order response. In this case, an extensive number of eigenstates accumulates at the *ends* of flux tubes when these terminate at sample surfaces. Concomitantly, the SIBC spectral point gap remains empty with PBC along the flux tubes, but is completely filled up in OBC when a surface termination perpendicular to the flux tubes is introduced.

(5) *Flux Majorana modes:* In just two NH symmetry classes, intrinsic 3D point gap topology furthermore manifests itself in an unpaired flux Majorana mode pinned to the center of the SIBC spectral point gap. For the case of NH class D, this mode is localized along the entire 1D length of the flux tube. Such localization is impossible in Hermitian systems where energetically isolated bound states must be pointlike.

In Tab. 1, we identify the flux response of each NH symmetry class that allows for intrinsic point gap topology with one of the 5 possible NH flux responses. In Sec. 2 we outline the formalism that we use to derive the flux responses, before discussing the individual effects:

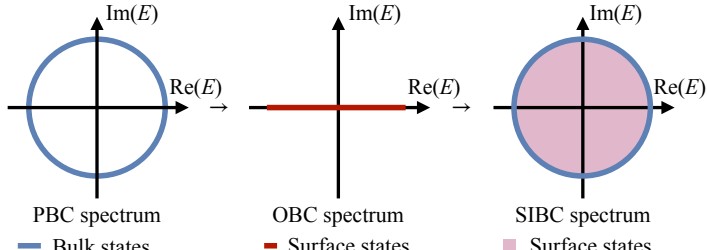

Figure 1: **NH bulk-boundary correspondence.** The NH skin effect, which corresponds to a nonzero topological invariant $W(E)$ [Eq. (1)], induces a spectral collapse of the point gap under OBC. However, the resulting OBC spectrum does not uniquely specify the nonzero value of $W(E)$. This ambiguity is resolved in the SIBC spectrum, which restores the NH bulk-boundary correspondence: surface states fill the point gap, and their degeneracy corresponds to $W(E)$.

in Sec. 3 we highlight the flux skin effect, Sec. 4 presents the flux spectral jump, in Sec. 5 we introduce the higher-order flux skin effect, and finally Sec. 6 unveils the flux Majorana modes. We close by discussing potential experimental realizations in Sec. 7. The appendices contain exhaustive auxiliary derivations and toy model Hamiltonians.

## 2 Formalism

We begin by pedagogically reviewing the concepts needed to characterize flux responses in the NH context.

### 2.1 NH Bulk-boundary correspondence

NH systems can differ dramatically in their spectra under PBC and OBC. One of the prime examples of this feature is the NH skin effect, whereby a point-gapped bulk collapses to a line under OBC (see Fig. 1). Connected with this is a pile-up of all states at a single boundary. In 1D, this effect is associated with a winding number $W(E)$ [39],

$$W(E) = \int_0^{2\pi} \frac{dk}{2\pi i} \frac{d}{dk} \log \det[\mathcal{H}(k) - E], \tag{1}$$

where the Bloch Hamiltonian $\mathcal{H}(k)$ is point-gapped about the reference energy $E$. In particular, if $W(E) \neq 0$, the NH skin effect must occur. However, it is important to note that $W(E) \in \mathbb{Z}$ is an integer-valued invariant, while the presence or absence of a skin effect only provides a $\mathbb{Z}_2$ quantifier. This observation raises the question of how the OBC spectrum of two systems with nonzero but different $W(E)$ can be crisply distinguished. The problem is further compounded by the dramatic sensitivity of the OBC spectrum [52] to small changes of the Hamiltonian.

To solve both issues, the $\epsilon$-pseudospectrum of NH systems was introduced in Refs. [38–40]. It describes the change in the spectrum under a small perturbation $\epsilon$,

$$\sigma_\epsilon(H) = \{E \in \mathbb{C} \mid ||(H - E)|v\rangle|| < \epsilon, \quad \text{for at least one } |v\rangle \text{ with } \langle v|v\rangle = 1\}. \tag{2}$$

This definition is practical: if a state is just $\mathcal{O}(\epsilon)$ away from being an eigenstate, it will still behave as one with regards to realistic measurements, which always include a small error. In contrast to other approaches reestablishing the NH bulk-boundary correspondence, for instance the OBC treatment of Ref. [53], the pseudospectrum, therefore, does not suffer from a

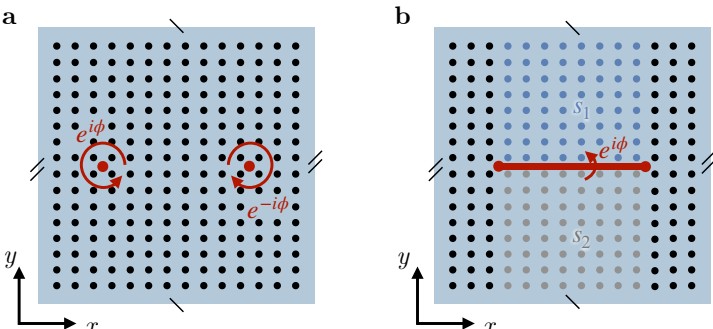

Figure 2: **Implementation of flux defects. a** Periodic systems must contain *pairs* of flux defects $\pm\phi$. **b** These can be implemented by multiplying all hoppings from sites $s_1$ to $s_2$ by $e^{i\phi}$, and by $e^{-i\phi}$ in the opposite direction.

sensitivity to infinitesimal errors. The pseudospectrum is particularly useful because it can be related to the spectrum in SIBC $\sigma_{\text{SIBC}}(H)$, which is the spectrum in presence of just a single boundary in the thermodynamic limit (system size $L \to \infty$). In particular, it holds that [38–40]

$$\lim_{\epsilon \to 0} \lim_{L \to \infty} \sigma_\epsilon(H) = \sigma_{\text{SIBC}}(H). \tag{3}$$

This correspondence between pseudospectrum and SIBC spectrum allows for a precise definition of a NH bulk-boundary correspondence. In particular, in the SIBC spectrum of a 1D system with nonzero $W(E)$, the point gap fills completely with boundary localized states whose degeneracy equals $W(E)$ (see Fig. 1).

## 2.2 Extended Hermitian Hamiltonian

To find the flux response of NH systems, we rely on the topological equivalence between a NH Hamiltonian $H$ and an extended Hermitian Hamiltonian (EHH) $\bar{H}$ [54,55], defined as

$$\bar{H} = \begin{pmatrix} 0 & H - E_0 \\ H^\dagger - E_0^* & 0 \end{pmatrix}. \tag{4}$$

By construction, $\bar{H}$ enjoys a chiral symmetry,

$$\bar{\Sigma}_C \bar{H} \bar{\Sigma}_C^\dagger = -\bar{H}, \quad \bar{\Sigma}_C = \begin{pmatrix} \mathbb{1} & 0 \\ 0 & -\mathbb{1} \end{pmatrix}. \tag{5}$$

The presence of a point gap of $H$ around $E = E_0$ translates into a gapped spectrum of $\bar{H}$ about zero energy. Conversely, exact topological zero-energy eigenvalues of $\bar{H}$ correspond to protected $E = E_0$ states within the NH point gap, because

$$\det(\bar{H}) = -\det(H - E_0)\det(H^\dagger - E_0^*) = 0 \quad \to \quad \det(H - E_0) = 0. \tag{6}$$

Consequently, we can predict the presence of protected in-gap modes from the EHH spectrum.

## 2.3 Flux defects in NH systems

Flux defects are routinely employed as probes of bulk topology in Hermitian systems [1–9,56]. For instance, $\pi$-flux cores in 2D topological insulators bind a single Kramers pair of midgap states [2–4]. While general defects [57], dislocations [41,42], and disclinations [58] have been investigated in the NH context, a systematic understanding of the flux response of all intrinsically point-gapped NH symmetry classes has been absent until now.

Any topological flux response relies on a quantization of the admissible values of flux $\phi$. Restricting to local NH symmetries corresponding to one of the 38 NH symmetry classes [16] – potentially taken together with a crystalline symmetry like pseudo-inversion – we find that, depending on the symmetry class, flux is either unquantized or must take values $\phi = 0, \pi$ in PBC. In the cases where $\phi = 0, \pi$, the NH symmetry class must either contain a time-reversal (TRS) or particle-hole (PHS) symmetry, or include crystalline pseudo-inversion symmetry.

Here, we focus on PBC in order to cleanly separate the NH flux response from boundary states or skin effects. The PBC geometry requires (at least) two flux cores/tubes with strength $\pm\phi$ [14]. Such a pair of fluxes can be introduced by the Peierls substitution: the hoppings encircling each flux must accumulate a phase $e^{\pm i\phi}$ (see Fig. 2a). A convenient electromagnetic gauge choice is then to multiply all hoppings across the line (plane) connecting the two fluxes in a 2D (3D) system by a factor of $e^{i\phi}$ in one direction ($s_1 \to s_2$, see Fig. 2b) and by $e^{-i\phi}$ in the other direction ($s_2 \to s_1$), where we denote the sites above the line (plane) as $s_1$ and the ones beneath as $s_2$. We choose the orientation of this line (plane) along the $x$- ($x$- and $z$-)direction in 2D (3D), respectively. SIBC corresponds to the idealized limit where only one flux (= only one "boundary") is present, while the second flux is infinitely far away. In our conventions, a system with SIBC then has a single flux core/tube at $x = 0$, is infinitely extended in the $x$-direction, and is finite with PBC in all remaining directions (but potentially still thermodynamically large).

We derive the flux response for all NH symmetry classes with intrinsic point gap topology from their respective EHH. The EHH experiences the same flux defect as the NH Hamiltonian: in the gauge described above, Peierls substitution implies

$$\langle s_1 | H_\phi | s_2 \rangle = e^{i\phi} \langle s_1 | H_{\phi=0} | s_2 \rangle. \tag{7}$$

The corresponding matrix element for the EHH reads $\langle s_1^\alpha | \bar{H}_\phi | s_2^\beta \rangle$, where $\alpha, \beta = 1, 2$ label the two sublattices introduced by the Hermitian extension. The only nonzero contributions are

$$\langle s_1^{\alpha=1} | \bar{H}_\phi | s_2^{\beta=2} \rangle = e^{i\phi} \langle s_1 | H_{\phi=0} | s_2 \rangle, \tag{8}$$

and

$$\langle s_1^{\alpha=2} | \bar{H}_\phi | s_2^{\beta=1} \rangle = e^{i\phi} \langle s_1 | H_{\phi=0}^\dagger | s_2 \rangle, \tag{9}$$

where, importantly, the flux enters in both cases as $e^{i\phi}$. The reason is that Hermitian conjugation in Eq. (9) not only flips the sign of the flux, but also transposes the matrix elements $s_1 \to s_2$ to $s_2 \to s_1$. In Eq. (7) we have used that $E_0$ multiplies an identity matrix in Eq. (4) and therefore remains unaffected under flux insertion.

In summary, our strategy for determining the flux response of a given NH symmetry class $X$ is to:

(1) Find the corresponding Hermitian symmetry class $\bar{X}$ of the EHH.

(2) Derive the flux response of $\bar{X}$ using the Dirac formalism detailed in the appendices.

(3) Infer the topological properties of the NH SIBC spectrum that result from exact EHH zeromodes.

# 3 NH flux skin effect

We begin our survey of NH flux responses with 2D systems. Here we first consider a specific example in NH class AII$^\dagger$ (see App. E.1.1 for details). Point-gapped systems in NH class AII$^\dagger$

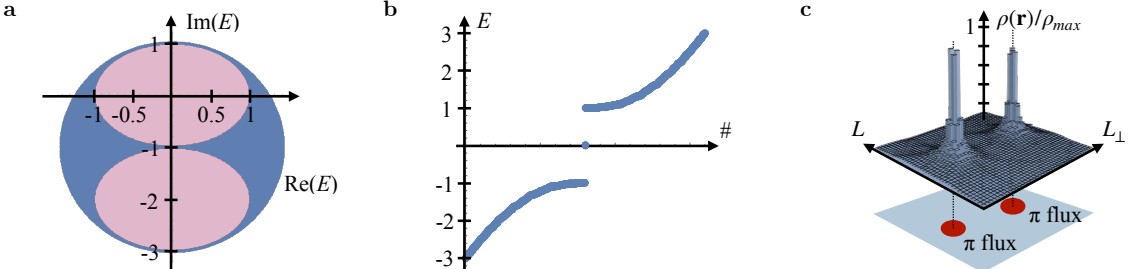

Figure 3: **NH flux skin effect in 2D. a** Spectrum in the complex energy plane under PBC (blue) and SIBC in presence of $\pi$-flux cores (blue and red) for a model exhibiting nontrivial point gap topology in NH class AII$^\dagger$ [Eq. (E.1)]. Flux-localized states fill the point gap in the SIBC spectrum, showing the flux skin effect. **b** The energy spectrum of the EHH for NH class AII$^\dagger$ [Eq. (E.1)] shows four topological zero energy modes within the bulk energy gap, caused by two $\pi$-flux cores. **c** The flux skin effect localizes an extensive number of eigenmodes at the fluxes, indicated by two peaks of the summed density $\rho(\boldsymbol{r}) = \sum_{\alpha,i} |\langle \boldsymbol{r}_i | \psi_\alpha \rangle|^2$, where $\alpha$ ranges over all eigenstates $\psi_\alpha$ of the Hamiltonian with flux defect, $\boldsymbol{r}$ denotes the lattice site, and the summation $i$ runs over sublattice degrees of freedom.

are classified by a $\mathbb{Z}_2$ invariant $\nu \in \{0,1\}$, defined in Ref. [16]. In order to derive the flux response of the nontrivial phase where $\nu = 1$, we rely on the EHH for a given energy $E_0$ inside the point gap (see Sec. 2.2). Irrespective of the choice of $E_0$, the EHH for NH class AII$^\dagger$ enjoys the symmetries of Hermitian class DIII (see App. B.4.1 and Tab. 2 therein). Moreover, since Hermitian class DIII is likewise $\mathbb{Z}_2$-classified in 2D, the EHH associated with a nontrivial NH phase in class AII$^\dagger$ must itself realize a nontrivial topological insulator phase in Hermitian class DIII. For this phase, it was shown in Ref. [10] that flux cores bind two degenerate zero-energy states (a Kramers pair). In the NH SIBC spectrum, these modes then correspond to a single flux localized state at complex energy $E_0$ in the point gap [39]. Since we can perform this construction for all $E_0$ inside the point gap, we obtain an extensive number of modes localized at the flux core (see Fig. 3a,b). Such an extensive accumulation of states defines the NH *flux skin effect* [39].

In a PBC geometry with two flux cores, this response is topologically equivalent to the $\mathbb{Z}_2$ skin effect [39] of a 1D model in NH class AII$^\dagger$ situated on the line terminated by the two flux cores (see Fig. 3c). Indeed, the number of flux-localized modes scales with the length of the 1D edge connecting the two defects, denoted by $L_\perp$ (see App. C.1 for details on the finite-size scaling). The equivalence of the flux skin effect and the 1D NH bulk-boundary correspondence is consistent with the fact that point gap topology in 1D and NH class AII$^\dagger$ is $\mathbb{Z}_2$-classified [16]. This observation leads us to the following generalizing hypothesis, which we prove in App. B by exhaustion:

(I)      *A 2D system with nontrivial intrinsic point gap topology in NH symmetry class X, where X pins $\phi = 0, \pi$, exhibits a flux skin effect for $\phi = \pi$ iff X also has a nontrivial point gap classification in 1D.*

The flux response of the complete set of NH symmetry classes with nontrivial intrinsic point gap topology [39] is summarized in Tab. 1. Only the classes satisfying the above condition, including NH class AII$^\dagger$, realize a flux skin effect. An exhaustive collection of model Hamiltonians can be found in App. E.1.

We note in closing that there is a fundamental difference between crystal dislocations and flux defects [14]: Whereas lattice Burgers vectors are integer-valued, flux is only defined

mod $2\pi$. Consequently, since $\phi = +\pi$ and $\phi = -\pi$ are physically equivalent, both fluxes $\phi = \pm\pi$ of a PBC geometry have to collect skin modes, which is compatible only with a $\mathbb{Z}_2$-skin effect but not a $\mathbb{Z}$-skin effect [39]. In contrast, dislocations can give rise to both the $\mathbb{Z}$ and the $\mathbb{Z}_2$-skin effect [41].

# 4 NH flux spectral jump

We now study the $\pi$-flux response of intrinsically NH topological phases in 3D [39] in presence of TRS$^{(\dagger)}$ or PHS$^{(\dagger)}$. We begin with the example of a point-gapped NH system in class AII$^\dagger$, which is classified by a nontrivial 3D winding number $W_{3D}(E) \in \mathbb{Z}$ [16] (see App. E.2.1 for details). We derive the flux response of the simplest nontrivial case, where $W_{3D}(E_0) = 1$ for a given energy $E_0$ inside the point gap, from the corresponding EHH (see Sec. 2.2 and Fig. 4a). Irrespective of the choice of $E_0$, this EHH enjoys the symmetries of Hermitian class DIII (see App. B.5.2 and Tab. 3 therein), which is also $\mathbb{Z}$-classified in 3D. Hence, the EHH associated with $W_{3D}(E) = 1$ corresponds to a nontrivial topological insulator phase [59]. For this phase, it was shown in Ref. [10] that flux tubes with $\phi = \pi$ bind a 1D helical Majorana mode. This mode crosses zero energy at a single fixed TRS-invariant momentum, $k_\parallel \in \{0, \pi\}$, contributing two exact zero-energy states (see Fig. 4b). Importantly, no symmetry allowed perturbation is able to move these flux localized states away from zero energy at $k_\parallel = 0, \pi$. Correspondingly, there is a single flux-localized state at complex energy $E_0$ in the NH SIBC spectrum [39]. Repeating this construction for all $E_0$ inside the NH point gap yields an extensive number of states localized along the 1D flux tube for a single momentum $k_\parallel \in \{0, \pi\}$ (see Fig. 4c). Away from this momentum, the EHH remains gapped, corresponding to the absence of in-gap modes in the NH SIBC spectrum. The extensive pile-up of flux-localized states only for a single value of $k_\parallel$ constitutes a novel type of NH bulk-defect correspondence, which we term NH *flux spectral jump*.

In a PBC geometry with two flux tubes, this response is equivalent to that of a nontrivial 2D NH phase situated in the plane separating the flux tubes: NH class AII$^\dagger$ is $\mathbb{Z}_2$ classified in 2D [16] and protects an infernal point, i.e. the extensive accumulation of edge states only at a single momentum.[1] The number of flux localized modes scales with $L_\perp$, the distance between the two defects (see App. C.2 for details on the finite-size scaling). For OBC in the $z-$direction, where $k_\parallel$ is not conserved, the NH flux spectral jump still implies an accumulation of states at the flux tube. There may potentially also be surface states contributed by a surface skin effect. Next to their different real-space localization, the two effects can be distinguished in terms of the scaling with system size: whereas the surface skin effect is expected to localize $\mathcal{O}(L_\parallel = L_z)$ modes, the flux spectral jump localizes $\mathcal{O}(L_\perp)$ modes.

We note that *all* NH phases with nontrivial point gap topology are $\mathbb{Z}_2$-classified in 2D. On the other hand, they may be $\mathbb{Z}_2$- or $\mathbb{Z}$-classified in 3D. In the latter case, the flux spectral jump only arises when the $\mathbb{Z}$-valued invariant is odd. In our example of NH class AII$^\dagger$, this means that only systems where $W_{3D}(E) \bmod 2 \neq 0$ exhibit a flux response (see Tab. 1). Additional responses would be needed to probe the $\mathbb{Z}$ nature of $W_{3D}(E)$. The above observations lead us to the following hypothesis, which we prove in App. B by exhaustion:

(II)       *A 3D system with nontrivial intrinsic point gap topology in NH symmetry class X and invariant $w \in \mathbb{Z}_2$ or $\mathbb{Z}$, where X pins $\phi = 0, \pi$, shows a flux spectral jump for $\phi = \pi$ iff class X has a nontrivial point gap classification in 2D and $w \bmod 2 \neq 0$.*

---

[1] M. M. Denner, T. Neupert and F. Schindler, *in preparation*.

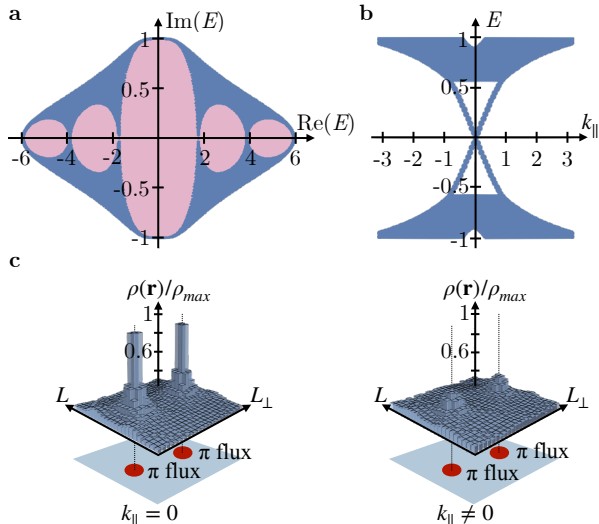

Figure 4: **NH flux spectral jump in 3D. a** Spectrum in the complex energy plane under PBC (blue) and SIBC in presence of $\pi$-flux tubes (blue and red) for a model exhibiting nontrivial point gap topology in NH class AII$^\dagger$ [Eq. (E.7)]. Flux-localized states fill the SIBC spectral point gap at a single momentum $k_\parallel = 0$ along the flux tube. **b** At this momentum, the EHH spectrum exhibits a helical crossing at zero energy for any $E_0$ located in the central point gap. **c** The flux spectral jump localizes an extensive number of eigenmodes at the flux tubes, indicated by two peaks of the summed density $\rho(r) = \sum_{\alpha,i} |\langle r_i | \psi_\alpha \rangle|^2$, where $\alpha$ ranges over all eigenstates $\psi_\alpha$ of the Hamiltonian with flux defect, $r$ denotes the lattice site and the summation $i$ runs over sublattice degrees of freedom.

The flux response of the complete set of NH symmetry classes with nontrivial intrinsic point gap topology [39] is summarized in Tab. 1. Only classes fulfilling the above condition, including NH class AII$^\dagger$, realize a flux spectral jump. An exhaustive collection of model Hamiltonians can be found in App. E.2.

## 5   NH higher-order flux skin effect

We next study the $\pi$-flux response of intrinsically NH topological phases in 3D [39] in the absence of TRS$^{(\dagger)}$ or PHS$^{(\dagger)}$. In order to still obtain a quantized flux response, we consider the presence of pseudo-inversion symmetry (see also App. A.1),

$$\mathcal{I}\mathcal{H}(k)^\dagger \mathcal{I}^\dagger = \mathcal{H}(-k), \tag{10}$$

which, in contrast to normal inversion symmetry, entails a Hermitian conjugation of the Hamiltonian $\mathcal{H}(k)$. As explained in App. D.2, pseudo-inversion represents the simplest crystalline symmetry that is compatible with nontrivial point-gap topology. In a PBC geometry with two inversion-related flux tubes, this symmetry implies that the magnitude of the two fluxes is the same: $\phi_1 = \phi_2$. At the same time, PBC leads to the constraint $\phi_1 + \phi_2 \mod 2\pi = 0$, thereby recovering the quantization of magnetic flux to $\phi = 0, \pi$. This argument only requires PBC *in the direction separating the flux tubes* (i.e. in our case in the $x$-direction); in fact, we will consider OBC in the direction *along* the flux tubes later on.[2] We note that the addition of

---

[2]For experimental settings with OBC in all directions, the magnetic flux $\phi$ is not quantized. For our theory to be applicable, the flux must then be fine-tuned to $\phi = \pi$.

Table 1: Flux response of all NH symmetry classes with intrinsic point gap topology in $d = 2, 3$. These are a subset of the 38 NH symmetry classes [16, 39]. Class labels refer to the Altland-Zirnbauer (AZ) classes, their AZ$^\dagger$ counterparts, and additional sublattice symmetry (SLS) [16]. The subscript of SLS, $S_\pm$, determines whether SLS commutes (+) or anticommutes (-) with time-reversal symmetry (TRS) and/or particle-hole symmetry (PHS). For the NH symmetry classes with both TRS and PHS (BDI, DIII, CII, and CI), the first subscript refers to TRS and the second one to PHS. All classes are identified by the square of TRS$^{(\dagger)}$, PHS$^{(\dagger)}$ and chiral symmetry CS. The topological classification is reproduced from Ref. [39], with $d_{\ell,1D}$ denoting the 1D line gap classification for the cases with trivial point-gapped phases in both 1D and 2D. $\mathcal{I}^\dagger$ denotes the additional presence and form of pseudo-inversion symmetry that, in NH symmetry classes without an anti-unitary operation, is needed to protect a topological flux response. The subscript of SLS/TRS next to the pseudo-inversion symmetry $\mathcal{I}^\dagger$, $S_\pm/T_\pm$, determines whether SLS/TRS commutes (+) or anticommutes (-) with $\mathcal{I}^\dagger$. $L_\perp$ denotes the distance between the flux tubes in a finite system, and $L_\parallel$ their length. We distinguish the flux skin effect (FSE) in 2D, the flux spectral jump (FSJ), higher-order flux skin effect (HO-FSE), the flux Majorana mode (FMM) and the higher-order flux Majorana mode (HO-FMM) in 3D.

| dim $d$ | class | symmetry | | | classification | | | | $\mathcal{I}^\dagger$ | $\pi$-flux response | # modes | App. |
| --- | --- | --- | --- | --- | --- | --- | --- | --- | --- | --- | --- | --- |
| | | TRS$^{(\dagger)}$ | PHS$^{(\dagger)}$ | CS | $d$ | $d-1$ | $d-2$ | $d_{\ell,1D}$ | | | | |
| 2 | AII$^\dagger$ | -1 | - | - | $\mathbb{Z}_2$ | $\mathbb{Z}_2$ | 0 | | - | FSE (Sec. 3) | $\mathcal{O}(L_\perp)$ | B.4.1 |
| 2 | DIII$^\dagger$ | -1 | +1 | 1 | $\mathbb{Z}_2$ | $\mathbb{Z}_2$ | 0 | | - | FSE (Sec. 3) | $\mathcal{O}(L_\perp)$ | B.4.2 |
| 2 | AIII$^{S_-}$ | - | - | 1 | $\mathbb{Z}_2$ | 0 | $\mathbb{Z}_2$ | | - | - | - | B.4.3 |
| 2 | BDI$^{S_{+-}}$ | +1 | +1 | 1 | $\mathbb{Z}_2$ | $\mathbb{Z}_2$ | $\mathbb{Z}_2$ | | - | FSE (Sec. 3) | $\mathcal{O}(L_\perp)$ | B.4.4 |
| 2 | D$^{S_-}$ | - | +1 | - | $\mathbb{Z}_2$ | $\mathbb{Z}_2$ | 0 | | - | FSE (Sec. 3) | $\mathcal{O}(L_\perp)$ | B.4.5 |
| 2 | DIII$^{S_{+-}}$ | -1 | +1 | 1 | $\mathbb{Z}_2$ | 0 | $\mathbb{Z}_2$ | | - | - | - | B.4.6 |
| 2 | CII$^{S_{-+}}$ | -1 | -1 | 1 | $\mathbb{Z}_2$ | 0 | $\mathbb{Z}_2$ | | - | - | - | B.4.7 |
| 2 | CII$^{S_{+-}}$ | -1 | -1 | 1 | $\mathbb{Z}_2$ | 0 | 0 | | - | - | - | B.4.8 |
| 2 | CI$^{S_{-+}}$ | +1 | -1 | 1 | $\mathbb{Z}_2$ | 0 | $\mathbb{Z}_2$ | | - | - | - | B.4.9 |
| 3 | AI$^\dagger$ | +1 | - | - | $2\mathbb{Z}$ | 0 | 0 | 0 | - | - | - | B.5.1 |
| 3 | AII$^\dagger$ | -1 | - | - | $\mathbb{Z}$ | $\mathbb{Z}_2$ | $\mathbb{Z}_2$ | | - | FSJ (Sec. 4) | $\mathcal{O}(L_\perp)$ | B.5.2 |
| 3 | A | - | - | - | $\mathbb{Z}$ | 0 | $\mathbb{Z}$ | | $\mathcal{I}^\dagger$ | HO-FSE (Sec. 5) | $\mathcal{O}(L_\parallel)$ | B.5.3 |
| 3 | A$^S$ | - | - | - | $\mathbb{Z}$ | 0 | $\mathbb{Z}$ | | $\mathcal{I}^{\dagger,S_-}$ | HO-FSE (Sec. 5) | $\mathcal{O}(L_\parallel)$ | B.5.4 |
| 3 | D | - | +1 | - | $\mathbb{Z}$ | 0 | 0 | $\mathbb{Z}_2$ | - | FMM (Sec. 6) | $\mathcal{O}(1)$ | B.5.5 |
| 3 | D$^{S_+}$ | - | +1 | - | $\mathbb{Z}$ | 0 | 0 | $\mathbb{Z}$ | $\mathcal{I}^{\dagger,S_-}$ | HO-FMM (Sec. 6) | $\mathcal{O}(1)$ | B.5.6 |
| 3 | D$^{S_-}$ | - | +1 | - | $\mathbb{Z}$ | $\mathbb{Z}_2$ | $\mathbb{Z}_2$ | | - | FSJ (Sec. 4) | $\mathcal{O}(L_\perp)$ | B.5.7 |
| 3 | DIII$^{S_{+-}}$ | -1 | +1 | 1 | $\mathbb{Z}_2$ | $\mathbb{Z}_2$ | 0 | | - | FSJ (Sec. 4) | $\mathcal{O}(L_\perp)$ | B.5.8 |
| 3 | AII$^{S_+}$ | -1 | - | - | $\mathbb{Z}_2$ | 0 | $\mathbb{Z}$ | | $\mathcal{I}^{\dagger,S_-,T_-}$ | HO-FSE (Sec. 5) | $\mathcal{O}(L_\parallel)$ | B.5.9 |
| 3 | C | - | -1 | - | $2\mathbb{Z}$ | 0 | 0 | 0 | - | - | - | B.5.10 |
| 3 | C$^{S_+}$ | - | -1 | - | $\mathbb{Z}$ | 0 | $\mathbb{Z}_2$ | | $\mathcal{I}^{\dagger,S_-}$ | HO-FSE (Sec. 5) | $\mathcal{O}(L_\parallel)$ | B.5.11 |
| 3 | C$^{S_-}$ | - | -1 | - | $\mathbb{Z}$ | 0 | 0 | 0 | - | - | - | B.5.12 |

pseudo-inversion symmetry may change the classification of NH point gap topology. However, in this work, we are only concerned with the flux response of point-gapped systems with local NH symmetries that are *enriched* by pseudo-inversion (see App. D.2).

We begin by considering the NH system discussed in Sec. 4 and App. E.2.1 *in absence of TRS$^\dagger$*. Recall that, in that section, we studied NH symmetry class AII$^\dagger$ which exhibits a flux spectral jump. Upon the relaxation of TRS$^\dagger$, we obtain NH symmetry class A, which is still classified by the same topological invariant $W_{3D}(E) \in \mathbb{Z}$ [16] (see App. E.2.2 for details). Our present example therefore again yields $W_{3D}(E_0) = 1$ for any given energy $E_0$ inside the point gap. As explained above, we furthermore assume pseudo-inversion symmetry.

We derive the flux response for this phase from the corresponding EHH (see Sec. 2.2). The EHH for NH class A retains the symmetries of Hermitian class AIII, irrespective of the choice of $E_0$ (see App. B.5.3 and Tab. 3 therein). Additionally, NH pseudo-inversion symmetry induces

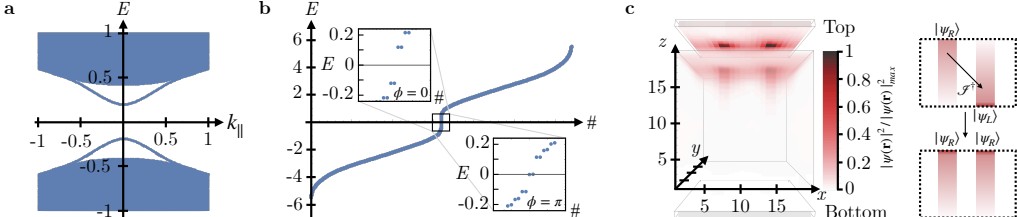

Figure 5: **NH higher-order flux skin effect in 3D. a** The energy spectrum of the EHH for a system with higher-order flux skin effect shows a gap for PBC along the flux tubes, here using a model in NH class A [Eq. (E.8)]. **b** Under OBC in the flux tube direction, the presence of a nontrivial flux $\phi = \pi$ introduces two exact zero-energy states in addition to the gapless Dirac cone surface dispersion, one for each flux tube. **c** In the NH system, the resulting flux skin modes are localized *at the same ends* of the two flux tubes. As explained in Sec. 5, this is a consequence of pseudo-inversion symmetry $\mathcal{I}^{\dagger}$, which maps states between the two flux tubes, but also changes their chirality. Only states with more than 85% support at the flux tubes are shown. All panels are generated for a model of size $20 \times 20 \times 20$ unit cells in NH class A with additional pseudo-inversion symmetry (see Sec. E.2.2).

a Hermitian inversion symmetry of the EHH. Hermitian class AIII is also $\mathbb{Z}$-classified in 3D, such that the EHH associated with a nontrivial point-gapped phase in NH class A must itself realize a nontrivial topological insulator phase in Hermitian class AIII [59]. Importantly, this class does not protect gapless modes along 1D flux tubes [10]. Hence, the PBC spectrum of the EHH in presence of two $\pi$-flux tubes is gapped (see Fig. 5a), where here we assume PBC both along and perpendicular to the flux tube directions. Correspondingly, the NH point gap shows no SIBC spectral in-gap modes. However, inversion symmetry can protect a flux response as soon as OBC are introduced in the direction of the flux tubes.

To derive this response, we first note that Ref. [44] showed that $W_{3D}(E_0) = 1$ is indicated by a double band inversion in the corresponding EHH when pseudo-inversion symmetry is present: to obtain a trivial (atomic limit) state, two occupied EHH inversion eigenvalues must flip sign at high-symmetry momenta in the 3D Brillouin zone. Without chiral symmetry (Hermitian class A), a double band inversion induces a higher-order topological (axion insulator) phase protected solely by inversion symmetry [60,61]. Such axion insulators exhibit a higher-order flux response: Ref. [14] derived that localized states appear at alternating *ends* of the two flux tubes, resulting in one state per flux tube for the EHH. In Hermitian class A, these states are generally not zero-modes, but instead contribute to a *filling anomaly* of the full inversion-symmetric system. Returning to the present case of Hermitian class AIII, chiral symmetry enforces that such flux end states arise at zero energy.

Besides pinning flux end states to zero energy, Hermitian class AIII differs in another important aspect from class A: it protects surface Dirac cones [59] that render the EHH OBC spectrum gapless even in absence of flux tubes (see Fig. 5b, left inset). However, the presence of such 2D surface states does not, in general, result in a skin effect in NH class A: The EHH surface Dirac cones are located at arbitrary surface momenta, which are in general not sampled over in the discrete surface Brillouin zone associated with any finite system size. Hence, there are no exact zero-energy states in the EHH spectrum.[3] Instead, each EHH Dirac cone implies

---

[3]One might similarly argue that flux bound states in realistic samples are always subject to a finite-size splitting, and therefore do not contribute to the NH spectrum: this splitting arises due to hybridization with other, spatially separated bound states. However, the gap resulting from such hybridization is exponentially small in the system size. On the other hand, the gap of a Dirac cone surface state that arises due to a finite-size quantization of crystal momentum decreases only algebraically with system size. Hence, in the thermodynamic limit and for an

a single exceptional point or single sheet of eigenvalues in the NH surface dispersion [21]. On the other hand, each flux tube introduces a single exact zero-mode per flux tube in the EHH. This constitutes a fractional flux response, since there exists no 1D Hermitian model with a single end state under OBC (see also App. D.1). The zero-energy flux end state of the EHH then corresponds to a flux-localized SIBC spectral in-gap state at complex energy $E_0$. Repeating this construction for all $E_0$ inside the NH point gap yields an extensive number of modes localized at the end of a flux tube (see Fig. 5c). In a finite system, the number of localized modes scales with the length of the flux tube, denoted by $L_\parallel$ (see App. C.2 for details on the finite-size scaling). This extensive pile-up of states on a region of dimension $d-3$ represents a NH *higher-order flux skin effect*.

Interestingly, the flux skin modes appear at the *same* end for both flux tubes, highlighting a uniquely NH feature (see Fig. 5c): if a right eigenstate skin mode localizes on the top of one flux tube, pseudo-inversion symmetry maps it to the bottom of the other flux tube. Due to the Hermitian transpose contained in the definition of pseudo-inversion (see App. A.1), this state is a left eigenstate of the NH Hamiltonian (see Fig. 5c, top-right). The corresponding right eigenstate is localized at the top end of the same flux tube [39] (see Fig. 5c, bottom-right).

Consequently, when the plane spanned between the two $\pi$-flux tubes is interpreted as a 1D NH system aligned along the flux tube direction, and with an extensively large unit cell along the perpendicular direction,[4] the higher-order flux skin effect is equivalent to the skin effect of a nontrivial 1D system [39] in NH class A. Due to pseudo-inversion symmetry, which maps between the two flux tubes, the resulting extensive number of end-localized modes is evenly distributed between the two flux tube ends, so that each flux tube on its own realizes *half* of a conventional 1D skin effect. On the other hand, the 2D point-gap classification of NH symmetry class A is trivial. This observation leads us to the following hypothesis, which we prove in App. B by exhaustion:

> (III)      *A 3D system with nontrivial point gap topology in NH symmetry class X and with pseudo-inversion symmetry exhibits a higher-order flux skin effect for $\phi = \pi$ iff the point gap classification of X is trivial in 2D but nontrivial in 1D.*

The flux response of the complete set of NH symmetry classes with nontrivial intrinsic point gap topology [39] is summarized in Tab. 1. Only classes fulfilling the above condition, including NH class A, show a NH higher-order flux skin effect. An exhaustive collection of model Hamiltonians can be found in App. E.2.

We note in closing that the higher-order flux skin effect realizes a novel flavour of NH topology that is fundamentally different from the Hermitian case: In a Hermitian system with gapless surface states, it is impossible to resolve a higher-order response to flux defects, because the nontrivial surface state would obscure any flux-localized modes. On the contrary the NH higher-order flux skin effect is observable: In addition to $\mathcal{O}(L_\perp \times L_\perp)$ NH surface modes deriving from gapless surface states, flux tubes induce a skin effect in $z$-direction, localizing another $\mathcal{O}(L_\parallel)$ modes on the surface, *only at the flux tube ends* (see App. C.2 for details on the finite-size scaling and App. D.3 for the implications on the EHH spectrum). Alternatively, one could reduce the extent of the flux tubes in the $z-$direction, thereby separating the flux response from the surface states.

---

exponentially small $\epsilon$, the $\epsilon$-pseudospectrum [38, 40] (Sec. 2.1) of the system discussed in the main text only exhibits a skin effect in presence of flux tubes. Even if surface modes do contribute to the pseudospectrum due to large experimental errors, they are still distributed across the entire surface. On the other hand, flux modes appear at the flux tubes only and therefore lead to an experimentally observable peak in the local density of states.

    [4]Reducing the separation of the two flux tubes will result in a phase transition (along the flux tubes) that removes all flux modes in the limit of zero distance.

# 6 Isolated first- and higher-order flux Majorana modes

All NH flux responses discussed so far have led to a SIBC spectrum with a completely filled point gap in presence of $\pi$-flux defects. Surprisingly, it is also possible to obtain *isolated* flux modes at specific energies within the point gap. These are similar to Hermitian bound states, but associated with NH point gap topology.

To understand this phenomenon, we consider NH systems in class D, which are classified by the $\mathbb{Z}$-valued 3D winding number $W_{3D}(E)$ [16] (see App. E.2.4 for details). We derive the flux response for a given energy $E_0$ inside the nontrivial point gap, in the simplest case with $W_{3D}(E_0) = 1$, from the corresponding EHH (see Sec. 2.2). The EHH for $E_0 = 0$ of NH class D retains the symmetries of Hermitian class DIII (see App. B.5.5 and Tab. 3 therein). Hermitian class DIII is also $\mathbb{Z}$-classified in 3D, such that the EHH associated with a nontrivial NH phase in class D must itself realize a nontrivial topological insulator phase in Hermitian class DIII [59]. For this phase, it was shown in Ref. [10] that flux tubes bind two degenerate zero-energy states (see Fig. 6a), corresponding to a single flux-localized state at $E_0 = 0$ in the NH SIBC spectrum (see Fig. 6b). Away from $E_0 = 0$, the EHH retains only a chiral symmetry, resulting in Hermitian class AIII (see App. B.5.5). The flux response of the EHH in Hermitian class AIII is trivial, such that the point gap remains empty for $E_0 \neq 0$ (see Fig. 6c).

We diagnose the nature of the unpaired NH flux state at $E_0 = 0$ by investigating (*in a gedankenexperiment*) its stability under coupling with a 1D line-gapped phase spanned between the two flux tubes. 1D line gap topology in NH class D is $\mathbb{Z}_2$-classified (see Tab. 1). The nontrivial phase is nothing but the NH generalization of the 1D Kitaev chain [39, 62], which hosts a single Majorana zero-mode at each end. Two zero-modes – the end state of the 1D line gap phase combined with the flux localized mode – are not protected. Consequently, the flux state must be a Majorana mode as well, and flux defects in NH class D can only probe $W_{3D}(E) \mod 2$. An unpaired *flux Majorana mode* remains pinned at zero complex energy. It is localized along the entire length of the flux tube (see Fig. 6d), highlighting another NH peculiarity: in Hermitian systems, spectrally isolated bound states are pointlike.

We next study NH class $D^{S_+}$, which is also classified by a $\mathbb{Z}$ invariant [16], the 3D winding number $W_{3D}(E)$ restricted to even values (see App. E.2.5 for details). Choosing an energy $E_0$ inside the nontrivial point gap with $W_{3D}(E_0) = 2$ allows to construct an EHH: for $E_0 = 0$ the EHH contains two interdependent unitary subspaces in Hermitian class AIII (see App. B.5.6 and Tab. 3 therein). Due to the $\mathbb{Z}$-classification of Hermitian class AIII in 3D, the EHH associated with a nontrivial phase in NH class $D^{S_+}$ must itself realize a nontrivial topological insulator phase *per unitary subspace* in Hermitian class AIII. However, without additional crystalline symmetries, the zero energy modes arising in the flux Dirac theory of systems in Hermitian class AIII are not protected [10] (see Fig. 6e). Adding an inversion symmetry (corresponding to pseudo-inversion in the NH Hamiltonian, App. D.2) quantizes the flux and allows for a stable flux response [14]: under OBC, each flux tube now hosts one zero energy mode *per* unitary subspace of the EHH. These flux end states then correspond to flux-localized NH SIBC spectral in-gap states at $E_0 = 0$ (see Fig. 6f and g). Away from $E_0 = 0$, the EHH is situated in Hermitian class CI: due to the absence of Kramers theorem, pairs of zero-energy states are no longer protected (see App. B.5.6). Consequently, the flux modes in each unitary subspace can be gapped, so that the higher order flux response persists only at $E_0 = 0$. The resulting NH *higher-order Majorana mode* localizes at the ends of the flux tubes (see Fig. 6h). Detecting its presence in a real system is, however, intricate, as it will in general be obscured by the NH surface state that is already present for $\phi = 0$.

We can understand the existence of (higher-order) flux Majorana modes by noting that, among all NH symmetry classes that do not exhibit a flux spectral jump or higher-order flux skin effect in 3D, NH class D and $D^{S_+}$ are the only ones with a nontrivial line gap classification

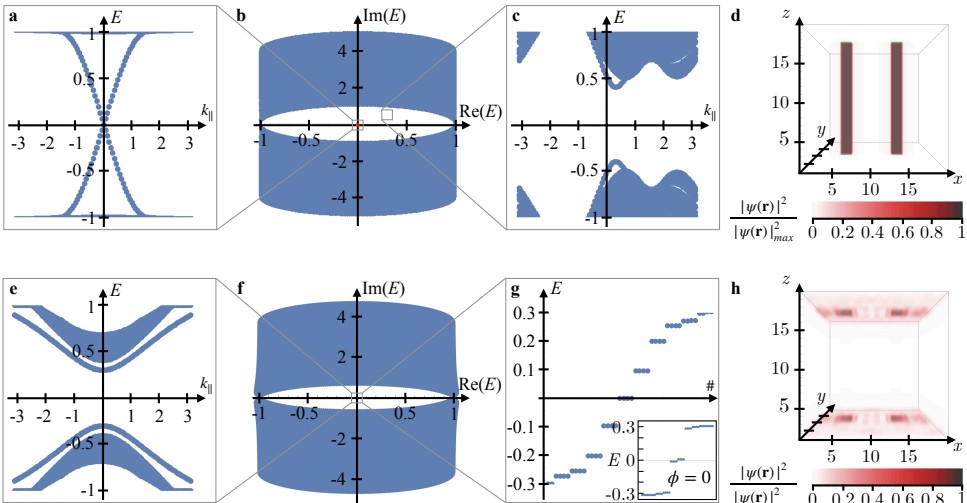

Figure 6: **NH flux Majorana modes in 3D. a** The energy spectrum of the EHH for NH class D realizes a helical metal for $E_0 = 0$. **b** This corresponds to *a single* Majorana mode at $E_0 = 0$ in the NH point gap. **c** Away from $E_0 = 0$, the EHH is gapped. This implies that the NH point gap does not fill up with flux-localized states, but only pins the Majorana mode. **d** The Majorana mode is localized along the entire flux defect ($k_\parallel = 0$), a scenario that is not possible for a single mode in a Hermitian system. **e** The energy spectrum of the EHH for NH class $D^{S_+}$ with pseudo-inversion symmetry is gapped in presence of two PBC-preserving $\pi$-fluxes. **f** This results in an empty point gap in the NH spectrum. **g** When terminating the system along a surface perpendicular to the flux tubes, the EHH features four exact zero modes at $E_0 = 0$, which are absent when the flux tubes are removed. (Note that in this limit, approximate zero-modes remain due to the gapless surface states in this symmetry class, highlighted in the inset for $\phi = 0$.) **h** The two resulting NH higher-order Majorana modes are then localized at the *ends* of the flux defects. Panels a-d are generated for a model of size $20 \times 20$ ($\times 20$ in OBC) unit cells in NH class D (see Sec. E.2.4), panels e-h for a model of size $20 \times 20$ ($\times 20$ in OBC) unit cells in NH class $D^{S_+}$ (see Sec. E.2.5).

(see Tab. 1) in 1D. As line-gapped phases are adiabatically deformable to Hermitian systems, flux tubes in these classes can only cause the presence of isolated modes (similar to the end states of 1D Hermitian topological insulators) instead of NH skin effects.

# 7 Experimental realization of NH flux response

In a solid-state setting, we usually assume isolated systems that are governed by a Hermitian Hamiltonian. Most meta-material platforms are however either accidentally or tunably lossy, such that the description with an effective Hamiltonian involves NH terms. The same holds for interacting electronic quantum systems in which quasiparticles acquire a finite lifetime. Such a scenario equips the single-electron Green's function with a complex self-energy, leading to an effective NH Hamiltonian [63–68]. Consequently, NH systems appear quite naturally in experimental settings.

We now outline the NH platforms in which flux defects can be studied experimentally. Superconductors are natural starting points for studying flux defects, as magnetic fields penetrate samples as flux vortices. Consequently, superconducting NH systems in 2D or those in direct

proximity to a superconductor may experience a *flux skin effect* in presence of vortices. The size of vortices should be matched with the lattice scale by the use of moire substrates/potentials, allowing for precisely localized defects and quantized responses. However, in general we expect flux localized states to not disappear immediately with increasing vortex size, but rather to spread out over the vortex region. In 3D, the prototypical NH point-gapped phase is the exceptional topological insulator (ETI) [21]. The ETI emerges naturally from a Hermitian 3D topological insulator or Weyl semimetal, if quasi-particles acquire a finite lifetime. This could for instance be caused by electron or electron-phonon interactions [21]. As the ETI does not require any symmetry to be stabilized, it is naturally situated in NH class A. With additional pseudo-inversion symmetry, it can thus give rise to a *higher-order flux skin effect* upon flux insertion. Additionally, the ETI phase was shown to arise in NH class AII[†] [21], thereby allowing for the flux realization of the NH *flux spectral jump*. Consequently, the ETI serves as the ideal platform to investigate 3D flux defects. Due to their versatility, meta-materials provide convenient classical analogs to quantum mechanical topological states. A flux defect can for instance be realized in electrical meta-materials [35,69] by introducing operational amplifiers. These equip arbitrary hoppings $t$ with a tuneable phase $t \to te^{i\phi}$, thereby allowing to implement $\phi$ flux tubes [70]. Alternatively, one might realize effective magnetic fluxes in photonic [71], mechanical [72], and ultra-cold atom systems [73,74]. Additionally, one can envision a realization in NH superconductors [75–77]. Such systems, for instance in NH class $C^{S_+}$ and $DIII^{S_{+-}}$, can give rise to both *higher-order flux skin effect* and *flux spectral jump* upon flux insertion, respectively. Similarly, the NH *flux Majorana mode* appears in a NH superconductor in class D.

## 8 Discussion

We have derived the flux response of all NH symmetry classes with intrinsic point gap topology in 2D and 3D. In 2D, we found the NH *flux skin effect*, in which a macroscopic number of states localizes at the flux core. We identified the necessary conditions for its emergence and predict its occurrence for four NH symmetry classes. In 3D, we discovered the *flux spectral jump*, where a $\pi$-flux tube causes a NH skin effect only at a *single* momentum along the flux tube direction. This response derives from a corresponding novel 2D phase exhibiting an infernal point, where an extensive number of states collapses to the boundary at a single momentum. Developing a physical understanding for this phenomenon, including a field-theoretical description, is an important future endeavor. Additionally, we found NH symmetry classes giving rise to a *higher-order flux skin effect*, which occurs at the *ends* of flux tubes. Being a higher-order response, it relies on the presence of a crystalline symmetry. We here considered the case of pseudo-inversion symmetry, but expect further exciting responses for other crystalline symmetries. Finally, we identified the presence of *flux Majorana modes*, forming spectrally isolated mid-gap states in the NH point gap. Realizing such modes in open quantum systems is an interesting question for future theoretical and experimental research.

## Acknowledgments

We thank Shinsei Ryu, Titus Neupert, Tomáš Bzdušek, Abhinav Prem and Kohei Kawabata for helpful discussions and valuable comments on the manuscript. F.S. also thanks Stepan Tsirkin, Titus Neupert, Andrei Bernevig, and Benjamin Wieder for previous collaboration on a related subject. This project has received funding from the European Research Council (ERC) under

the European Union's Horizon 2020 research and innovation programm (ERC-StG-Neupert-757867-PARATOP). M.M.D. acknowledges support from the Graduate Research Campus of the University of Zurich and thanks the Princeton Center for Theoretical Science for hosting during some stages of this work. F.S. was supported by a fellowship at the Princeton Center for Theoretical Science.

# A  Symmetries in NH systems

This section defines symmetries in NH systems and their relation to the corresponding symmetries of the EHH. Hermitian quantities are denoted with an overline.

## A.1  NH symmetries

The Hermitian time-reversal symmetry given as

$$\bar{U}_{\mathcal{T}}\bar{\mathcal{H}}(\boldsymbol{k})^{*}\bar{U}_{\mathcal{T}}^{\dagger} = \bar{\mathcal{H}}(-\boldsymbol{k}), \qquad \bar{U}_{\mathcal{T}}\bar{U}_{\mathcal{T}}^{*} = \pm 1, \tag{A.1}$$

is generalized to a NH time-reversal symmetry TRS

$$U_{\mathcal{T}}\mathcal{H}(\boldsymbol{k})^{*}U_{\mathcal{T}}^{\dagger} = \mathcal{H}(-\boldsymbol{k}), \qquad U_{\mathcal{T}}U_{\mathcal{T}}^{*} = \pm 1, \tag{A.2}$$

as well as a pseudo time-reversal symmetry TRS$^{\dagger}$

$$U_{\mathcal{T}}\mathcal{H}(\boldsymbol{k})^{T}U_{\mathcal{T}}^{\dagger} = \mathcal{H}(-\boldsymbol{k}), \qquad U_{\mathcal{T}}U_{\mathcal{T}}^{*} = \pm 1. \tag{A.3}$$

The Hermitian particle-hole symmetry given as

$$\bar{U}_{\mathcal{P}}\bar{\mathcal{H}}(\boldsymbol{k})^{*}\bar{U}_{\mathcal{P}}^{\dagger} = -\bar{\mathcal{H}}(-\boldsymbol{k}), \qquad \bar{U}_{\mathcal{P}}\bar{U}_{\mathcal{P}}^{*} = \pm 1, \tag{A.4}$$

is generalized to a NH particle-hole symmetry PHS

$$U_{\mathcal{P}}\mathcal{H}(\boldsymbol{k})^{T}U_{\mathcal{P}}^{\dagger} = -\mathcal{H}(-\boldsymbol{k}), \qquad U_{\mathcal{P}}U_{\mathcal{P}}^{*} = \pm 1, \tag{A.5}$$

as well as a pseudo particle-hole symmetry PHS$^{\dagger}$

$$U_{\mathcal{P}}\mathcal{H}(\boldsymbol{k})^{*}U_{\mathcal{P}}^{\dagger} = -\mathcal{H}(-\boldsymbol{k}), \qquad U_{\mathcal{P}}U_{\mathcal{P}}^{*} = \pm 1. \tag{A.6}$$

The Hermitian chiral symmetry given as

$$\bar{U}_{\mathcal{C}}\bar{\mathcal{H}}(\boldsymbol{k})\bar{U}_{\mathcal{C}}^{\dagger} = -\bar{\mathcal{H}}(\boldsymbol{k}), \qquad \bar{U}_{\mathcal{C}}^{2} = 1, \tag{A.7}$$

is generalized to a NH chiral symmetry CS

$$U_{\mathcal{C}}\mathcal{H}(\boldsymbol{k})^{\dagger}U_{\mathcal{C}}^{\dagger} = -\mathcal{H}(\boldsymbol{k}), \qquad U_{\mathcal{C}}^{2} = 1. \tag{A.8}$$

Additonally, one can have sublattice symmetry SLS, defined by

$$\mathcal{S}\mathcal{H}(\boldsymbol{k})\mathcal{S}^{\dagger} = -\mathcal{H}(\boldsymbol{k}), \qquad \mathcal{S}^{2} = 1, \tag{A.9}$$

as well as pseudo-Hermiticity

$$\eta\mathcal{H}(\boldsymbol{k})^{\dagger}\eta^{\dagger} = \mathcal{H}(\boldsymbol{k}), \qquad \eta^{2} = 1. \tag{A.10}$$

Finally, we investigate the presence of inversion

$$\mathcal{I}\mathcal{H}(\boldsymbol{k})\mathcal{I}^{\dagger} = \mathcal{H}(-\boldsymbol{k}), \tag{A.11}$$

and pseudo-inversion symmetry

$$\mathcal{I}\mathcal{H}(\boldsymbol{k})^{\dagger}\mathcal{I}^{\dagger} = \mathcal{H}(-\boldsymbol{k}). \tag{A.12}$$

## A.2 Appearance of NH symmetries in the EHH

The presence of symmetries for the NH Hamiltonian $\mathcal{H}(\boldsymbol{k})$ imposes constraints on the EHH $\bar{\mathcal{H}}(\boldsymbol{k})$,

$$\bar{\mathcal{H}}(\boldsymbol{k}) = \begin{pmatrix} 0 & \mathcal{H}(\boldsymbol{k}) - E_0 \\ \mathcal{H}^\dagger(\boldsymbol{k}) - E_0^* & 0 \end{pmatrix}. \tag{A.13}$$

Specifically, for $E_0 = 0$, it holds:

$$\bar{U}_{\mathcal{T}} \bar{\mathcal{H}}(\boldsymbol{k})^* \bar{U}_{\mathcal{T}}^\dagger = \bar{\mathcal{H}}(-\boldsymbol{k}), \tag{A.14}$$

with

$$\bar{U}_{\mathcal{T}} = \begin{pmatrix} U_{\mathcal{T}} & 0 \\ 0 & U_{\mathcal{T}} \end{pmatrix}, \tag{A.15}$$

for TRS and

$$\bar{U}_{\mathcal{T}} = \begin{pmatrix} 0 & U_{\mathcal{T}} \\ U_{\mathcal{T}} & 0 \end{pmatrix}, \tag{A.16}$$

for TRS$^\dagger$ of the NH Hamiltonian.

$$\bar{U}_{\mathcal{P}} \bar{\mathcal{H}}(\boldsymbol{k})^* \bar{U}_{\mathcal{P}}^\dagger = -\bar{\mathcal{H}}(-\boldsymbol{k}), \tag{A.17}$$

with

$$\bar{U}_{\mathcal{P}} = \begin{pmatrix} 0 & U_{\mathcal{P}} \\ U_{\mathcal{P}} & 0 \end{pmatrix}, \tag{A.18}$$

for PHS and

$$\bar{U}_{\mathcal{P}} = \begin{pmatrix} U_{\mathcal{P}} & 0 \\ 0 & U_{\mathcal{P}} \end{pmatrix}, \tag{A.19}$$

for PHS$^\dagger$ of the NH Hamiltonian. Additionally we have

$$\bar{U}_{\mathcal{C}} \bar{\mathcal{H}}(\boldsymbol{k}) \bar{U}_{\mathcal{C}}^\dagger = -\bar{\mathcal{H}}(\boldsymbol{k}), \qquad \bar{U}_{\mathcal{C}} = \begin{pmatrix} 0 & U_{\mathcal{C}} \\ U_{\mathcal{C}} & 0 \end{pmatrix}, \tag{A.20}$$

$$\bar{\mathcal{S}} \bar{\mathcal{H}}(\boldsymbol{k}) \bar{\mathcal{S}}^\dagger = -\bar{\mathcal{H}}(\boldsymbol{k}), \qquad \bar{\mathcal{S}} = \begin{pmatrix} \mathcal{S} & 0 \\ 0 & \mathcal{S} \end{pmatrix}, \tag{A.21}$$

$$\bar{\eta} \bar{\mathcal{H}}(\boldsymbol{k})^\dagger \bar{\eta}^\dagger = \bar{\mathcal{H}}(\boldsymbol{k}), \qquad \bar{\eta} = \begin{pmatrix} 0 & \eta \\ \eta & 0 \end{pmatrix}. \tag{A.22}$$

By construction, $\bar{\mathcal{H}}(\boldsymbol{k})$ enjoys an additional chiral (sublattice) symmetry for arbitrary $E_0$:

$$\bar{\Sigma}_{\mathcal{C}} \bar{\mathcal{H}}(\boldsymbol{k}) \bar{\Sigma}_{\mathcal{C}}^\dagger = -\bar{\mathcal{H}}(\boldsymbol{k}), \quad \bar{\Sigma}_{\mathcal{C}} = \begin{pmatrix} \mathbb{1} & 0 \\ 0 & -\mathbb{1} \end{pmatrix}. \tag{A.23}$$

Besides the chiral symmetry introduced in (A.23), sublattice symmetry $\mathcal{S}$ and PHS $U_{\mathcal{P}}$ allow a TRS

$$\bar{U}_{\mathcal{T}} \bar{\mathcal{H}}(\boldsymbol{k})^* \bar{U}_{\mathcal{T}}^\dagger = \bar{\mathcal{H}}(-\boldsymbol{k}), \tag{A.24}$$

with

$$\bar{U}_{\mathcal{T}} = \begin{pmatrix} 0 & \mathcal{S} U_{\mathcal{P}} \\ \mathcal{S} U_{\mathcal{P}} & 0 \end{pmatrix}, \tag{A.25}$$

and a PHS

$$\bar{U}_{\mathcal{P}} \bar{\mathcal{H}}(\boldsymbol{k})^* \bar{U}_{\mathcal{P}}^\dagger = -\bar{\mathcal{H}}(-\boldsymbol{k}), \tag{A.26}$$

with

$$\bar{U}_{\mathcal{P}} = \begin{pmatrix} 0 & \mathcal{S}U_{\mathcal{P}} \\ -\mathcal{S}U_{\mathcal{P}} & 0 \end{pmatrix}. \tag{A.27}$$

For daggered NH symmetry classes, TRS$^{\dagger}$

$$\bar{U}_{\mathcal{T}}\bar{\mathcal{H}}(\boldsymbol{k})^{*}\bar{U}_{\mathcal{T}}^{\dagger} = \bar{\mathcal{H}}(-\boldsymbol{k}), \tag{A.28}$$

with

$$\bar{U}_{\mathcal{T}} = \begin{pmatrix} 0 & U_{\mathcal{T}} \\ U_{\mathcal{T}} & 0 \end{pmatrix}, \tag{A.29}$$

can be combined with (A.23) to a PHS

$$\bar{U}_{\mathcal{P}}\bar{\mathcal{H}}(\boldsymbol{k})^{*}\bar{U}_{\mathcal{P}}^{\dagger} = -\bar{\mathcal{H}}(-\boldsymbol{k}), \tag{A.30}$$

with

$$\bar{U}_{\mathcal{P}} = \begin{pmatrix} 0 & U_{\mathcal{T}} \\ -U_{\mathcal{T}} & 0 \end{pmatrix}. \tag{A.31}$$

# B  Classification of flux response in all NH point-gapped symmetry classes

We classify the flux response of all NH systems with nontrivial intrinsic point gap topology. Since topological zero energy eigenvalues of the EHH $\bar{\mathcal{H}}(k)$ correspond to states at $E_0$ within the NH point gap, we first investigate the flux response of Hermitian systems. This procedure relies on a Dirac treatment of the flux defect, introduced in App. B.1. We employ this procedure for all relevant Hermitian symmetry classes in 2D (App. B.2) and 3D (App. B.3). The corresponding NH flux response follows from protected EHH zero energy modes, derived in App. B.4 for 2D and App. B.5 for 3D.

## B.1  Dirac theory of Hermitian flux response

For a Hermitian topological insulator in one of the 10 Altland-Zirnbauer (AZ) symmetry classes [78, 79], we can derive the response for a $\phi = \pi$ flux using the Dirac theory of its edge states [80].

To derive the flux response of 2D (3D) topological insulators, we first cut the 2D (3D) bulk in half (see Fig. 7**a**) to create two sets of edge (surface) states, one for the top layer and one for the bottom layer. In general, these will be described by a 1D (2D) Dirac Hamiltonian

$$\bar{H}_{1D}(k) = \tau_z \otimes \bar{h}(k) + M, \tag{B.1}$$

$$\bar{H}_{2D}(\mathbf{k}) = \tau_z \otimes \bar{h}(\mathbf{k}) + M, \tag{B.2}$$

where $k$ is the momentum along the edge and $\mathbf{k}$ the momenta on the surface, with $\tau_z$ a Pauli matrix acting on layer space. In the presence of $N$ edge (surface) states, $\bar{h}(k)$ is a $N \times N$ matrix capturing the edge (surface) state dispersion, and $M$ is a $2N \times 2N$ mass matrix that couples the edge (surface) states to yield a gapped bulk. The insertion of a flux core (flux tube) $\phi$ can be viewed as a modification of all hoppings crossing a line (plane) emanating from the flux core (flux tube), multiplying all such hopping amplitudes with a Peierls phase $t \to te^{i\phi}$. For a flux of $\phi = \pi$, this corresponds to a sign flip [14]. The mass term $M$ coupling the two layers therefore changes sign at the flux tube, forming a domain wall binding point-like (line-like) states (see Fig. 7**b**,**c**). In the following, we therefore consider a flux-Dirac theory for all relevant Altland-Zirnbauer classes.

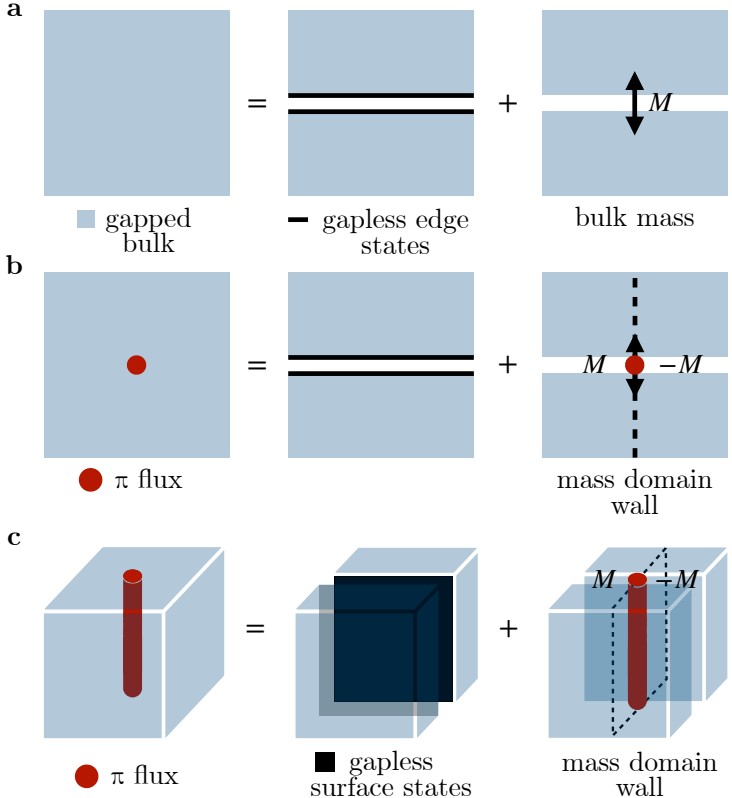

Figure 7: **Dirac theory of flux response. a** The gapped bulk of a topological insulator can be viewed as a combination of two subsystems, whose edge states are gapped by a bulk mass term. **b** Bound states localized at $\pi$-fluxes in Hermitian insulators can be viewed as domain wall bound states of the Dirac mass associated with a pair of edge states [14]. **c** In 3D, $\pi$-fluxes similarly bind 1D states.

## B.2 2D systems, symmetry classes of $\bar{\mathcal{H}}(k)$

In 2D, the EHH $\bar{\mathcal{H}}(k)$ for nontrivial intrinsic point-gapped NH models resides in Hermitian classes A, AII, D, DIII or C (see Tab. 2). In the following, we investigate whether a flux induced mass domain wall leads to zero energy modes of $\bar{\mathcal{H}}(k)$.

### B.2.1 Class A

The prototypical model in Hermitian class A in 2D is a Chern insulator, whose edge Hamiltonian is described in the nontrivial phase by

$$\bar{h}(k) = k, \tag{B.3}$$

where $k$ is the momentum along the edge. The 1D Dirac Hamiltonian can then be written as

$$\bar{H}(k) = \tau_z \otimes \bar{h}(k) + mx\tau_x + \delta\tau_y, \tag{B.4}$$

where $mx$ implements the flux core induced domain wall in the mass term at $x = 0$, $\tau_\mu$ are Pauli matrices ($\mu = 0, x, y, z$) and $\delta$ multiplies a symmetry allowed perturbation. In order to derive the presence of topological zero energy states, we consider OBC along the edge ($x-$direction) and solve for boundary-localized zero-energy states determined by the Hamiltonian

$$\bar{H}_x = -i\tau_z\frac{\partial}{\partial x} + mx\tau_x + \delta\tau_y. \tag{B.5}$$

Table 2: NH symmetry classes with intrinsic point gap topology in 2D and the corresponding Hermitian classes of the EHH.

| dim | class $H$ | class $\bar{H}$ ($E_0 = 0$) | class $\bar{H}$ ($E_0 \neq 0$) | App. |
|---|---|---|---|---|
| 2 | AII$^\dagger$ | DIII | DIII | B.4.1 |
| 2 | DIII$^\dagger$ | D | DIII | B.4.2 |
| 2 | AIII$^{S_-}$ | A $\oplus$ A | AIII | B.4.3 |
| 2 | BDI$^{S_{+-}}$ | D $\oplus$ D | DIII | B.4.4 |
| 2 | D$^{S_-}$ | DIII $\oplus$ DIII | DIII | B.4.5 |
| 2 | DIII$^{S_{+-}}$ | AII $\oplus$ AII | DIII | B.4.6 |
| 2 | CII$^{S_{-+}}$ | A | DIII | B.4.7 |
| 2 | CII$^{S_{+-}}$ | C $\oplus$ C | CI | B.4.8 |
| 2 | CI$^{S_{-+}}$ | A | DIII | B.4.9 |

For $\delta = 0$ we can find one normalizable zero energy solution $|\psi\rangle$, $\bar{H}_x|\psi\rangle = 0$, given by

$$|\psi\rangle = \frac{1}{N} e^{-\frac{1}{2}mx^2} \begin{pmatrix} i \\ 1 \end{pmatrix}. \tag{B.6}$$

To study the effect of the perturbation $\delta$, we rely on first-order perturbation theory

$$\langle\psi|\bar{H}_x|\psi\rangle = -\delta. \tag{B.7}$$

Consequently, the EHH for flux cores in Hermitian class A ($d = 2$) has no zero energy mode, as any finite $\delta$ is able to remove the in gap state. This means the flux response of 2D NH systems that lead to Hermitian class A for $\bar{\mathcal{H}}(k)$ is trivial. Note that additional crystalline symmetries do not change this statement, contrary to the case of 3D.

### B.2.2 Class AII

Hermitian class AII has a TRS with $\bar{U}_\mathcal{T}\bar{U}_\mathcal{T}^* = -1$, describing 2D topological insulators. As such, the surface Hamiltonian of the nontrivial phase is gapless,

$$\bar{h}(k) = k\sigma_x, \tag{B.8}$$

where $k$ is the momentum along the edge and mass terms are forbidden by the presence of TRS $\bar{U}_\mathcal{T} = \sigma_y$. The 1D Dirac Hamiltonian can then be written as

$$\bar{H}(k) = \tau_z \otimes \bar{h}(k) + mx\tau_x\sigma_0 + \delta O, \tag{B.9}$$

where $mx$ implements the flux core induced domain wall in the mass term at $x = 0$, $\tau_\mu, \sigma_\mu$ are Pauli matrices ($\mu = 0, x, y, z$) and $\delta$ multiplies a collection of symmetry allowed perturbation terms $O$. In order to derive the presence of topological zero energy states, we consider OBC along the edge ($x-$direction) and solve for boundary-localized zero-energy states determined by the Hamiltonian

$$\bar{H}_x = -i\tau_z\sigma_x\frac{\partial}{\partial x} + mx\tau_x\sigma_0 + \delta O. \tag{B.10}$$

For $\delta = 0$ we can find two normalizable zero energy solutions $|\psi\rangle$, $\bar{H}_x|\psi\rangle = 0$, given by

$$|\psi_1\rangle = \frac{1}{N} e^{-\frac{1}{2}mx^2} \begin{pmatrix} i \\ 0 \\ 0 \\ 1 \end{pmatrix}, \quad |\psi_2\rangle = \frac{1}{N} e^{-\frac{1}{2}mx^2} \begin{pmatrix} 0 \\ i \\ 1 \\ 0 \end{pmatrix}. \tag{B.11}$$

In order to reveal that symmetry allowed perturbations $O$ are able to gap out these modes, we rely on degenerate first-order perturbation theory in $\delta$. Fixing $O = \tau_0\sigma_0$ as one symmetry preserving choice under TRS $\bar{U}_{\mathcal{T}} = \tau_0\sigma_y$, we obtain

$$\left[\bar{H}_{\text{flux}}\right]_{mn} = \langle\psi_m|\bar{H}_x|\psi_n\rangle = [\delta\sigma_0]_{mn}\,. \tag{B.12}$$

Consequently, the EHH for flux cores in Hermitian class AII ($d = 2$) is gapped and hosts no zero energy modes. This means the flux response of 2D NH systems that lead to Hermitian class AII for $\bar{\mathcal{H}}(k)$ is trivial.

### B.2.3  Class D

Hermitian class D describes the thermal quantum Hall effect, with a PHS with $\bar{U}_{\mathcal{P}}\bar{U}_{\mathcal{P}}^* = +1$. The surface Hamiltonian for the nontrivial phase is gapless,

$$\bar{h}(k) = k\,, \tag{B.13}$$

where $k$ is the momentum along the edge. The 1D Dirac Hamiltonian follows as

$$\bar{H}(k) = \tau_z \otimes \bar{h}(k) + mx\tau_x\,, \tag{B.14}$$

where $mx$ implements the flux core induced domain wall in the mass term at $x = 0$, $\tau_\mu$ are Pauli matrices ($\mu = 0, x, y, z$) and further symmetry allowed perturbation terms $O$ are not allowed by the presence of $\bar{U}_{\mathcal{P}} = \tau_z$. In order to derive the presence of topological zero energy states, we consider OBC along the edge ($x$−direction) and solve for boundary-localized zero-energy states determined by the Hamiltonian

$$\bar{H}_x = -i\tau_z\frac{\partial}{\partial x} + mx\tau_x\,. \tag{B.15}$$

We can find one normalizable zero energy solution $|\psi\rangle$, $\bar{H}_x|\psi\rangle = 0$, given by

$$|\psi\rangle = \frac{1}{N}e^{-\frac{1}{2}mx^2}\begin{pmatrix} i \\ 1 \end{pmatrix}\,. \tag{B.16}$$

Consequently, the EHH for flux cores in Hermitian class D ($d = 2$) is gapless, and hosts one topologically protected zero mode localized at the flux core.

### B.2.4  Class DIII

In Hermitian symmetry class DIII, the surface Hamiltonian along the edge $\bar{h}(k)$ is gapless in the nontrivial phase and assumes the form

$$\bar{h}(k) = k\sigma_x\,, \tag{B.17}$$

where $k$ is the momentum along the edge and mass terms are forbidden by the presence of TRS, PHS and chiral symmetry. Choosing chiral symmetry as $\bar{U}_{\mathcal{C}} = \tau_z\sigma_z$, the 1D Dirac Hamiltonian can be written as

$$\bar{H}(k) = \tau_z \otimes \bar{h}(k) + mx\tau_x\sigma_0 + \delta O\,, \tag{B.18}$$

where $mx$ implements the flux core induced domain wall in the mass term at $x = 0$, $\tau_\mu, \sigma_\mu$ are Pauli matrices ($\mu = 0, x, y, z$) and $\delta$ multiplies a collection of symmetry allowed perturbation terms $O$. In order to derive the presence of topological zero energy states, we consider OBC along the edge ($x$−direction) and solve for boundary-localized zero-energy states determined by the Hamiltonian

$$\bar{H}_x = -i\tau_z\sigma_x\frac{\partial}{\partial x} + mx\tau_x\sigma_0 + \delta O. \tag{B.19}$$

For $\delta = 0$ we can find two normalizable zero energy solutions $|\psi\rangle$, $\bar{H}_x|\psi\rangle = 0$, given by

$$|\psi_1\rangle = \frac{1}{N}e^{-\frac{1}{2}mx^2}\begin{pmatrix} i \\ 0 \\ 0 \\ 1 \end{pmatrix}, \quad |\psi_2\rangle = \frac{1}{N}e^{-\frac{1}{2}mx^2}\begin{pmatrix} 0 \\ i \\ 1 \\ 0 \end{pmatrix}. \tag{B.20}$$

In order to reveal that symmetry allowed perturbations $O$ are not able to gap out these modes, we rely on degenerate first-order perturbation theory in $\delta$. Fixing $O = \tau_y\sigma_z$ as the only symmetry preserving choice under TRS $\bar{U}_{\mathcal{T}} = \tau_0\sigma_y$ and PHS $\bar{U}_{\mathcal{P}} = \tau_z\sigma_x$, we obtain

$$\left[\bar{H}_{\text{flux}}\right]_{mn} = \langle\psi_m|\bar{H}_x|\psi_n\rangle = 0. \tag{B.21}$$

Consequently, the EHH for flux cores in Hermitian class DIII ($d = 2$) is gapless, and hosts two topologically protected zero modes localized at the flux cores.

### B.2.5 Class C

In Hermitian symmetry class C with $\bar{U}_{\mathcal{P}}\bar{U}_{\mathcal{P}}^* = -1$, the surface Hamiltonian along the edge $\bar{h}(k)$ is gapless in the nontrivial phase and assumes the form

$$\bar{h}(k) = k\sigma_0 + \tilde{m}_1\sigma_x + \tilde{m}_2\sigma_y + \tilde{m}_3\sigma_z, \tag{B.22}$$

where $k$ is the momentum along the edge and the terms multiplied by $\tilde{m}_{1,2,3}$ only shift the zero energy crossing. Using $\bar{U}_{\mathcal{P}} = \tau_z\sigma_y$, the 1D Dirac Hamiltonian can then be written as

$$\bar{H}(k) = \tau_z \otimes \bar{h}(k) + mx\tau_x\sigma_0 + \delta O, \tag{B.23}$$

where $mx$ implements the flux core induced domain wall in the mass term at $x = 0$, $\tau_\mu, \sigma_\mu$ are Pauli matrices ($\mu = 0, x, y, z$) and $\delta$ multiplies a collection of symmetry allowed perturbation terms $O$. In order to derive the presence of topological zero energy states, we consider OBC along the edge ($x-$direction) and solve for boundary-localized zero-energy states determined by the Hamiltonian

$$\bar{H}_x = -i\tau_z\sigma_0\frac{\partial}{\partial x} + mx\tau_x\sigma_0 + \delta O. \tag{B.24}$$

For $\delta = 0$ we can find two normalizable zero energy solutions $|\psi\rangle$, $\bar{H}_x|\psi\rangle = 0$, given by

$$|\psi_1\rangle = \frac{1}{N}e^{-\frac{1}{2}mx^2}\begin{pmatrix} 0 \\ i \\ 0 \\ 1 \end{pmatrix}, \quad |\psi_2\rangle = \frac{1}{N}e^{-\frac{1}{2}mx^2}\begin{pmatrix} i \\ 0 \\ 1 \\ 0 \end{pmatrix}. \tag{B.25}$$

To study the influence of symmetry allowed perturbations $O$, we rely on degenerate first-order perturbation theory in $\delta$. Fixing $O = \tau_0\sigma_z$ as one symmetry preserving choice under PHS $\bar{U}_{\mathcal{P}} = \tau_z\sigma_y$, we obtain

$$\left[\bar{H}_{\text{flux}}\right]_{mn} = \langle\psi_m|\bar{H}_x|\psi_n\rangle = [-\delta\sigma_z]_{mn}. \tag{B.26}$$

Consequently, the EHH for flux cores in Hermitian class C ($d = 2$) is gapped, and hosts no zero modes localized at the flux cores. This is a consequence of PHS being able to protect only a single state, not two.

Table 3: NH symmetry classes with intrinsic point gap topology in 3D and the corresponding Hermitian classes of the EHH.

| dim | class $H$ | class $\bar{H}$ ($E_0 = 0$) | class $\bar{H}$ ($E_0 \neq 0$) | App. |
|---|---|---|---|---|
| 3 | AI$^\dagger$ | CI | CI | B.5.1 |
| 3 | AII$^\dagger$ | DIII | DIII | B.5.2 |
| 3 | A | AIII | AIII | B.5.3 |
| 3 | A$^S$ | AIII $\oplus$ AIII | AIII | B.5.4 |
| 3 | D | DIII | AIII | B.5.5 |
| 3 | D$^{S_+}$ | AIII | CI | B.5.6 |
| 3 | D$^{S_-}$ | DIII $\oplus$ DIII | DIII | B.5.7 |
| 3 | DIII$^{S_{+-}}$ | AII $\oplus$ AII | DIII | B.5.8 |
| 3 | AII$^{S_+}$ | CII $\oplus$ CII | AIII | B.5.9 |
| 3 | C | CI | AIII | B.5.10 |
| 3 | C$^{S_+}$ | AIII | DIII | B.5.11 |
| 3 | C$^{S_-}$ | CI $\oplus$ CI | CI | B.5.12 |

## B.3  3D systems, symmetry classes of $\bar{\mathcal{H}}(k)$

In 3D, the EHH $\bar{\mathcal{H}}(k)$ for nontrivial point-gapped NH models resides in Hermitian classes AII, AIII, DIII, CI or CII (see Tab. 3). In the following, we investigate whether a flux induced mass domain wall leads to zero energy modes of $\bar{\mathcal{H}}(k)$.

### B.3.1  Class AII

In App. B.2.2 we derived the absence of zero energy modes at flux defects in 2D Hermitian class AII. In 3D, the edge Hamiltonian in the nontrivial phase is given as

$$\bar{h}(k_1, k_2) = k_1 \sigma_x + k_2 \sigma_y, \tag{B.27}$$

where $k_1$ is the momentum along the edge, $k_2$ the momentum along the flux tube, we obtain the 1D Dirac Hamiltonian as

$$\bar{H}(k_1, k_2) = \tau_z \otimes \bar{h}(k_1, k_2) + m x \tau_x \sigma_0 + \delta O, \tag{B.28}$$

with the Pauli matrices $\tau_\mu, \sigma_\mu$ ($\mu = 0, x, y, z$). For $k_2 = \delta = 0$ we still find the two normalizable zero energy solutions given by (B.11). In order to derive the effective Hamiltonian along the flux tube, we rely on degenerate first-order perturbation theory in $k_2$ and $\delta$. Using $\delta O = \sum_{i=x,y,z} \delta_i \tau_y \sigma_i + \delta_4 \tau_0 \sigma_0 + \delta_5 \tau_z \sigma_0$ as the symmetry preserving choices under TRS $\bar{U}_{\mathcal{T}} = \tau_0 \sigma_y$, we obtain

$$\left[ \bar{H}_{\text{flux}}(k_2) \right]_{mn} = \langle \psi_m | \bar{H}_x(k_2) | \psi_n \rangle = \left[ k_2 \sigma_y + (\delta_4 - \delta_1) \sigma_0 \right]_{mn}. \tag{B.29}$$

Since the identity matrix constitutes just a trivial spectral shift, the Hamiltonian along the flux tube is still gapless. Consequently, flux tubes in Hermitian class AII ($d=3$) form a helical metal, with zero modes of $\bar{\mathcal{H}}(k)$ appearing at $k_2 = 0$.

### B.3.2  Class AIII

Hermitian class AIII contains a chiral (sublattice) symmetry, such that the edge Hamiltonian for the nontrivial phase appears in (B.2) as

$$\bar{H}(k_1, k_2) = \tau_z \otimes (k_1 \sigma_x + k_2 \sigma_y) + M + \delta O, \tag{B.30}$$

where $k_1$ is the momentum along the edge, $k_2$ the momentum along the flux tube, $M$ the mass matrix that couples the edge states to yield a gapped bulk, $\tau_\mu, \sigma_\mu$ Pauli matrices ($\mu = 0, x, y, z$) and $\delta$ multiplies a collection of symmetry-allowed perturbation terms $O$. The insertion of a $\pi-$flux tube forms a mass domain wall, causing a sign change in $M$. Note however that Hermitian class AIII does not contain a symmetry fixing the flux to zero or $\pi$. In order to derive the presence of topological zero energy states, we consider OBC along the edge ($x-$direction). Without further symmetries, there exists an ambiguity in the suitable mass domain wall, which can equivalently be realized as $\tau_y\sigma_0$ or $\tau_x\sigma_0$. Therefore there exists no bound state without additional (crystalline) symmetries. Adding for instance inversion symmetry $\bar{\mathcal{I}}$, realized here as $\bar{\mathcal{I}} = \tau_x\sigma_0$, removes the ambiguity by rendering $\tau_y\sigma_0$ an incompatible choice. However, our current single mass domain wall inherently breaks inversion symmetry. We therefore have to introduce a second flux tube, representing the opposite mass domain wall. A corresponding 2D-Dirac theory is discussed in Ref. [14], showing that each flux tube localizes one zero energy mode under OBC.

### B.3.3 Class DIII

Since the DIII 3D topological insulator can be understood as a pumping cycle of the corresponding 2D system along a third momentum direction, the flux response follows directly from the respective case in 2D. In App. B.2.4 we derived the presence of two zero modes under $\pi$-flux insertion. Accounting for the additional dimension by using

$$\bar{h}(k_1, k_2) = k_1\sigma_x + k_2\sigma_y, \tag{B.31}$$

where $k_1$ is the momentum along the edge, $k_2$ the momentum along the flux tube, we obtain the 2D Dirac Hamiltonian as

$$\bar{H}(k_1, k_2) = \tau_z \otimes \bar{h}(k_1, k_2) + mx\tau_x\sigma_0 + \delta O, \tag{B.32}$$

with the Pauli matrices $\tau_\mu, \sigma_\mu$ ($\mu = 0, x, y, z$). For $k_2 = \delta = 0$ we still find the two normalizable zero energy solutions given by (B.20). In order to derive the effective Hamiltonian along the flux tube, we rely on degenerate first-order perturbation theory in $k_2$ and $\delta$. Using again $O = \tau_y\sigma_z$ as the only symmetry preserving choice under TRS $\bar{U}_\mathcal{T} = \tau_0\sigma_y$ and PHS $\bar{U}_\mathcal{P} = \tau_z\sigma_x$, we obtain

$$\left[\bar{H}_{\text{flux}}(k_2)\right]_{mn} = \langle\psi_m|\bar{H}_x(k_2)|\psi_n\rangle = \left[k_2\sigma_y\right]_{mn}. \tag{B.33}$$

Consequently, flux tubes in Hermitian class DIII ($d$=3) form a gapless helical metal, with zero modes of $\bar{\mathcal{H}}(k)$ appearing at $k_2 = 0$.

### B.3.4 Class CI

Hamiltonians in Hermitian symmetry class CI possess TRS, chiral and PHS, where only the latter squares to minus one. The corresponding surface Hamiltonian along the edge $\bar{h}(k_1, k_2)$ is gapless in the nontrivial phase and assumes the form

$$\bar{h}(k_1, k_2) = k_1\rho_y\sigma_x + k_2\rho_0\sigma_y, \tag{B.34}$$

where $k_1$ is the momentum along the edge, $k_2$ the momentum along the flux tube, $\rho_\mu, \tau_\mu, \sigma_\mu$ Pauli matrices ($\mu = 0, x, y, z$) and allowed mass terms only constitute trivial shifts of the location of the zero energy mode in momentum space. Using $\bar{U}_\mathcal{T} = \tau_0\rho_0\sigma_0$, $\bar{U}_\mathcal{P} = \tau_z\rho_y\sigma_z$, $\bar{U}_\mathcal{C} = \tau_z\rho_y\sigma_z$ the 1D Dirac Hamiltonian can then be written as

$$\bar{H}(k) = \tau_z \otimes \bar{h}(k) + mx\tau_x\rho_0\sigma_0 + \delta O, \tag{B.35}$$

where $mx$ implements the flux tube induced domain wall in the mass term at $x = 0$ and $\delta$ multiplies a collection of symmetry allowed perturbation terms $O$. In order to derive the presence of topological zero energy states, we consider OBC along the edge ($x$−direction) and solve for boundary-localized zero-energy states determined by the Hamiltonian

$$\bar{H}_x = -i\tau_z\rho_y\sigma_x\frac{\partial}{\partial x} + k_2\tau_z\rho_0\sigma_y + mx\tau_x\rho_0\sigma_0 + \delta O\,. \tag{B.36}$$

For $k_2 = \delta = 0$ we can find four normalizable zero energy solutions $|\psi\rangle$, $\bar{H}_x|\psi\rangle = 0$, given by

$$|\psi_1\rangle = \frac{1}{N}e^{-\frac{1}{2}mx^2}\begin{pmatrix}0\\1\\0\\0\\0\\0\\0\\1\end{pmatrix}, \quad |\psi_2\rangle = \frac{1}{N}e^{-\frac{1}{2}mx^2}\begin{pmatrix}-1\\0\\0\\0\\0\\0\\1\\0\end{pmatrix},$$

$$|\psi_3\rangle = \frac{1}{N}e^{-\frac{1}{2}mx^2}\begin{pmatrix}0\\0\\0\\1\\0\\1\\0\\0\end{pmatrix}, \quad |\psi_4\rangle = \frac{1}{N}e^{-\frac{1}{2}mx^2}\begin{pmatrix}0\\0\\-1\\0\\1\\0\\0\\0\end{pmatrix}. \tag{B.37}$$

In order to derive the effective Hamiltonian along the flux tube, we rely on degenerate first-order perturbation theory in $k_2$ and $\delta$. Using $O = \tau_y\rho_y\sigma_0$ as one symmetry preserving choice, we obtain

$$\begin{aligned}\left[\bar{H}_{\text{flux}}(k_2)\right]_{mn} &= \langle\psi_m|\bar{H}_x(k_2)|\psi_n\rangle\\ &= \left[k_2\rho_0\sigma_y - \delta\rho_z\sigma_z\right]_{mn}\,.\end{aligned} \tag{B.38}$$

Consequently, flux tubes in Hermitian class CI ($d$=3) are gapped, and also host no zero energy end states under OBC.

### B.3.5 Class CII

In Hermitian symmetry class CII, the surface Hamiltonian along the edge $\bar{h}(k_1, k_2)$ is gapless for the nontrivial phase and assumes the form

$$\bar{h}(k_1, k_2) = k_1\rho_0\sigma_x + k_2\rho_0\sigma_y\,, \tag{B.39}$$

where $k_1$ is the momentum along the edge, $k_2$ the momentum along the flux tube, $\rho_\mu, \sigma_\mu$ are Pauli matrices ($\mu = 0, x, y, z$) and mass terms are forbidden by the presence of TRS, PHS and chiral symmetry. Choosing chiral symmetry as $\bar{U}_{\mathcal{C}} = \tau_z\rho_z\sigma_z$, the 1D Dirac Hamiltonian can be written as

$$\bar{H}(k_1, k_2) = \tau_z \otimes (k_1\rho_0\sigma_x + k_2\rho_0\sigma_y) + M + \delta O\,, \tag{B.40}$$

where $M$ implements the $\pi$−flux tube induced domain wall in the mass term at $x = 0$, $\tau_\mu$ are Pauli matrices ($\mu = 0, x, y, z$) and $\delta$ multiplies a collection of symmetry allowed perturbation terms $O$. Note however that Hermitian class AIII does not contain a symmetry fixing the flux to zero or $\pi$. Requiring TRS as $\bar{U}_{\mathcal{T}} = i\tau_0\rho_x\sigma_y$ and PHS as $\bar{U}_{\mathcal{P}} = i\tau_z\rho_y\sigma_x$, one can find

two possible mass terms $M$, $\tau_x\rho_0\sigma_0$ or $\tau_y\rho_z\sigma_0$. Due to this ambiguity there exists no bound state without additional (crystalline) symmetries. Adding for instance inversion symmetry $\bar{\mathcal{I}}$, realized here as $\bar{\mathcal{I}} = \tau_x\rho_0\sigma_0$, removes the ambiguity by rendering $\tau_y\rho_z\sigma_0$ an incompatible choice. We therefore choose the mass domain wall to be realized as $M = mx\tau_x\rho_0\sigma_0$. However, the single mass domain wall again breaks inversion symmetry. We therefore have to introduce a second flux tube, representing the opposite mass domain wall. Using again the results of the 2D-Dirac theory discussed in Ref. [14], one obtains two zero energy modes per flux tube, localized at the flux tube ends.

## B.4  NH flux response in 2D

Zero modes of the EHH and NH point gap states are in one-to-one correspondence. Using the Hermitian flux Dirac theory, we here derive the flux response and the corresponding topological properties of the SIBC spectrum of all intrinsically nontrivial 2D point-gapped NH symmetry classes, summarized in Tab. 2.

### B.4.1  Class AII$^\dagger$

NH class AII$^\dagger$ has a TRS$^\dagger$ squaring to minus one, which quantizes the flux $\phi = 0, \pi$. As outlined in A.2, the corresponding EHH introduces a chiral symmetry $\bar{\Sigma}_{\mathcal{C}}$, which combines with TRS to form a PHS with $\bar{U}_{\mathcal{P}}\bar{U}_{\mathcal{P}}^* = -\bar{U}_{\mathcal{T}}\bar{U}_{\mathcal{T}}^* = +1$. Consequently, the EHH is in Hermitian class DIII, which is $\mathbb{Z}_2$ classified for both 1D and 2D [79].

The flux response of the EHH in Hermitian class DIII shows two zero energy modes for each flux core (see App. B.2.4). As the construction of the EHH can be repeated for every complex energy inside the point gap, the point gap of a corresponding NH system fills with flux core localized modes. Consequently, models in NH class AII$^\dagger$ show a flux skin effect for $\phi = \pi$. A periodic system contains two flux cores $\phi = \pm\pi$, each localizing an extensive number of modes. This number scales with the extension of the system along the direction containing the two defects, denoted by $L_\perp$ (see App. C.1).

### B.4.2  Class DIII$^\dagger$

The NH class DIII$^\dagger$ possesses TRS, PHS and chiral symmetry with $(U_{\mathcal{T}}U_{\mathcal{T}}^*, U_{\mathcal{P}}U_{\mathcal{P}}^*) = (-1, 1)$, where $U_{\mathcal{C}} = U_{\mathcal{T}}U_{\mathcal{P}}$. As outlined in App. A.2, the corresponding EHH at $E_0 = 0$ obtains TRS $\bar{U}_{\mathcal{T}}$, PHS $\bar{U}_{\mathcal{P}}$ and two chiral symmetries $\{\bar{U}_{\mathcal{C}}, \bar{\Sigma}_{\mathcal{C}}\} = 0$. Both chiral symmetries can be combined to a unitary symmetry $\bar{U} = \bar{U}_{\mathcal{C}}\bar{\Sigma}_{\mathcal{C}}$ with $\bar{U}^2 = -1$.

Since $\bar{U}$ has an imaginary spectrum and $[\bar{U}_{\mathcal{T}}, \bar{U}] = 0$, the eigenspaces of $\bar{U}$ are exchanged by anti-unitary TRS. Since $\{\bar{U}_{\mathcal{P}}, \bar{U}\} = 0$, we retain anti-unitary PHS in each eigenspace, corresponding to Hermitian class D which has a $\mathbb{Z}$ classification in 2D [79]. After modding out line-gap phases, this is reduced to a $\mathbb{Z}_2$ classification [39]. Since 1D Hermitian class D is also $\mathbb{Z}_2$ classified, we expect that a $\pi$-flux induces a single bound state per $\bar{U}$ eigensector (see App. B.2.3). Hence, the full Hermitian spectrum hosts a zero-energy Kramers pair per flux.

Away from $E_0 = 0$, we retain only $\bar{U}_{\mathcal{T}}\bar{U}_{\mathcal{T}}^* = -1$ where $\bar{U}_{\mathcal{T}}$ and $\bar{\Sigma}_{\mathcal{C}}$ anti-commute, resulting in Hermitian class DIII. Due to TRS and chiral symmetry, the flux-bound Kramers pair cannot move away from zero energy. Hence the entire point gap of a corresponding NH system fills with flux core localized modes. Consequently, models in NH class DIII$^\dagger$ quantize the flux to $\phi = 0, \pi$ and show a flux skin effect for $\phi = \pi$.

### B.4.3  Class AIII$^{S-}$

Models in NH class AIII$^{S-}$ possess chiral symmetry $U_{\mathcal{C}}$ and sublattice symmetry $\mathcal{S}$, with $\{U_{\mathcal{C}}, \mathcal{S}\} = 0$. As outlined in A.2, the corresponding EHH at $E_0 = 0$ obtains three chiral sym-

metries $\bar{U}_{\mathcal{C}}, \bar{\Sigma}_{\mathcal{C}}, \bar{\mathcal{S}}$, which can be combined to two unitary symmetries $\bar{U}_1 = \bar{U}_{\mathcal{C}}\bar{\mathcal{S}}$ with $\bar{U}_1^2 = -1$ and $\bar{U}_2 = \bar{\mathcal{S}}\bar{\Sigma}_{\mathcal{C}}$ with $\bar{U}_2^2 = +1$. Due to the anti-commutation of $\bar{U}_1$ with the chiral symmetries, the eigenspaces of $\bar{U}_1$ are exchanged by $\bar{U}_{\mathcal{C}}, \bar{\Sigma}_{\mathcal{C}}, \bar{\mathcal{S}}$. Since $[\bar{U}_1, \bar{U}_2] = 0$, we retain $\bar{U}_2$ in each subspace. However, each eigenspace of $\bar{U}_2$ has no symmetry left, corresponding to Hermitian class A⊕A.

In App. B.2.1, we derived the flux response of EHHs in Hermitian class A, which did not yield protected zero energy modes pinned to the flux defects. Consequently, NH systems in class AIII$^{S-}$ do not show flux induced states within the NH point gap. Therefore, the flux response for this NH symmetry class is trivial.

### B.4.4  Class BDI$^{S+-}$

The NH class BDI$^{S+-}$ possesses TRS, PHS and chiral symmetry with $(U_{\mathcal{T}}U_{\mathcal{T}}^*, U_{\mathcal{P}}U_{\mathcal{P}}^*) = (1,1)$, where $U_{\mathcal{C}} = U_{\mathcal{T}}U_{\mathcal{P}}$, as well as sublattice symmetry $\mathcal{S}$. As outlined in A.2, the corresponding EHH at $E_0 = 0$ obtains TRS $\bar{U}_{\mathcal{T}}$, PHS $\bar{U}_{\mathcal{P}}$ and three chiral symmetries $\bar{U}_{\mathcal{C}}, \bar{\Sigma}_{\mathcal{C}}, \bar{\mathcal{S}}$. The chiral symmetries can be combined to two commuting unitary symmetries $\bar{U}_1 = \bar{U}_{\mathcal{C}}\bar{\Sigma}_{\mathcal{C}}$ with $\bar{U}_1^2 = -1$ and $\bar{U}_2 = \bar{\mathcal{S}}\bar{\Sigma}_{\mathcal{C}}$ with $\bar{U}_2^2 = +1$.

Since $\bar{U}_1$ has an imaginary spectrum and

$$[\bar{U}_{\mathcal{T}}, \bar{U}_1] = \{\bar{U}_{\mathcal{P}}, \bar{U}_1\} = \{\bar{U}_{\mathcal{C}}, \bar{U}_1\} = \{\bar{\Sigma}_{\mathcal{C}}, \bar{U}_1\} = \{\bar{\mathcal{S}}, \bar{U}_1\},$$

the eigenspaces of $\bar{U}_1$ are not independent and individually enjoy $\bar{U}_{\mathcal{P}}$ and $\bar{U}_2$ symmetry. Moreover, we have $[\bar{U}_{\mathcal{T}}, \bar{U}_2] = [\bar{U}_{\mathcal{P}}, \bar{U}_2] = [\bar{\mathcal{S}}, \bar{U}_2] = [\bar{U}_{\mathcal{C}}, \bar{U}_2] = [\bar{\Sigma}_{\mathcal{C}}, \bar{U}_2] = 0$, so that the $\bar{U}_2$ eigenspaces are independent and individually preserve $\bar{U}_{\mathcal{P}}$ symmetry. They therefore lie in Hermitian class D and yield a $\mathbb{Z} \oplus \mathbb{Z}$ classification in 2D [79] that is reduced to $\mathbb{Z}_2$ by line gap phases [39]. The nontrivial point gap phase is the one where only one $\bar{U}_2$ subspace is nontrivial, corresponding to a single $p + \mathrm{i}p$ superconductor per $\bar{U}_1$ eigenspace (see App. B.2.3). The full Hermitian spectrum will therefore contain two exact zeromodes in presence of a $\pi$-flux.

Away from $E_0 = 0$, we can only form the TRS $\bar{U}_{\mathcal{T}}' = \bar{\mathcal{S}}\bar{U}_{\mathcal{P}}$ with $\bar{U}_{\mathcal{T}}'\bar{U}_{\mathcal{T}}'^* = -1$ and PHS $\bar{U}_{\mathcal{P}}' = \bar{\Sigma}_{\mathcal{C}}\bar{U}_{\mathcal{T}}'$ with $\bar{U}_{\mathcal{P}}'\bar{U}_{\mathcal{P}}'^* = +1$ (see App. A.2). This results in Hermitian class DIII, such that due to TRS and chiral symmetry, the flux-bound Kramers pair cannot move away from zero energy. Hence the entire point gap of a corresponding NH system fills with flux core localized modes. Consequently, models in NH class BDI$^{S+-}$ quantize the flux to $\phi = 0, \pi$ and show a flux skin effect for $\phi = \pi$.

### B.4.5  Class D$^{S-}$

Models in NH class D$^{S-}$ possess PHS $U_{\mathcal{P}}$ ($U_{\mathcal{P}}U_{\mathcal{P}}^* = +1$) and sublattice symmetry $\mathcal{S}$, with $\{U_{\mathcal{P}}, \mathcal{S}\} = 0$. As outlined in App. A.2, the corresponding EHH introduces a chiral symmetry $\bar{\Sigma}_{\mathcal{C}}$, which combines with sublattice symmetry to form a unitary symmetry $\bar{U} = \bar{\mathcal{S}}\bar{\Sigma}_{\mathcal{C}}$ with $\bar{U}^2 = +1$. The unitary $\bar{U}$ satisifies $[\bar{U}_{\mathcal{P}}, \bar{U}] = [\bar{\mathcal{S}}, \bar{U}] = [\bar{\Sigma}_{\mathcal{C}}, \bar{U}] = 0$. We may furthermore define a TRS $\bar{U}_{\mathcal{T}} = \bar{U}_{\mathcal{P}}\bar{\Sigma}_{\mathcal{C}}$ which satisfies $\bar{U}_{\mathcal{T}}\bar{U}_{\mathcal{T}}^* = -1$. Hence we obtain Hermitian class DIII in each $\bar{U}$ subspace, giving a $\mathbb{Z}_2 \oplus \mathbb{Z}_2$ classification in 2D [79]. By modding out line-gap phases, this is reduced to a $\mathbb{Z}_2$ classification where the nontrivial element corresponds to having only a single $\bar{U}$ subspace being nontrivial [39]. Hence the full Hermitian spectrum hosts a single zero-energy Kramers pair under $\pi$-flux insertion (see App. B.2.4).

Away from $E_0 = 0$, we can only form the TRS $\bar{U}_{\mathcal{T}}' = \bar{\mathcal{S}}\bar{U}_{\mathcal{P}}$ with $\bar{U}_{\mathcal{T}}'\bar{U}_{\mathcal{T}}'^* = -1$ and PHS $\bar{U}_{\mathcal{P}}' = \bar{\Sigma}_{\mathcal{C}}\bar{U}_{\mathcal{T}}'$ with $\bar{U}_{\mathcal{P}}'\bar{U}_{\mathcal{P}}'^* = +1$ (see Sec. A.2). This results in Hermitian class DIII, such that the flux-bound Kramers pair cannot move away from zero energy. Hence the entire point gap of a corresponding NH system fills with flux core localized modes. Consequently, models in NH class D$^{S-}$ quantize the flux to $\phi = 0, \pi$ and show a flux skin effect for $\phi = \pi$.

### B.4.6 Class DIII$^{S+-}$

The NH class DIII$^{S+-}$ possesses TRS, PHS and chiral symmetry with $(U_{\mathcal{T}}U_{\mathcal{T}}^*, U_{\mathcal{P}}U_{\mathcal{P}}^*) = (-1, 1)$, where $U_{\mathcal{C}} = U_{\mathcal{T}}U_{\mathcal{P}}$, as well as sublattice symmetry $\mathcal{S}$. As outlined in App. A.2, the corresponding EHH at $E_0 = 0$ obtains TRS $\bar{U}_{\mathcal{T}}$, PHS $\bar{U}_{\mathcal{P}}$ and three chiral symmetries $\bar{U}_{\mathcal{C}}, \bar{\Sigma}_{\mathcal{C}}, \bar{\mathcal{S}}$. The chiral symmetries can be combined to two commuting unitary symmetries $\bar{U}_1 = \bar{U}_{\mathcal{C}}\bar{\Sigma}_{\mathcal{C}}$ with $\bar{U}_1^2 = -1$ and $\bar{U}_2 = \bar{\mathcal{S}}\bar{\Sigma}_{\mathcal{C}}$ with $\bar{U}_2^2 = +1$.

Since $\bar{U}_1$ has an imaginary spectrum and

$$\{\bar{U}_{\mathcal{T}}, \bar{U}_1\} = [\bar{U}_{\mathcal{P}}, \bar{U}_1] = \{\bar{U}_{\mathcal{C}}, \bar{U}_1\} = \{\bar{\Sigma}_{\mathcal{C}}, \bar{U}_1\} = \{\bar{\mathcal{S}}, \bar{U}_1\},$$

the eigenspaces of $\bar{U}_1$ are not independent and individually enjoy $\bar{U}_{\mathcal{T}}$ and $\bar{U}_2$ symmetry. Moreover, we have $[\bar{U}_{\mathcal{T}}, \bar{U}_2] = [\bar{U}_{\mathcal{P}}, \bar{U}_2] = [\bar{\mathcal{S}}, \bar{U}_2] = [\bar{U}_{\mathcal{C}}, \bar{U}_2] = [\bar{\Sigma}_{\mathcal{C}}, \bar{U}_2] = 0$, so that the $\bar{U}_2$ eigenspaces are independent and individually preserve $\bar{U}_{\mathcal{T}}$ symmetry. They therefore lie in Hermitian class AII and yield a $\mathbb{Z}_2 \oplus \mathbb{Z}_2$ classification in 2D [79] that is reduced to $\mathbb{Z}_2$ by line-gap phases [39]. The nontrivial element corresponds to having only a single $\bar{U}_2$ subspace nontrivial. In App. B.2.2, we derived the flux response of EHHs in Hermitian class AII, which did not yield protected zero energy modes pinned to the flux defects. Consequently, systems in NH class DIII$^{S+-}$ do not show flux induced states within the NH point gap. Therefore, the flux response for this NH symmetry class is trivial.

### B.4.7 Class CII$^{S-+}$

The NH class CII$^{S-+}$ possesses TRS, PHS and chiral symmetry with $(U_{\mathcal{T}}U_{\mathcal{T}}^*, U_{\mathcal{P}}U_{\mathcal{P}}^*) = (-1, -1)$, where $U_{\mathcal{C}} = U_{\mathcal{T}}U_{\mathcal{P}}$, as well as sublattice symmetry $\mathcal{S}$. As outlined in App. A.2, the corresponding EHH at $E_0 = 0$ obtains TRS $\bar{U}_{\mathcal{T}}$, PHS $\bar{U}_{\mathcal{P}}$ and three chiral symmetries $\bar{U}_{\mathcal{C}}, \bar{\Sigma}_{\mathcal{C}}, \bar{\mathcal{S}}$. The chiral symmetries can be combined to two commuting unitary symmetries $\bar{U}_1 = \bar{U}_{\mathcal{C}}\bar{\Sigma}_{\mathcal{C}}$ with $\bar{U}_1^2 = -1$ and $\bar{U}_2 = \bar{\mathcal{S}}\bar{\Sigma}_{\mathcal{C}}$ with $\bar{U}_2^2 = +1$.

Since $\bar{U}_1$ has an imaginary spectrum and

$$[\bar{U}_{\mathcal{T}}, \bar{U}_1] = \{\bar{U}_{\mathcal{P}}, \bar{U}_1\} = \{\bar{U}_{\mathcal{C}}, \bar{U}_1\} = \{\bar{\Sigma}_{\mathcal{C}}, \bar{U}_1\} = \{\bar{\mathcal{S}}, \bar{U}_1\},$$

the eigenspaces of $\bar{U}_1$ are not independent and individually enjoy $\bar{U}_{\mathcal{P}}$ and $\bar{U}_2$ symmetry. Moreover, we have $\{\bar{U}_{\mathcal{T}}, \bar{U}_2\} = \{\bar{U}_{\mathcal{P}}, \bar{U}_2\} = [\bar{\mathcal{S}}, \bar{U}_2] = [\bar{U}_{\mathcal{C}}, \bar{U}_2] = [\bar{\Sigma}_{\mathcal{C}}, \bar{U}_2] = 0$, so that the $\bar{U}_2$ eigenspaces are exchanged by $\bar{U}_{\mathcal{P}}$ symmetry. This leaves us with Hermitian class A, which has a $\mathbb{Z}$ classification in 2D [79] that is reduced to $\mathbb{Z}_2$ by line-gap phases [39]. In App. B.2.1, we derived the flux response of EHHs in Hermitian class A, which did not yield protected zero energy modes pinned to the flux defects. Consequently, systems in NH class CII$^{S-+}$ do not show flux induced states within the NH point gap. Therefore, the flux response for this NH symmetry class is trivial.

### B.4.8 Class CII$^{S+-}$

The NH class CII$^{S+-}$ possesses TRS, PHS and chiral symmetry with $(U_{\mathcal{T}}U_{\mathcal{T}}^*, U_{\mathcal{P}}U_{\mathcal{P}}^*) = (-1, -1)$, where $U_{\mathcal{C}} = U_{\mathcal{T}}U_{\mathcal{P}}$, as well as sublattice symmetry $\mathcal{S}$. As outlined in App. A.2, the corresponding EHH at $E_0 = 0$ obtains TRS $\bar{U}_{\mathcal{T}}$, PHS $\bar{U}_{\mathcal{P}}$ and three chiral symmetries $\bar{U}_{\mathcal{C}}, \bar{\Sigma}_{\mathcal{C}}, \bar{\mathcal{S}}$. The chiral symmetries can be combined to two commuting unitary symmetries $\bar{U}_1 = \bar{U}_{\mathcal{C}}\bar{\Sigma}_{\mathcal{C}}$ with $\bar{U}_1^2 = -1$ and $\bar{U}_2 = \bar{\mathcal{S}}\bar{\Sigma}_{\mathcal{C}}$ with $\bar{U}_2^2 = +1$.

Since $\bar{U}_1$ has an imaginary spectrum and

$$[\bar{U}_{\mathcal{T}}, \bar{U}_1] = \{\bar{U}_{\mathcal{P}}, \bar{U}_1\} = \{\bar{U}_{\mathcal{C}}, \bar{U}_1\} = \{\bar{\Sigma}_{\mathcal{C}}, \bar{U}_1\} = \{\bar{\mathcal{S}}, \bar{U}_1\},$$

the eigenspaces of $\bar{U}_1$ are not independent and individually enjoy $\bar{U}_{\mathcal{P}}$ and $\bar{U}_2$ symmetry. Moreover, we have $[\bar{U}_{\mathcal{T}}, \bar{U}_2] = [\bar{U}_{\mathcal{P}}, \bar{U}_2] = [\bar{\mathcal{S}}, \bar{U}_2] = [\bar{U}_{\mathcal{C}}, \bar{U}_2] = [\bar{\Sigma}_{\mathcal{C}}, \bar{U}_2] = 0$, so that the $\bar{U}_2$

eigenspaces are independent and individually preserve $\bar{U}_{\mathcal{P}}$ symmetry. We therefore obtain Hermitian class C, giving a $\mathbb{Z} \oplus \mathbb{Z}$ classification in 2D [79]. This is reduced to $\mathbb{Z}_2$ under the addition of line-gap phases, and the nontrivial element corresponds to only having one $\bar{U}_2$ eigenspace nontrivial [39]. Since the 1D classification of Hermitian class C is trivial and App. B.2.5 showed no protected zero energy modes, we do not expect a stable flux response. Therefore, the flux response for systems in NH class $\text{CII}^{S+-}$ is trivial.

### B.4.9  Class $\text{CI}^{S-+}$

The NH class $\text{CI}^{S-+}$ possesses TRS, PHS and chiral symmetry with $(U_{\mathcal{T}} U_{\mathcal{T}}^*, U_{\mathcal{P}} U_{\mathcal{P}}^*) = (1, -1)$, where $U_{\mathcal{C}} = U_{\mathcal{T}} U_{\mathcal{P}}$, as well as sublattice symmetry $\mathcal{S}$. As outlined in App. A.2, the corresponding EHH at $E_0 = 0$ obtains TRS $\bar{U}_{\mathcal{T}}$, PHS $\bar{U}_{\mathcal{P}}$ and three chiral symmetries $\bar{U}_{\mathcal{C}}, \bar{\Sigma}_{\mathcal{C}}, \bar{\mathcal{S}}$. The chiral symmetries can be combined to two commuting unitary symmetries $\bar{U}_1 = \bar{U}_{\mathcal{C}} \bar{\Sigma}_{\mathcal{C}}$ with $\bar{U}_1^2 = -1$ and $\bar{U}_2 = \bar{\mathcal{S}} \bar{\Sigma}_{\mathcal{C}}$ with $\bar{U}_2^2 = +1$.

Since $\bar{U}_1$ has an imaginary spectrum and

$$\{\bar{U}_{\mathcal{T}}, \bar{U}_1\} = [\bar{U}_{\mathcal{P}}, \bar{U}_1] = \{\bar{U}_{\mathcal{C}}, \bar{U}_1\} = \{\bar{\Sigma}_{\mathcal{C}}, \bar{U}_1\} = \{\bar{\mathcal{S}}, \bar{U}_1\},$$

the eigenspaces of $\bar{U}_1$ are not independent and individually enjoy $\bar{U}_{\mathcal{T}}$ and $\bar{U}_2$ symmetry. Moreover, we have $\{\bar{U}_{\mathcal{T}}, \bar{U}_2\} = \{\bar{U}_{\mathcal{P}}, \bar{U}_2\} = [\bar{\mathcal{S}}, \bar{U}_2] = [\bar{U}_{\mathcal{C}}, \bar{U}_2] = [\bar{\Sigma}_{\mathcal{C}}, \bar{U}_2] = 0$, so that the $\bar{U}_2$ eigenspaces are exchanged by $\bar{U}_{\mathcal{T}}$ symmetry. This leaves us with Hermitian class A, which has a $\mathbb{Z}$ classification in 2D [79] that is reduced to $\mathbb{Z}_2$ by line-gap phases [39]. In App. B.2.1, we derived the flux response of EHHs in Hermitian class A, which did not yield protected zero energy modes pinned to the flux defects. Consequently, systems in NH class $\text{CI}^{S-+}$ do not show flux induced states within the NH point gap. Therefore, the flux response for this NH symmetry class is trivial.

## B.5  NH flux response in 3D

Similar to the case of 2D, we rely on the Hermitian flux Dirac theory to derive the flux response of all nontrivial 3D point-gapped NH symmetry classes, summarized in Tab. 3.

### B.5.1  Class $\text{AI}^\dagger$

NH class $\text{AI}^\dagger$ has a $\text{TRS}^\dagger$ squaring to plus one. As outlined in App. A.2, the corresponding EHH introduces a chiral symmetry $\bar{\Sigma}_{\mathcal{C}}$, which combines with TRS to form a PHS with $\bar{U}_{\mathcal{P}} \bar{U}_{\mathcal{P}}^* = -\bar{U}_{\mathcal{T}} \bar{U}_{\mathcal{T}}^* = -1$. Consequently, the EHH for all energies inside the point gap $E_0$ is in Hermitian class CI.

The flux response of the EHH in Hermitian class CI shows no protected zero energy modes (see App. B.3.4). Consequently, systems in NH class $\text{AI}^\dagger$ do not show flux induced states within the NH point gap. Therefore, the flux response for this NH symmetry class is trivial.

### B.5.2  Class $\text{AII}^\dagger$

As for the case of 2D, NH class $\text{AII}^\dagger$ has a $\text{TRS}^\dagger$ squaring to minus one, which quantizes the flux $\phi = 0, \pi$. As outlined in App. A.2, the corresponding EHH introduces a chiral symmetry $\bar{\Sigma}_{\mathcal{C}}$, which combines with TRS to form a PHS with $\bar{U}_{\mathcal{P}} \bar{U}_{\mathcal{P}}^* = -\bar{U}_{\mathcal{T}} \bar{U}_{\mathcal{T}}^* = +1$. Consequently, the EHH is in Hermitian class DIII, which is $\mathbb{Z}$ classified for 3D and hosts a $\mathbb{Z}_2$ index for 2D [79].

The flux response of the EHH in Hermitian class DIII forms a gapless helical metal along the flux tube (see App. B.3.3). As the construction of the EHH can be repeated for every complex energy inside the point gap, the point gap of a corresponding NH system fills with an extensive number of states at a single momentum $k_\parallel$. This is the defining signature of the NH

flux spectral jump, discussed in Sec. 4. Therefore, systems in NH class AII$^\dagger$ show a NH flux spectral jump when introducing a flux defect. Note that due to the $\mathbb{Z}_2$-classification in 2D, flux defects only probe the $\mathbb{Z}_2$ nature of the 3D $\mathbb{Z}$ invariant in NH class AII$^\dagger$.

### B.5.3 Class A+$\mathcal{I}^\dagger$

As NH class A does not contain any symmetries, the corresponding EHH only introduces a chiral symmetry $\bar\Sigma_{\mathcal{C}}$ (see App. A.2). Consequently, the EHH is in Hermitian class AIII, which is $\mathbb{Z}$ classified for 3D and 1D while being trivial in 2D [79].

Without additional crystalline symmetries, the zero energy modes arising in the flux Dirac theory of systems in Hermitian class AIII are not protected (see App. B.3.2). This indicates the absence of a NH flux spectral jump, but allows for a different, higher order phase. Adding a crystalline symmetry $\mathcal{I}^\dagger$ serves two purposes: first, it should fix the flux to be either zero or $\pi$, to allow for a topological flux response. Second, it should protect zero energy states from being gapped by perturbations. A crystalline symmetry fixing the flux requires the presence of at least two defects, resulting in a 2D Hermitian flux Dirac theory. The results derived in App. B.3.2 show the presence of zero energy states under OBC along the flux tubes, which can be distinguished from the surface signature for $\phi = 0$. Consequently, these zero modes appear extensively in the point gap of the NH model. This constitutes a higher order skin effect, with skin modes appearing at the ends of the flux tubes in a finite geometry. The precise localization of skin modes is then determined by the present pseudo inversion symmetry $\mathcal{I}^\dagger$. This is the defining signature of the NH *higher-order flux skin effect*, discussed in Sec. 5.

### B.5.4 Class A$^S$ + $\mathcal{I}^{\dagger,S_-}$

NH class A$^S$ possesses a sublattice symmetry $\mathcal{S}$, which combines with the chiral symmetry $\bar\Sigma_{\mathcal{C}}$ in the corresponding EHH (see App. A.2) to form a unitary symmetry $\bar U = \bar{\mathcal{S}}\bar\Sigma_{\mathcal{C}}$ with $\bar U^2 = +1$. The unitary $\bar U$ satisifies $[\bar{\mathcal{S}}, \bar U] = [\bar\Sigma_{\mathcal{C}}, \bar U] = 0$. Hence we obtain Hermitian class AIII in each $\bar U$ subspace, giving a $\mathbb{Z} \oplus \mathbb{Z}$ classification in 3D and 1D, while 2D is trivial [79]. By modding out line-gap phases, this is reduced to a $\mathbb{Z}$ classification where the nontrivial element corresponds to having only a single $\bar U$ subspace being nontrivial [39].

Away from $E_0 = 0$, we are only left with the chiral symmetry $\bar\Sigma_{\mathcal{C}}$, resulting in Hermitian class AIII. Without additional crystalline symmetries, the zero energy modes arising in the flux Dirac theory of systems in Hermitian class AIII are not protected (see App. B.3.2). Adding a crystalline symmetry $\mathcal{I}^{\dagger,S_-}$, however, quantizes the flux and allows for a stable flux response. The results derived in App. B.3.2 show the presence of zero energy states under OBC along the flux tubes, which can be distinguished from the surface signature for $\phi = 0$. Consequently, these zero modes appear extensively in the point gap of the NH model, constituting a *higher-order flux skin effect* (see Sec. 5) for NH class A$^S$ + $\mathcal{I}^{\dagger,S_-}$.

### B.5.5 Class D

Models in NH class D possess a PHS $U_{\mathcal{P}}$ with $U_{\mathcal{P}}U_{\mathcal{P}}^* = +1$, yielding a $\mathbb{Z}$ classification in 3D, while lower dimensions are trivial. As outlined in App. A.2, the corresponding EHH introduces a chiral symmetry $\bar\Sigma_{\mathcal{C}}$, which combines with PHS to a TRS $\bar U_{\mathcal{T}} = \bar U_{\mathcal{P}}\bar\Sigma_{\mathcal{C}}$ which satisfies $\bar U_{\mathcal{T}}\bar U_{\mathcal{T}}^* = -1$. Hence we obtain Hermitian class DIII, giving a $\mathbb{Z}$ classification in 3D [79]. The flux response of the EHH in Hermitian class DIII forms a gapless helical metal along the flux tube (see App. B.3.3).

Away from $E_0 = 0$, we are only left with the chiral symmetry $\bar\Sigma_{\mathcal{C}}$, resulting in Hermitian class AIII. Hermitian class AIII allows to gap the helical metal by introducing suitable mass terms. Consequently, only at $E_0 = 0$ we expect flux localized modes to appear in the NH point

gap. The number of modes does not scale extensively with the system size, the NH system only pins the Hermitian surface state of the extended model. This surface state is a Majorana, models in NH class D therefore show a flux Majorana mode, as introduced in Sec. 6. Note that due to the $\mathbb{Z}_2$ line gap classification in 1D, flux defects only probe the $\mathbb{Z}_2$ nature of the 3D $\mathbb{Z}$ invariant in NH class D.

### B.5.6 Class $\mathrm{D}^{S_+} + \mathcal{I}^{\dagger,S_-}$

Models in NH class $\mathrm{D}^{S_+}$ possess PHS $U_{\mathcal{P}}$ ($U_{\mathcal{P}}U_{\mathcal{P}}^* = +1$) and sublattice symmetry $\mathcal{S}$, with $[U_{\mathcal{P}}, \mathcal{S}] = 0$. As outlined in App. A.2, the corresponding EHH introduces a chiral symmetry $\bar{\Sigma}_{\mathcal{C}}$, which combines with sublattice symmetry to form a unitary symmetry $\bar{U} = \bar{\mathcal{S}}\bar{\Sigma}_{\mathcal{C}}$ with $\bar{U}^2 = +1$. The unitary $\bar{U}$ satisifies $\{\bar{U}_{\mathcal{P}}, \bar{U}\} = [\bar{\mathcal{S}}, \bar{U}] = [\bar{\Sigma}_{\mathcal{C}}, \bar{U}] = 0$. We may furthermore define a TRS $\bar{U}_{\mathcal{T}} = \bar{U}_{\mathcal{P}}\bar{\Sigma}_{\mathcal{C}}$ which satisfies $\bar{U}_{\mathcal{T}}\bar{U}_{\mathcal{T}}^* = -1$ and anti-commutes with $\bar{U}$. Hence the eigenspaces of $\bar{U}$ are not independent and individually enjoy a chiral symmetry, leading to Hermitian class AIII with a $\mathbb{Z}$ classification in 3D [79]. Without additional crystalline symmetries, the zero energy modes arising in the flux Dirac theory of systems in Hermitian class AIII are not protected (see App. B.3.2). Adding a crystalline symmetry $\mathcal{I}^{\dagger,S_-}$, however, quantizes the flux and allows for a stable flux response. The results derived in App. B.3.2 show the presence of one zero energy mode per flux under OBC, occurring in each $\bar{U}$ subspace.

Away from $E_0 = 0$, we can only form the TRS $\bar{U}_{\mathcal{T}}' = \bar{\mathcal{S}}\bar{U}_{\mathcal{P}}$ with $\bar{U}_{\mathcal{T}}'\bar{U}_{\mathcal{T}}'^* = +1$ and PHS $\bar{U}_{\mathcal{P}}' = \bar{\Sigma}_{\mathcal{C}}\bar{U}_{\mathcal{T}}'$ with $\bar{U}_{\mathcal{P}}'\bar{U}_{\mathcal{P}}'^* = -1$ (see App. A.2). This results in Hermitian class CI which does not protect a Kramers degeneracy. Consequently, the pair of flux modes can be now gapped, yielding a higher order response only at $E_0 = 0$. Models in NH class $\mathrm{D}^{S_+} + \mathcal{I}^{\dagger,S_-}$ therefore show a higher-order Hermitian flux response, the higher-order flux Majorana mode, as introduced in Sec. 6.

### B.5.7 Class $\mathrm{D}^{S_-}$

As for the case of 2D, models in NH class $\mathrm{D}^{S_-}$ possess PHS $U_{\mathcal{P}}$ ($U_{\mathcal{P}}U_{\mathcal{P}}^* = +1$) and sublattice symmetry $\mathcal{S}$, with $\{U_{\mathcal{P}}, \mathcal{S}\} = 0$. As outlined in App. A.2, the corresponding EHH introduces a chiral symmetry $\bar{\Sigma}_{\mathcal{C}}$, which combines with sublattice symmetry to form a unitary symmetry $\bar{U} = \bar{\mathcal{S}}\bar{\Sigma}_{\mathcal{C}}$ with $\bar{U}^2 = +1$. The unitary $\bar{U}$ satisifies $[\bar{U}_{\mathcal{P}}, \bar{U}] = [\bar{\mathcal{S}}, \bar{U}] = [\bar{\Sigma}_{\mathcal{C}}, \bar{U}] = 0$. We may furthermore define a TRS $\bar{U}_{\mathcal{T}} = \bar{U}_{\mathcal{P}}\bar{\Sigma}_{\mathcal{C}}$ which satisfies $\bar{U}_{\mathcal{T}}\bar{U}_{\mathcal{T}}^* = -1$. Hence we obtain Hermitian class DIII in each $\bar{U}$ subspace, giving a $\mathbb{Z} \oplus \mathbb{Z}$ classification in 3D [79]. By modding out line-gap phases, this is reduced to a $\mathbb{Z}$ classification where the nontrivial element corresponds to having only a single $\bar{U}$ subspace being nontrivial [39]. Hence the full Hermitian spectrum hosts a helical metal under $\pi$-flux insertion (see App. B.3.3).

Away from $E_0 = 0$, we can only form the TRS $\bar{U}_{\mathcal{T}}' = \bar{\mathcal{S}}\bar{U}_{\mathcal{P}}$ with $\bar{U}_{\mathcal{T}}'\bar{U}_{\mathcal{T}}'^* = -1$ and PHS $\bar{U}_{\mathcal{P}}' = \bar{\Sigma}_{\mathcal{C}}\bar{U}_{\mathcal{T}}'$ with $\bar{U}_{\mathcal{P}}'\bar{U}_{\mathcal{P}}'^* = +1$ (see App. A.2). This results in Hermitian class DIII, such that the flux-bound helical metal cannot move away from zero energy. Hence the entire point gap of a corresponding NH system fills with flux tube localized modes. Consequently, models in NH class $\mathrm{D}^{S_-}$ quantize the flux to $\phi = 0, \pi$ and show a NH flux spectral jump, discussed in Sec. 4. Note that due to the $\mathbb{Z}_2$-classification of NH class $\mathrm{D}^{S_-}$ in 2D, flux defects only probe the $\mathbb{Z}_2$ nature of the 3D $\mathbb{Z}$ invariant.

### B.5.8 Class $\mathrm{DIII}^{S_{+-}}$

As for the case of 2D, models in NH class $\mathrm{DIII}^{S_{+-}}$ possess TRS, PHS and chiral symmetry with $(U_{\mathcal{T}}U_{\mathcal{T}}^*, U_{\mathcal{P}}U_{\mathcal{P}}^*) = (-1, 1)$, where $U_{\mathcal{C}} = U_{\mathcal{T}}U_{\mathcal{P}}$, as well as sublattice symmetry $\mathcal{S}$. As outlined in App. A.2, the corresponding EHH at $E_0 = 0$ obtains TRS $\bar{U}_{\mathcal{T}}$, PHS $\bar{U}_{\mathcal{P}}$ and three chiral symmetries $\bar{U}_{\mathcal{C}}, \bar{\Sigma}_{\mathcal{C}}, \bar{\mathcal{S}}$. The chiral symmetries can be combined to two commuting unitary symmetries $\bar{U}_1 = \bar{U}_{\mathcal{C}}\bar{\Sigma}_{\mathcal{C}}$ with $\bar{U}_1^2 = -1$ and $\bar{U}_2 = \bar{\mathcal{S}}\bar{\Sigma}_{\mathcal{C}}$ with $\bar{U}_2^2 = +1$.

Since $\bar{U}_1$ has an imaginary spectrum and

$$\{\bar{U}_{\mathcal{T}}, \bar{U}_1\} = [\bar{U}_{\mathcal{P}}, \bar{U}_1] = \{\bar{U}_{\mathcal{C}}, \bar{U}_1\} = \{\bar{\Sigma}_{\mathcal{C}}, \bar{U}_1\} = \{\bar{\mathcal{S}}, \bar{U}_1\},$$

the eigenspaces of $\bar{U}_1$ are not independent and individually enjoy $\bar{U}_{\mathcal{T}}$ and $\bar{U}_2$ symmetry. Moreover, we have $[\bar{U}_{\mathcal{T}}, \bar{U}_2] = [\bar{U}_{\mathcal{P}}, \bar{U}_2] = [\bar{\mathcal{S}}, \bar{U}_2] = [\bar{U}_{\mathcal{C}}, \bar{U}_2] = [\bar{\Sigma}_{\mathcal{C}}, \bar{U}_2] = 0$, so that the $\bar{U}_2$ eigenspaces are independent and individually preserve $\bar{U}_{\mathcal{T}}$ symmetry. They therefore lie in Hermitian class AII and yield a $\mathbb{Z}_2 \oplus \mathbb{Z}_2$ classification in 3D [79] that is reduced to $\mathbb{Z}_2$ by line-gap phases [39]. The nontrivial element corresponds to having only a single $\bar{U}_2$ subspace nontrivial. In App. B.3.1, we derived the flux response of EHHs in Hermitian class AII, which shows a helical metal pinned to the flux defects.

Away from $E_0 = 0$, we can only form the TRS $\bar{U}'_{\mathcal{T}} = \bar{\mathcal{S}}\bar{U}_{\mathcal{P}}$ with $\bar{U}'_{\mathcal{T}}\bar{U}'^*_{\mathcal{T}} = -1$ and PHS $\bar{U}'_{\mathcal{P}} = \bar{\Sigma}_{\mathcal{C}}\bar{U}'_{\mathcal{T}}$ with $\bar{U}'_{\mathcal{P}}\bar{U}'^*_{\mathcal{P}} = +1$ (see App. A.2). This results in Hermitian class DIII, such that the flux-bound helical metal cannot move away from zero energy. Hence the entire point gap of a corresponding NH system fills with an extensive number of states at a single momentum $k_{\parallel}$. Consequently, models in NH class DIII$^{S+-}$ quantize the flux to $\phi = 0, \pi$ and show a NH flux spectral jump, discussed in Sec. 4.

### B.5.9 Class AII$^{S_+}$ + $\mathcal{I}^{\dagger, S_-, T_-}$

NH class AII$^{S_+}$ possesses TRS $U_{\mathcal{T}}$ ($U_{\mathcal{T}}U_{\mathcal{T}}^* = -1$) and sublattice symmetry $\mathcal{S}$, with $[U_{\mathcal{T}}, \mathcal{S}] = 0$. As outlined in App. A.2, the corresponding EHH introduces a chiral symmetry $\bar{\Sigma}_{\mathcal{C}}$, which combines with sublattice symmetry to form a unitary symmetry $\bar{U} = \bar{\mathcal{S}}\bar{\Sigma}_{\mathcal{C}}$ with $\bar{U}^2 = +1$. The unitary $\bar{U}$ satisifies $[\bar{U}_{\mathcal{T}}, \bar{U}] = [\bar{\mathcal{S}}, \bar{U}] = [\bar{\Sigma}_{\mathcal{C}}, \bar{U}] = 0$. We may furthermore define a PHS $\bar{U}_{\mathcal{P}} = \bar{U}_{\mathcal{T}}\bar{\Sigma}_{\mathcal{C}}$ which satisfies $\bar{U}_{\mathcal{P}}\bar{U}_{\mathcal{P}}^* = -1$ and commutes with $\bar{U}$. Hence the eigenspaces of $\bar{U}$ are independent and individually enjoy TRS, PHS and chiral symmetry, leading to Hermitian class CII with a $\mathbb{Z}_2$ classification in each sector in 3D [79]. Without additional crystalline symmetries, the zero energy modes arising in the flux Dirac theory of systems in Hermitian class CII are not protected (see App. B.3.5). Adding a crystalline symmetry $\mathcal{I}^{\dagger, S_-, T_-}$, however, quantizes the flux and allows for a stable flux response. Introducing a $\pi$-flux results in the presence of zero energy states under OBC along the flux tubes, in addition to the surface signature for $\phi = 0$.

Away from $E_0 = 0$, we are only left with the chiral symmetry $\bar{\Sigma}_{\mathcal{C}}$, resulting in Hermitian class AIII. As Hermitian class AIII with additional crystalline symmetry $\mathcal{I}^{\dagger, S_-, T_-}$ protects flux modes localized at the end of the flux tubes, these zero modes appear extensively in the point gap of the NH model, constituting a *higher-order flux skin effect* (see Sec. 5) for NH class AII$^{S_+}$ + $\mathcal{I}^{\dagger, S_-, T_-}$.

### B.5.10 Class C

Models in NH class C possess a PHS $U_{\mathcal{P}}$ with $U_{\mathcal{P}}U_{\mathcal{P}}^* = -1$, yielding a $2\mathbb{Z}$ classification in 3D, while lower dimensions are trivial. As outlined in App. A.2, the corresponding EHH introduces a chiral symmetry $\bar{\Sigma}_{\mathcal{C}}$, which combines with PHS to a TRS $\bar{U}_{\mathcal{T}} = \bar{U}_{\mathcal{P}}\bar{\Sigma}_{\mathcal{C}}$ which satisfies $\bar{U}_{\mathcal{T}}\bar{U}_{\mathcal{T}}^* = +1$. Hence we obtain Hermitian class CI, giving a $2\mathbb{Z}$ classification in 3D [79]. The flux response of the EHH in Hermitian class CI is trivial, without protected flux localized zero modes (see App. B.3.4). Consequently, systems in NH class C do not show flux induced states within the NH point gap. Therefore, the flux response for this NH symmetry class is trivial.

### B.5.11 Class C$^{S_+}$ + $\mathcal{I}^{\dagger, S_-}$

Models in NH class C$^{S_+}$ possess PHS $U_{\mathcal{P}}$ ($U_{\mathcal{P}}U_{\mathcal{P}}^* = -1$) and sublattice symmetry $\mathcal{S}$, with $[U_{\mathcal{P}}, \mathcal{S}] = 0$. As outlined in App. A.2, the corresponding EHH introduces a chiral symmetry $\bar{\Sigma}_{\mathcal{C}}$, which combines with sublattice symmetry to form a unitary symmetry $\bar{U} = \bar{\mathcal{S}}\bar{\Sigma}_{\mathcal{C}}$ with

$\bar{U}^2 = +1$. The unitary $\bar{U}$ satisifies $\{\bar{U}_{\mathcal{P}}, \bar{U}\} = [\bar{\mathcal{S}}, \bar{U}] = [\bar{\Sigma}_{\mathcal{C}}, \bar{U}] = 0$. We may furthermore define a TRS $\bar{U}_{\mathcal{T}} = \bar{U}_{\mathcal{P}} \bar{\Sigma}_{\mathcal{C}}$ which satisfies $\bar{U}_{\mathcal{T}} \bar{U}_{\mathcal{T}}^* = +1$ and anti-commutes with $\bar{U}$. Hence the eigenspaces of $\bar{U}$ are not independent and individually enjoy a chiral symmetry, leading to Hermitian class AIII with a $\mathbb{Z}$ classification in 3D [79]. Without additional crystalline symmetries, the zero energy modes arising in the flux Dirac theory of systems in Hermitian class AIII are not protected (see App. B.3.2). Adding a crystalline symmetry $\mathcal{I}^{\dagger, S_-}$, however, quantizes the flux and allows for a stable higher order flux response, with a single mode at the end of each flux tube per $\bar{U}$ eigenspace.

Away from $E_0 = 0$, we can only form the TRS $\bar{U}_{\mathcal{T}}' = \bar{\mathcal{S}} \bar{U}_{\mathcal{P}}$ with $\bar{U}_{\mathcal{T}}' \bar{U}_{\mathcal{T}}'^* = -1$ and PHS $\bar{U}_{\mathcal{P}}' = \bar{\Sigma}_{\mathcal{C}} \bar{U}_{\mathcal{T}}'$ with $\bar{U}_{\mathcal{P}}' \bar{U}_{\mathcal{P}}'^* = +1$ (see App. A.2). This results in Hermitian class DIII, such that the two flux modes in each $\bar{U}$ eigenspace are still protected. Consequently, these zero modes appear extensively in the point gap of the NH model. This constitutes a higher order skin effect, with skin modes appearing at the ends of the flux tubes in a finite geometry. The precise localization of skin modes is then determined by the present crystalline symmetry $\mathcal{I}^{\dagger, S_-}$. This is the defining signature of the NH *higher-order flux skin effect*, discussed in Sec. 5.

### B.5.12 Class C$^{S_-}$

NH class C$^{S_-}$ possesses PHS $U_{\mathcal{P}}$ ($U_{\mathcal{P}} U_{\mathcal{P}}^* = -1$) and sublattice symmetry $\mathcal{S}$, with $\{U_{\mathcal{P}}, \mathcal{S}\} = 0$. As outlined in App. A.2, the corresponding EHH introduces a chiral symmetry $\bar{\Sigma}_{\mathcal{C}}$, which combines with sublattice symmetry to form a unitary symmetry $\bar{U} = \bar{\mathcal{S}} \bar{\Sigma}_{\mathcal{C}}$ with $\bar{U}^2 = +1$. The unitary $\bar{U}$ satisifies $[\bar{U}_{\mathcal{P}}, \bar{U}] = [\bar{\mathcal{S}}, \bar{U}] = [\bar{\Sigma}_{\mathcal{C}}, \bar{U}] = 0$. We may furthermore define a TRS $\bar{U}_{\mathcal{T}} = \bar{U}_{\mathcal{P}} \bar{\Sigma}_{\mathcal{C}}$ which satisfies $\bar{U}_{\mathcal{T}} \bar{U}_{\mathcal{T}}^* = +1$, commuting with $\bar{U}$. Hence we obtain Hermitian class CI in each $\bar{U}$ subspace, giving a $2\mathbb{Z} \oplus 2\mathbb{Z}$ classification in 3D [79]. By modding out line-gap phases, this is reduced to a $\mathbb{Z}$ classification where the nontrivial element corresponds to having only a single $\bar{U}$ subspace being nontrivial [39].

Away from $E_0 = 0$, we can only form the TRS $\bar{U}_{\mathcal{T}}' = \bar{\mathcal{S}} \bar{U}_{\mathcal{P}}$ with $\bar{U}_{\mathcal{T}}' \bar{U}_{\mathcal{T}}'^* = +1$ and PHS $\bar{U}_{\mathcal{P}}' = \bar{\Sigma}_{\mathcal{C}} \bar{U}_{\mathcal{T}}'$ with $\bar{U}_{\mathcal{P}}' \bar{U}_{\mathcal{P}}'^* = -1$ (see App. A.2). This results again in Hermitian class CI, which shows a trivial flux response (see App. B.3.4). Consequently, systems in NH class C$^{S_-}$ do not show flux induced states within the NH point gap. Therefore, the flux response for this NH symmetry class is trivial.

## C  Scaling analysis of flux defect localized modes

This appendix highlights the scaling of flux localized modes with system extent for the flux responses discussed in the main text.

### C.1  Scaling of flux core-localized modes in 2D

Sec. 3 predicts a skin effect localizing modes at the points where flux cores with $\phi = \pi$ pierce a 2D system. Their number scales with the system size, precisely with the extension of the system parallel to the line connecting the two flux cores $L_\perp$, as shown in Fig. 8. All states in this 1D slice collapse towards the flux cores. For small separations $d$, states at the two flux cores can hybridize, yielding a smaller number of localized states. The topologically relevant regime has both flux cores well separated, for which the number of skin modes does not change with $d$. Consequently, all $L_\perp$ states experience the flux skin effect, irrespective of the extension of the system to the left and right of the flux defects.

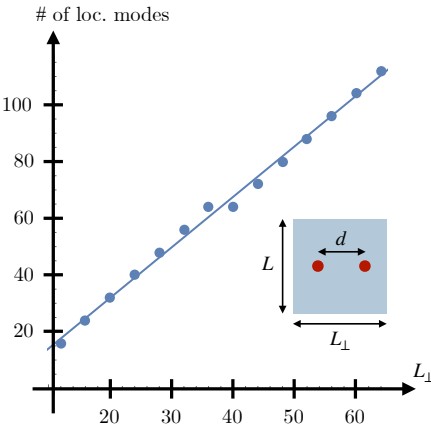

Figure 8: **Scaling analysis for the NH flux skin effect.** The number of modes with more than 70% localization at the flux cores scales linearly with the extent of the system along the direction connecting the two flux cores $L_\perp$. The separation between the flux cores $d$ is held fixed. The fit is given by $\# = -3.81538 + 1.78462\, L_\perp$, using a model in NH class AII$^\dagger$ [Eq. (E.1)].

## C.2  Scaling of flux tube-localized modes in 3D

In 3D, NH flux defects probe 4 unique topological phases: the flux spectral jump, the higher-order flux skin effect, the NH flux Majorana mode and its higher-order cousin. For a $\pi$-flux, the flux spectral jump causes an extensive number of defect localized modes. Their number scales with the extension of the system parallel to the line connecting the two flux tubes $L_\perp$, as shown in Fig. 9a. Conversely, the higher-order flux skin effect appears along $\pi$-flux tubes, thereby localizing $\mathcal{O}(L_\parallel)$ modes, where $L_\parallel$ is the extension of the system along the defect as shown in Fig. 9b. In contrast, flux Majorana modes appear only as isolated modes, without a connected skin effect. Their number then derives from the EHH Dirac theory, which predicts two zero-energy modes per flux tube, localized at the flux defects (see App. B.2.4). Correspondingly, the model in NH class D hosts one mode per flux tube (see Fig. 9c and App. B.5.5). Higher-order flux Majorana modes appear in addition to the surface state in NH class D$^{S_+}$ (see App. B.5.6). The number of modes localized at the ends of the flux tubes, however, does also not scale with the system size, indicating the absence of a flux induced skin effect (see Fig. 9d).

# D  More details on the higher-order flux skin effect

This appendix provides details on the fractional nature of the higher-order flux skin effect and the role of (pseudo-)inversion symmetry in it. Additionally, we provide details for NH symmetry classes AII$^{S_+}$ and C$^{S_+}$.

## D.1  Fractional nature of the higher-order flux skin effect

The NH higher-order flux skin effect results from the presence of one flux-end localized zero energy mode per flux tube in the corresponding EHH. The single flux bound state is fractional, as there exists no 1D system with a single end state. In this appendix, we investigate the fractional nature of this state by considering the weight of a given chirality on each surface, which reveals an imbalance only present for finite flux $\phi = \pi$.

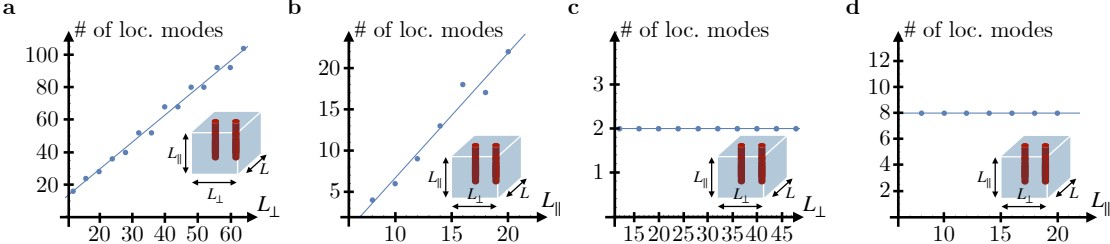

Figure 9: **Scaling analysis for the flux effects in 3D. a** In the flux spectral jump, the number of modes with more than 70% localization at the flux tubes scales linearly with the extent of the system connecting the two flux tubes $L_\perp$. The fit is given by $\# = -4.04396 + 1.67033\,L_\perp$ for a model in NH class AII$^\dagger$ [Eq. (E.7)]. **b** The higher-order flux skin effect localizes all modes along the flux defects, thereby scaling with $L_\parallel$. The fit is given by $\# = -8.53571 + 1.51786\,L_\parallel$ for modes localized at the ends of the flux tubes. Scaling generated for a model in NH class A with fixed size $L_\perp = 20$ unit cells [Eq. (E.8)]. **c** The number of Majorana modes with more than 70% localization at the flux tubes does not scale with the system size. The number follows directly from the Dirac flux theory for models in NH class D [Eq. (E.11)] **d** The number of higher-order Majorana modes with more than 55% localization at the flux tubes does not scale with the system size, here specifically shown for the dimension along the length of the flux defect, $L_\parallel$. Scaling generated for a model in NH class D$^{S_+}$ with fixed size $L_\perp = 20$ unit cells [Eq. (E.12)].

The eigenstates of the EHH are given as

$$\bar{H}|\psi_j\rangle = E_j|\psi_j\rangle\,, \qquad j \in \Xi = \{1,\dots,L_\perp^2 \cdot L_\parallel \cdot L_i\}\,, \tag{D.1}$$

where $L_\perp, L_\parallel$ are the system dimensions, with $L_i$ describing internal (sublattice, spin) degrees of freedom. We can then consider an interval around zero energy $-\epsilon < E_j < \epsilon$: requiring that we do not separate degenerate states, the set of states in the interval $2\epsilon$ forms the subset $\mathcal{E} \subset \Xi$, for instance as highlighted in the inset of Fig. 5b. Expressing the chiral symmetry of the EHH $\bar{\Sigma}_\mathcal{C}$ in this subspace,

$$\Sigma_{ij} = \langle\psi_i|\bar{\Sigma}_\mathcal{C}|\psi_j\rangle\,, \tag{D.2}$$

allows to diagonalize $\Sigma$

$$\Sigma \mathbf{v}_l^\pm = \pm\mathbf{v}_l^\pm\,, \qquad l \in \{1,\dots,n_\pm\}\,, \tag{D.3}$$

and obtain eigenstates of the chiral symmetry $\bar{\Sigma}_\mathcal{C}$. We denote eigenvectors with positive (negative) chiral eigenvalue as $\mathbf{v}_l^+$ ($\mathbf{v}_l^-$) and their number as $n_+$ ($n_-$). We are interested in the chirality of the original eigenstates of the EHH $\bar{H}$, which we expand in the obtained chiral basis:

$$|\psi_\pm\rangle = \sum_{l,j} v_{l,j}^\pm|\psi_j\rangle\,, \tag{D.4}$$

where $|\psi_\pm\rangle$ are now eigenstates of positive (+) or negative (-) chirality. For a system without flux defects, for instance in Hermitian class AIII as considered in Sec. 5, each surface has zero net-chirality [79]. We calculate the net chirality by summing all states of (+)/(-) chirality for the top/bottom surface, $\rho_{\text{top,bottom}}^\pm$, and investigating their differences. Specifically, we use

$$\rho_{\text{top}}^\pm = \sum_{\mathbf{r}_\perp,\mathbf{r}_\parallel=L_\parallel}^{|\psi_\pm(\mathbf{r}_\parallel)|^2<\delta} |\psi_\pm(\mathbf{r})|^2\,, \tag{D.5}$$

$$\rho_{\text{bot}}^{\pm} = \sum_{\mathbf{r}_\perp, \mathbf{r}_\parallel = 0}^{|\psi_{\pm}(\mathbf{r}_\parallel)|^2 < \delta} |\psi_{\pm}(\mathbf{r})|^2, \tag{D.6}$$

where we restrict the summation for bottom (top) surface, denoted by $\mathbf{r}_\parallel = 0$ ($\mathbf{r}_\parallel = L_\parallel$), once the states $\psi_\pm$ decay in the bulk ($|\psi_\pm(\mathbf{r}_\parallel)|^2 < \delta$, $\delta \ll 1$).

Forming the differences between top and bottom surface for a fixed chirality, e.g. (+), yields

$$\Delta^+ = |\rho_{\text{top}}^+ - \rho_{\text{bot}}^+|. \tag{D.7}$$

In the absence of a flux defect, $\phi = 0$, each chirality has equal weight one each surface, $\Delta^\pm = 0$. Conversely, for a nontrivial flux $\phi = \pi$, each chirality has predominant weight on one surface. For a PBC system with two flux tubes, $\Delta^\pm$ is equal to one: there is a single state per chirality per flux tube (see Fig. 10). Viewing a flux tube as an effective 1D model highlights the fractional nature of this response: A 1D model always contains end states of both chiralities, for instance as in the Su-Schrieffer-Heeger model [81]. The fact that we obtain a single state per chirality shows the fractional nature in the EHH, causing the higher-order flux skin effect in the NH system.

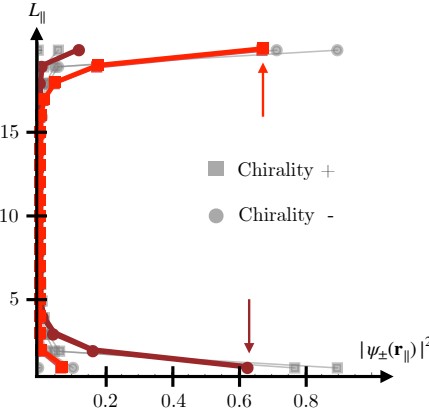

Figure 10: **Fractional nature of the NH higher-order flux skin effect.** Eigenstates of the chiral symmetry in the EHH appear with zero net-chirality per surface in the absence of a magnetic flux $\phi$. Introducing a flux defect with $\phi = \pi$ leads to additional contributions to top and bottom surface (depicted in red), having opposite chirality. Panel is generated for a model of size $20 \times 20 \times 20$ unit cells in NH class A with additional pseudo-inversion symmetry (see App. E.2.2).

## D.2 Role of inversion symmetry in the higher-order flux skin effect

This appendix investigates the effect of (pseudo-)inversion symmetry on NH symmetry classes with a higher-order flux skin effect under $\pi$-flux insertion. The relevant NH symmetry classes are A, $A^S$, $AII^{S_+}$ and $C^{S_+}$. In order to obtain a nontrivial response to flux defects in these classes, we require the presence of additional crystalline symmetries, restricting the magnetic flux $\phi$ to $0, \pi$ in PBC, while still allowing for nontrivial topology. Specifically, we investigate inversion symmetry

$$\mathcal{I}\mathcal{H}(\boldsymbol{k})\mathcal{I}^\dagger = \mathcal{H}(-\boldsymbol{k}), \tag{D.8}$$

appearing in the EHH as

$$\bar{\mathcal{I}}\bar{\mathcal{H}}(\boldsymbol{k})\bar{\mathcal{I}}^\dagger = -\bar{\mathcal{H}}(\boldsymbol{k}), \quad \bar{\mathcal{I}} = \begin{pmatrix} \mathcal{I} & 0 \\ 0 & \mathcal{I} \end{pmatrix}, \tag{D.9}$$

and pseudo-inversion symmetry

$$\mathcal{I}\mathcal{H}(\boldsymbol{k})^{\dagger}\mathcal{I}^{\dagger} = \mathcal{H}(-\boldsymbol{k}),\tag{D.10}$$

appearing in the EHH as

$$\bar{\mathcal{I}}\bar{\mathcal{H}}(\boldsymbol{k})\bar{\mathcal{I}}^{\dagger} = -\bar{\mathcal{H}}(\boldsymbol{k}),\quad \bar{\mathcal{I}} = \begin{pmatrix} 0 & \mathcal{I} \\ \mathcal{I} & 0 \end{pmatrix}.\tag{D.11}$$

We have to identify choices of (pseudo-)inversion symmetry that leave the intrinsic point gap classification invariant, i.e. do not trivialize the flux response.

### D.2.1 Class A

In NH symmetry class A, the EHH can be formed as outlined in Eq. (4) of the main text, which adds a chiral symmetry $\bar{\Sigma}_{\mathcal{C}}$, resulting in Hermitian class AIII. We can either use an inversion symmetry (D.8) or a pseudo-inversion symmetry (D.10) to fix the flux $\phi$ to $0, \pi$ in PBC in the NH Hamiltonian. However, these two choices differ in their commutation with the chiral symmetry $\bar{\Sigma}_{\mathcal{C}}$ when considering the EHH: Whereas "normal" inversion symmetry commutes with chiral symmetry

$$\left[\bar{\Sigma}_{\mathcal{C}}, \begin{pmatrix} \mathcal{I} & 0 \\ 0 & \mathcal{I} \end{pmatrix}\right] = 0,\tag{D.12}$$

pseudo-inversion symmetry anticommutes,

$$\left\{\bar{\Sigma}_{\mathcal{C}}, \begin{pmatrix} 0 & \mathcal{I} \\ \mathcal{I} & 0 \end{pmatrix}\right\} = 0.\tag{D.13}$$

This means both chiral subspaces have the same inversion eigenvalues. Consequently, at every inversion-symmetric momenta in the Brillouin zone, the occupied and unoccupied subspace have the same number of positive and negative inversion eigenvalues. Therefore every Hermitian model with these symmetry properties cannot have a band inversion and so is topologically trivial [82]. As topological zero modes of Hermitian extended models stand in one-to-one correspondence with NH modes within a point-gapped bulk (see Sec 2.2), also the corresponding NH model is trivial. Accordingly, Ref. [83] outlines that with commuting inversion symmetry, the resulting Hermitian class AIII$^{\mathcal{I}_{+}}$ is trivial, in agreement with the vanishing 3D winding number of the NH system. Conversely, anticommuting pseudo-inversion symmetry yields a EHH in Hermitian class AIII$^{\mathcal{I}_{-}}$ which is classified by a $\mathbb{Z}$ index. Contrary to normal inversion, pseudo-inversion hence allows for a nontrivial flux response, the higher-order flux skin effect.

### D.2.2 Class A$^{S}$

In NH symmetry class A$^{S}$, the EHH can be formed for each sublattice symmetry block (see App. B.5.4) as outlined in Eq. (4). This construction adds a chiral symmetry $\bar{\Sigma}_{\mathcal{C}}$, resulting in Hermitian class AIII in each block, of which only one has to be nontrivial. Again, we can either use an inversion symmetry (D.8) or a pseudo-inversion symmetry (D.10) to fix the flux $\phi$ to $0, \pi$ in PBC in the NH Hamiltonian. However, these two choices differ in their commutation with the chiral symmetry $\bar{U}_{\mathcal{C}}$ when considering the EHH: Whereas "normal" inversion symmetry commutes with chiral symmetry pseudo-inversion symmetry anticommutes (see Sec. D.2.1). Additionally, the sublattice symmetry $\bar{\mathcal{S}}$ of the NH Hamiltonian acts as a chiral symmetry of $\bar{\mathcal{H}}$, too. Depending on the commutation or anti-commutation of both $\bar{U}_{\mathcal{C}}$ and $\bar{\mathcal{S}}$ with (pseudo-)inversion symmetry the topological classification differs. Ref. [83] outlines that both chiral symmetries have to anti-commute with inversion symmetry to form a nontrivial phase classified

by a $\mathbb{Z}$ index. Correspondingly, an inversion symmetry in the NH Hamiltonian always results in a trivial phase. Pseudo-inversion symmetry on the other hand only results in a nontrivial flux response for an anti-commuting sublattice symmetry $\mathcal{S}$. Contrary to normal inversion, pseudo-inversion hence allows for a nontrivial flux response, the higher-order flux skin effect.

### D.2.3 Class AII$^{S_+}$

In NH symmetry class AII$^{S_-}$, the EHH follows as outlined in Eq. (4) (see Sec. B.5.9). The EHH possesses a unitary symmetry $\bar{U}$, whose eigenspaces are independent and individually enjoy TRS, PHS and chiral symmetry, leading to Hermitian class CII in each sector. A stable flux response in this class requires the presence of a crystalline symmetry: we can either use an inversion symmetry (D.8) or a pseudo-inversion symmetry (D.10). We have to analyse these two choices with respect to their commutation with TRS and PHS, to derive their topological classification [83]. In the EHH we can form two TRS:

$$\bar{U}_{\mathcal{T}} = \begin{pmatrix} U_{\mathcal{T}} & 0 \\ 0 & U_{\mathcal{T}} \end{pmatrix}, \tag{D.14}$$

and

$$\bar{U}_{\mathcal{T}} = \begin{pmatrix} U_{\mathcal{T}}\mathcal{S} & 0 \\ 0 & -U_{\mathcal{T}}\mathcal{S} \end{pmatrix}. \tag{D.15}$$

Similarly, we can form two PHS,

$$\bar{U}_{\mathcal{P}} = \begin{pmatrix} U_{\mathcal{T}} & 0 \\ 0 & -U_{\mathcal{T}} \end{pmatrix}, \tag{D.16}$$

and

$$\bar{U}_{\mathcal{P}} = \begin{pmatrix} U_{\mathcal{T}}\mathcal{S} & 0 \\ 0 & U_{\mathcal{T}}\mathcal{S} \end{pmatrix}. \tag{D.17}$$

A $4\mathbb{Z}$ classified phase is obtained if both TRSs commute while the PHSs anticommute with inversion. Conversely, if both TRSs anticommute while the PHSs commute with inversion, we obtain a $\mathbb{Z}_2$ classification [83]. Only the latter is compatible with the $\mathbb{Z}_2$-point gap classification without crystalline symmetries. All other cases, also including inversion symmetry, do not yield a nontrivial response. Therefore, we should choose a pseudo-inversion symmetry commuting with sublattice symmetry $\mathcal{S}$, but anticommuting with TRS $U_{\mathcal{T}}$. Contrary to normal inversion, pseudo-inversion hence allows for a nontrivial flux response, the higher-order flux skin effect.

### D.2.4 Class C$^{S_+}$

In NH symmetry class C$^{S_+}$, the EHH possesses chiral symmetries $\bar{\mathcal{S}}$, $\bar{\Sigma}_{\mathcal{C}}$ in each independent unitary eigenspace (see App. B.5.11). Consequently, the EHH sits in Hermitian class AIII for each block, of which only one has to be nontrivial. We can either use an inversion symmetry (D.8) or a pseudo-inversion symmetry (D.10) to fix the flux $\phi$ to $0, \pi$ in the NH Hamiltonian. However, these two choices differ in their commutation with the chiral symmetry $\bar{\Sigma}_{\mathcal{C}}$ when considering the EHH: Whereas "normal" inversion symmetry commutes with chiral symmetry, pseudo-inversion symmetry anticommutes (see Sec. D.2.1). Additionally, the sublattice symmetry $\bar{\mathcal{S}}$ of the NH Hamiltonian acts as a chiral symmetry of $\bar{\mathcal{H}}$, too. Depending on the commutation or anti-commutation of both $\bar{\Sigma}_{\mathcal{C}}$ and $\bar{\mathcal{S}}$ with (pseudo-)inversion symmetry the topological classification differs. Ref. [83] outlines that both chiral symmetries have to anti-commute with inversion symmetry to form a nontrivial phase classified by a $\mathbb{Z}$ index. Correspondingly, an inversion symmetry in the NH Hamiltonian always results in a trivial phase. Pseudo-inversion symmetry on the other hand only results in a nontrivial flux response for an anti-commuting sublattice symmetry $\mathcal{S}$. Contrary to normal inversion, pseudo-inversion hence allows for a nontrivial flux response, the higher-order flux skin effect.

### D.3 Resolving the higher-order flux skin effect in the EHH spectrum

In Sec. 5, we derived the presence of a higher-order flux skin effect in NH symmetry classes A, $A^S$, $AII^{S_+}$, and $C^{S_+}$. While being observable in the NH system due to the extensive localization of states, the occurrence in the corresponding EHH is more intricate: the inherent nontrivial surface state of the EHH obscures flux-localized modes. We can nevertheless cleanly identify the presence of flux induced zero-energy modes in the EHH by considering an interval around zero energy $-\epsilon < E < \epsilon$, within the bulk energy gap ($|\epsilon| < E_{\text{bulk}}$[5]). We denote the set of states within this interval by $\mathcal{E}$ (see Fig. 5b and Fig. 11 for examples). The number of states $|\mathcal{E}|$ can be used to resolve the higher-order flux skin effect.

NH class A and $A^S$ map to Hermitian class AIII, where the total number of states $|\mathcal{E}|$ has to be a multiple of 4 in the absence of a magnetic flux $\phi = 0$,

$$|\mathcal{E}| \mod 4 = 0. \tag{D.18}$$

This can be understood from the fact that the EHH in Hermitian class AIII hosts an integer number of Dirac cones on each surface [79]. Since there is no net chirality per surface,[6] energy eigenvalues appear in pairs $(E, -E)$ on each surface. Inversion symmetry maps between the two surfaces, such that $\mathcal{E}$ contains a multiple of 4 states.

On the other hand, for a finite flux $\phi = \pi$, we obtain one additional zero-energy mode per flux tube (see App. B.3.2), and so we find for a PBC system hosting two flux tubes that

$$|\mathcal{E}| \mod 4 = 2. \tag{D.19}$$

Calculating $|\mathcal{E}| \mod 4$ for our example system in NH class A (Fig. 5b) yields $|\mathcal{E}|_{\phi=0} \mod 4 = 0$ (upper left inset), while we find $|\mathcal{E}|_{\phi=\pi} \mod 4 = 2$ as predicted (lower right inset).

This relation is modified for the NH classes $AII^{S_+}$ and $C^{S_+}$. NH class $AII^{S_+}$ maps to Hermitian class CII⊗CII, of which only one subspace is nontrivial (see App. B.5.9). Hermitian class CII shows a 4-fold degenerate Dirac cone per surface, as chiral symmetry combined with TRS yields pairs of doubly degenerate states. As inversion symmetry maps between the two surfaces, states appear as multiples of eight in $\mathcal{E}$. The introduction of two flux tubes amends this by 4 additional states (see App. B.3.5), hence we obtain for the number of states in $\mathcal{E}$, denoted by $|\mathcal{E}|$, that

$$|\mathcal{E}| \mod 8 = 0 \, (4), \tag{D.20}$$

for flux $\phi = 0 \, (\pi)$. Fig. 11a and b show the comparison of both cases and the validity of the above index.

NH class $C^{S_+}$ maps to Hermitian class AIII in two interdependent unitary subspaces (see App. B.5.11). As outlined in Sec. 5, each subspace comes with a multiple of 4 states. The full system hence has $\mathcal{E}$ containing multiples of 8 states. The introduction of two flux tubes amends this by 2 additional states per unitary subspace (see App. B.3.2), hence

$$|\mathcal{E}| \mod 8 = 0 \, (4), \tag{D.21}$$

for flux $\phi = 0 \, (\pi)$. Fig. 11c and d show the comparison of both cases and the validity of the above index.

---

[5]$E_{\text{bulk}}$ corresponds to the smallest EHH energy $E_{\text{bulk}} = \min(|E|)$ in absence of flux tubes and where PBC are implemented in all directions.

[6]We refer to chirality as the trace of chiral symmetry projected into the surface states, with details explained in App. D.1.

# E  Toy models for all NH symmetry classes with nontrivial flux response

## E.1  2D models

In this appendix we present models for all intrinsically nontrivial 2D point-gapped NH classes showing a nontrivial flux response. As outlined in the main text, we consider flux defects oriented along the $x$-direction.

### E.1.1  Class AII$^\dagger$

The model in NH class AII$^\dagger$ is given by the Hamiltonian

$$\mathcal{H}_{\text{AII}^\dagger}(\boldsymbol{k}) = \sin(k_x)\sigma_y - \sin(k_y)\sigma_x + i\left(\sum_{i=x,y}\cos(k_i) - \mu\right)\sigma_0 + \Delta\sigma_0, \tag{E.1}$$

with the Pauli matrices $\sigma_\mu$ ($\mu = 0, x, y, z$), possessing pseudo TRS as $U_{\mathcal{T}} = i\sigma_y$ and $\Delta \neq 0$ is a real parameter breaking residual symmetries.

### E.1.2  Class DIII$^\dagger$

The nontrivial point-gapped model in NH class DIII$^\dagger$ is given by the Hamiltonian

$$\mathcal{H}_{\text{DIII}^\dagger}(\boldsymbol{k}) = \sin(k_x)\sigma_x - \sin(k_y)\sigma_y - i\left(\sum_{i=x,y}\cos(k_i) - 1\right)\sigma_0, \tag{E.2}$$

with the Pauli matrices $\sigma_\mu$ ($\mu = 0, x, y, z$). We obtain TRS$^\dagger$ $U_{\mathcal{T}} = \sigma_y$, PHS$^\dagger$ $U_{\mathcal{P}} = \sigma_x$ and chiral symmetry $U_{\mathcal{C}} = \sigma_z$, fulfilling the required commutation relations.

### E.1.3  Class BDI$^{S+-}$

The point gap-nontrivial model in NH class BDI$^{S+-}$ is given by the Hamiltonian

$$\mathcal{H}_{\text{BDI}^{S+-}}(\boldsymbol{k}) = i\begin{pmatrix} 0 & Q(\boldsymbol{k};1) \\ Q(\boldsymbol{k};3) & 0 \end{pmatrix}, \tag{E.3}$$

with

$$Q(\boldsymbol{k};\mu) = -\sin(k_x)\sigma_y - \sin(k_y)\sigma_x - \left(\sum_{i=x,y}\cos(k_i) - \mu\right)\sigma_z, \tag{E.4}$$

and the Pauli matrices $\sigma_\mu$ ($\mu = 0, x, y, z$). We obtain TRS $U_{\mathcal{T}} = \tau_0\sigma_x$, PHS $U_{\mathcal{P}} = \tau_x\sigma_x$, chiral symmetry $U_{\mathcal{C}} = \tau_x\sigma_0$ and sublattice symmetry $\mathcal{S} = \tau_z\sigma_0$, fulfilling the required commutation relations.

### E.1.4  Class D$^{S-}$

A nontrivial point-gapped model in NH class D$^{S-}$ is realized by the Hamiltonian

$$\mathcal{H}_{\text{D}^{S-}}(\boldsymbol{k}) = \begin{pmatrix} 0 & Q(\boldsymbol{k};2) \\ Q(\boldsymbol{k};6) & 0 \end{pmatrix} + \Delta\tau_x\sigma_y, \tag{E.5}$$

with

$$Q(\boldsymbol{k};\mu) = i\sin(k_x)\sigma_z - \sin(k_y)\sigma_0 + i\left(2\sum_{i=x,y}\cos(k_i) - \mu\right)\sigma_y, \tag{E.6}$$

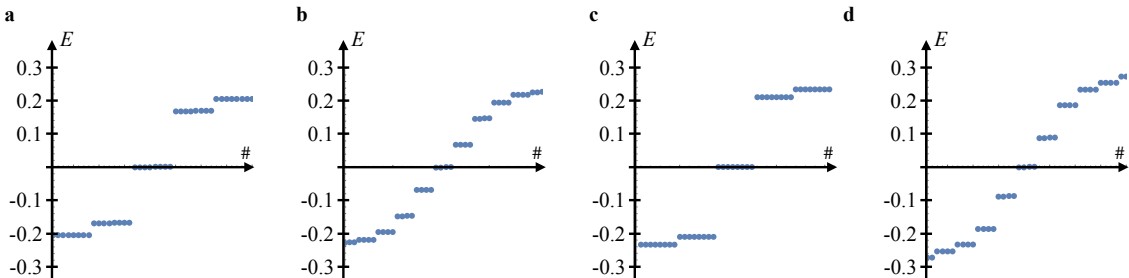

Figure 11: **Higher-order flux skin effect in NH class AII$^{S_+}$ and C$^{S_+}$. a** The EHH for models in NH class AII$^{S_+}$ [Eq. E.17] shows a gapless surface even for $\phi = 0$. **b** Under OBC in NH class AII$^{S_+}$, the presence of a nontrivial flux $\phi = \pi$ introduces four additional flux states. **c** Similarly, the EHH for models in NH class C$^{S_+}$ [Eq. E.19] shows a gapless surface for $\phi = 0$. **d** Under OBC, however, the presence of a nontrivial flux $\phi = \pi$ introduces two additional flux states per unitary subspace, yielding in total four additional modes. All panels are generated for systems of size $20 \times 20 \times 20$ unit cells.

with the Pauli matrices $\sigma_\mu$ and $\tau_\mu$ ($\mu = 0, x, y, z$). The Hamiltonian possesses PHS $U_\mathcal{P} = \tau_x \sigma_0$ and sublattice symmetry $\mathcal{S} = \tau_z \sigma_0$, fulfilling the required commutation relations. Note that $\Delta$ multiplies a term to remove unwanted residual symmetries.

## E.2 3D models

In this appendix we present models for all intrinsically nontrivial 3D point-gapped NH classes showing a nontrivial flux response. As outlined in the main text, we consider flux defects oriented along the $x$- and $z$-direction.

### E.2.1 Class AII$^\dagger$

The model in NH class AII$^\dagger$ is based on the exceptional topological insulator introduced in Ref. [21]. Its Hamiltonian is given by

$$\mathcal{H}_{\text{AII}^\dagger}(\boldsymbol{k}) = \left( \sum_{j=x,y,z} \cos(k_j) - M \right) \tau_z \sigma_0 + \lambda \sum_{j=x,y,z} \sin(k_j) \tau_x \sigma_j + i\delta \tau_x \sigma_0 + \Delta \tau_0 \sigma_0, \quad \text{(E.7)}$$

where the Pauli matrices $\sigma_\mu$ and $\tau_\mu$ act on the spin and orbital degrees of freedom, respectively, with $\mu = 0, x, y, z$ and the 0-th Pauli matrix as the $2 \times 2$ identity matrix. $\Delta \neq 0$ is a real parameter breaking residual symmetries. The invariant is $w_{3D} = 1$ for $3 - \delta/\lambda < M < 3 + \delta/\lambda$. By changing $M$, we transition to the trivial phase at $M = 3 \pm \delta/\lambda$. The Hamiltonian has pseudo-inversion symmetry with $\mathcal{I} = \tau_z \sigma_0$ and pseudo TRS represented by $U_\mathcal{T} = \tau_0 \sigma_y$.

### E.2.2 Class A+$\mathcal{I}^\dagger$

The prototypical phase in 3D NH class A is formed by the exceptional exceptional topological insulator introduced in Ref. [21]:

$$\mathcal{H}_{\text{A}}(\boldsymbol{k}) = \left( \sum_{j=x,y,z} \cos(k_j) - M \right) \tau_z \sigma_0 + \lambda \sum_{j=x,y,z} \sin(k_j) \tau_x \sigma_j + i\delta \tau_x \sigma_0 + \Delta (\tau_0 \sigma_z + \tau_z \sigma_x), \quad \text{(E.8)}$$

where the Pauli matrices $\sigma_\mu$ and $\tau_\mu$ act on the spin and orbital degrees of freedom, respectively, with $\mu = 0, x, y, z$ and the 0-th Pauli matrix as the $2 \times 2$ identity matrix. $\Delta \neq 0$ is a real

parameter breaking residual symmetries. The Hamiltonian is only left with pseudo-inversion symmetry with $\mathcal{I} = \tau_z \sigma_0$.

### E.2.3   Class $\mathrm{A}^S + \mathcal{I}^\dagger$

The nontrivial model in NH class $\mathrm{A}^S$ is given by

$$\mathcal{H}_{\mathrm{A}^S}(\boldsymbol{k}) = \begin{pmatrix} 0 & Q(\boldsymbol{k};2) \\ Q(\boldsymbol{k};0) & 0 \end{pmatrix} + \Delta \tau_x \sigma_x \,, \tag{E.9}$$

with the Pauli matrices $\sigma_\mu$ and $\tau_\mu$ ($\mu = 0, x, y, z$), $\Delta \neq 0$ a real parameter breaking residual symmetries and

$$Q(\boldsymbol{k};\mu) = i \sin(k_x)\sigma_x + i \sin(k_y)\sigma_y + i \sin(k_z)\sigma_z + \left( \sum_{i=x,y,z} \cos(k_i) - \mu \right)\sigma_0 \,. \tag{E.10}$$

$\mathcal{H}_{\mathrm{A}^S}$ has a sublattice symmetry $\mathcal{S} = \tau_z \sigma_0$ and pseudo-inversion symmetry $\mathcal{I} = \tau_x \sigma_0$.

### E.2.4   Class D

The model in NH class D is given by

$$\mathcal{H}_{\mathrm{D}}(\boldsymbol{k}) = -i \sin(k_x)\sigma_z - i \sin(k_y)\sigma_x + \sin(k_z)\sigma_0 + i \left( \sum_{i=x,y,z} \cos(k_i) - 2 \right)\sigma_y \,, \tag{E.11}$$

with the Pauli matrices $\sigma_\mu$ ($\mu = 0, x, y, z$), possessing a PHS $U_{\mathcal{P}} = \sigma_0$.

### E.2.5   Class $\mathrm{D}^{S_+} + \mathcal{I}^\dagger$

The nontrivial point-gapped phase in NH class $\mathrm{D}^{S_-}$ is based on the Hamiltonian in NH class D, formed by

$$\mathcal{H}_{\mathrm{D}^{S_+}}(\boldsymbol{k}) = \begin{pmatrix} 0 & \mathcal{H}_{\mathrm{D}}(\boldsymbol{k}) \\ \mathcal{H}_{\mathrm{D}}(\boldsymbol{k}) & 0 \end{pmatrix} + \Delta(\tau_y \sigma_0 + i\tau_x \sigma_y) \,, \tag{E.12}$$

with the Pauli matrices $\sigma_\mu$ and $\tau_\mu$ ($\mu = 0, x, y, z$) and $\Delta \neq 0$ a real parameter breaking residual symmetries. We obtain PHS $U_{\mathcal{P}} = \tau_0 \sigma_0$ and sublattice symmetry $\mathcal{S} = \tau_z \sigma_0$, fulfilling the required commutation relations. Additionally, $\mathcal{H}_{\mathrm{D}^{S_+}}$ has pseudo-inversion symmetry with $\mathcal{I} = \tau_y \sigma_y$.

### E.2.6   Class $\mathrm{D}^{S_-}$

The nontrivial point-gapped phase in NH class $\mathrm{D}^{S_-}$ is formed by

$$\mathcal{H}_{\mathrm{D}^{S_-}}(\boldsymbol{k}) = \begin{pmatrix} 0 & Q(\boldsymbol{k};2) \\ Q(\boldsymbol{k};0) & 0 \end{pmatrix} \,, \tag{E.13}$$

with the Pauli matrices $\sigma_\mu$ and $\tau_\mu$ ($\mu = 0, x, y, z$) and

$$Q(\boldsymbol{k};\mu) = i \sin(k_x)\sigma_x + i \sin(k_y)\sigma_y + i \sin(k_z)\sigma_z + \left( \sum_{i=x,y,z} \cos(k_i) - \mu \right)\sigma_y \,. \tag{E.14}$$

We obtain PHS $U_{\mathcal{P}} = \tau_y \sigma_y$ and sublattice symmetry $\mathcal{S} = \tau_z \sigma_0$, fulfilling the required commutation relations.

### E.2.7  Class DIII$^{S+-}$

The nontrivial point-gapped phase in NH class DIII$^{S+-}$ is formed by

$$\mathcal{H}_{\text{DIII}^{S+-}}(\boldsymbol{k}) = i\begin{pmatrix} 0 & Q(\boldsymbol{k};2) \\ Q(\boldsymbol{k};4) & 0 \end{pmatrix} + \Delta\rho_x\tau_x\sigma_0, \tag{E.15}$$

with the Pauli matrices $\sigma_\mu$, $\tau_\mu$ and $\rho_\mu$ ($\mu = 0, x, y, z$), $\Delta \neq 0$ a real parameter breaking residual symmetries and

$$Q(\boldsymbol{k};\mu) = \sin(k_x)\tau_x\sigma_x + \sin(k_y)\tau_x\sigma_y + i\sin(k_z)\tau_x\sigma_z + \left(\sum_{i=x,y,z}\cos(k_i) - \mu\right)\tau_z\sigma_0. \tag{E.16}$$

We obtain PHS $U_\mathcal{P} = \rho_x\tau_y\sigma_y$, TRS $U_\mathcal{T} = \rho_z\tau_y\sigma_0$, chiral symmetry $U_\mathcal{C} = \rho_y\tau_0\sigma_y$ and sublattice symmetry $\mathcal{S} = \rho_z\tau_0\sigma_0$, fulfilling the required commutation relations.

### E.2.8  Class AII$^{S+} + \mathcal{I}^\dagger$

The nontrivial point-gapped phase in NH class AII$^{S+}$ is formed by

$$\mathcal{H}_{\text{AII}^{S+}}(\boldsymbol{k}) = i\begin{pmatrix} 0 & Q(\boldsymbol{k};4) \\ Q(\boldsymbol{k};2) & 0 \end{pmatrix} + \Delta\rho_y\tau_x\sigma_z, \tag{E.17}$$

with the Pauli matrices $\sigma_\mu$, $\tau_\mu$ and $\rho_\mu$ ($\mu = 0, x, y, z$), $\Delta \neq 0$ a real parameter breaking residual symmetries and

$$\begin{aligned} Q(\boldsymbol{k};\mu) = &\sin(k_x)\tau_x\sigma_x + \sin(k_y)\tau_x\sigma_y + i\sin(k_z)\tau_x\sigma_z \\ &+ \left(\sum_{i=x,y,z}\cos(k_i) - \mu\right)\tau_z\sigma_0 + \Delta(\tau_y\sigma_x + i\tau_0\sigma_x). \end{aligned} \tag{E.18}$$

We obtain TRS$^\dagger$ $U_\mathcal{T} = \rho_0\tau_0\sigma_y$ and sublattice symmetry $\mathcal{S} = \rho_z\tau_0\sigma_0$, fulfilling the required commutation relations. Additionally, $\mathcal{H}_{\text{AII}^{S+}}$ has pseudo-inversion symmetry with $\mathcal{I} = \rho_y\tau_x\sigma_0$.

### E.2.9  Class C$^{S+} + \mathcal{I}^\dagger$

The nontrivial point-gapped phase in NH class C$^{S+}$ is formed by

$$\mathcal{H}_{\text{C}^{S+}}(\boldsymbol{k}) = i\begin{pmatrix} 0 & Q(\boldsymbol{k};3) \\ Q(\boldsymbol{k};3)^\dagger & 0 \end{pmatrix} + \Delta(\rho_y\tau_x\sigma_z + i\rho_x\tau_x\sigma_0), \tag{E.19}$$

with the Pauli matrices $\sigma_\mu$, $\tau_\mu$ and $\rho_\mu$ ($\mu = 0, x, y, z$), $\Delta \neq 0$ a real parameter breaking residual symmetries and

$$\begin{aligned} Q(\boldsymbol{k};\mu) = &\sin(k_x)\tau_x\sigma_x + \sin(k_y)\tau_x\sigma_y + i\sin(k_z)\tau_x\sigma_z \\ &+ \left(\sum_{i=x,y,z}\cos(k_i) - \mu\right)\tau_z\sigma_0 + \Delta(\tau_y\sigma_x + i\tau_0\sigma_x). \end{aligned} \tag{E.20}$$

We obtain PHS $U_\mathcal{P} = \rho_z\tau_0\sigma_y$ and sublattice symmetry $\mathcal{S} = \rho_z\tau_0\sigma_0$, fulfilling the required commutation relations. Additionally, $\mathcal{H}_{\text{C}^{S+}}$ has pseudo-inversion symmetry with $\mathcal{I} = \rho_y\tau_x\sigma_0$.

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
