# Peer review of "Magnetic Flux Response of Non-Hermitian Topological Phases"

_SciPost Physics, doi:SciPost Phys. 14, 107 (2023)_

## Round 1 · Referee Report · Anonymous (Referee 1) · 2022-9-28

Strengths

1. The manuscript is written in a clear, detailed, and self-contained manner.

2. Interesting new physical phenomena arising from the interplay of non-Hermiticity and magnetic fluxes are clearly presented with the typical microscopic models.

3. The analysis is general and comprehensive, as shown in Table I and its derivation in Appendix.

Report

The authors generally and systematically study the magnetic flux response of non-Hermitian topological systems. On the basis of the topological classification, they identify the relevant symmetry classes with and without pseudo-inversion symmetry in two and three dimensions for which magnetic fluxes nontrivially induce the non-Hermitian skin effect. For these symmetry classes, they elucidate the possible types of the flux skin effects—flux skin effect, flux spectral jump, higher-order flux skin effect, and flux Majorana mode. Their results are supplemented by Appendix, in which the topological classification and the relevant lattice models are discussed in a detailed manner.

Despite the growing recent interest in non-Hermitian topological systems, the magnetic flux response has yet to be completely understood, as also pointed out by the authors themselves. In my humble opinion, the authors’ results provide its general and comprehensive understanding and reveal new interesting physical phenomena that can be experimentally observed in the near future. Thus, I believe that this manuscript makes a significant contribution in the research fields of topological physics and non-Hermitian physics, and I would like to recommend publication of this manuscript in SciPost Physics.

Requested changes

The manuscript is already clear and comprehensive. I only have a couple of relatively minor comments, as explained below.

1. In the explanation of the flux spectral jump in the introduction, the manuscript reads, “Such a gapless NH “dispersion” cannot be realized in any purely 1D lattice system, and therefore represents an intrinsically NH anomalous 1D state.”. I fail to clearly understand this sentence and would appreciate a more detailed explanation. For example, do the authors imply a special meaning by the quotation marks for “dispersion”?

2. In Sec. IV, the authors state that only the parity of the topological invariant (i.e., $W_{3D}$ mod 2) is relevant to the flux response although the integer-valued topological invariant $W_{3D}$ can be well defined in non-Hermitian systems in three dimensions. Does this mean that the magnetic flux response cannot completely detect the integer-valued topological invariant $W_{3D}$ and hence that we need to study an additional response to fully probe $W_{3D}$?

3. In Sec. VII, the authors briefly mention meta-material platforms as candidates for the experimental realization of the authors’ theoretical results. I suppose that the “meta-material platforms” here mean, for example, classical photonic systems, mechanical systems, and electrical circuits, as well as quantum ultracold atoms, in which non-Hermiticity can be introduced and controlled in a judicious manner. However, in such meta-material platforms, how can we introduce magnetic fluxes? In contrast to electronic materials, constituents are charge neutral, and hence it is nontrivial to introduce magnetic fluxes.

  • validity: good
  • significance: good
  • originality: good
  • clarity: good
  • formatting: good
  • grammar: reasonable

Author:  Marco Michael Denner  on 2023-01-09  [id 3220]

(in reply to Report 1 on 2022-09-28)

We thank the Reviewer for taking the time to read our manuscript in detail and for raising some cogent points. We address the Reviewer's comments below:

The authors generally and systematically study the magnetic flux response of non-Hermitian topological systems. On the basis of the topological classification, they identify the relevant symmetry classes with and without pseudo-inversion symmetry in two and three dimensions for which magnetic fluxes nontrivially induce the non-Hermitian skin effect. For these symmetry classes, they elucidate the possible types of the flux skin effects—flux skin effect, flux spectral jump, higher-order flux skin effect, and flux Majorana mode. Their results are supplemented by Appendix, in which the topological classification and the relevant lattice models are discussed in a detailed manner. Despite the growing recent interest in non-Hermitian topological systems, the magnetic flux response has yet to be completely understood, as also pointed out by the authors themselves. In my humble opinion, the authors’ results provide its general and comprehensive understanding and reveal new interesting physical phenomena that can be experimentally observed in the near future. Thus, I believe that this manuscript makes a significant contribution in the research fields of topological physics and non-Hermitian physics, and I would like to recommend publication of this manuscript in SciPost Physics. The manuscript is already clear and comprehensive. I only have a couple of relatively minor comments, as explained below.

We thank the Reviewer for this positive assessment of our work and are glad that it was found to be clearly written, relevant and interesting.

1 In the explanation of the flux spectral jump in the introduction, the manuscript reads, “Such a gapless NH “dispersion” cannot be realized in any purely 1D lattice system, and therefore represents an intrinsically NH anomalous 1D state.”. I fail to clearly understand this sentence and would appreciate a more detailed explanation. For example, do the authors imply a special meaning by the quotation marks for “dispersion”?

We thank the Reviewer for notifying us that this point was not explained clearly enough in the previous version of our manuscript. We here refer to the special nature of this response, where the point gap is completely filled with edge states only at a specific edge momentum. Since such discontinuities do not appear in Hermitian edge state dispersions, we dressed the word dispersion with quotation marks. To resolve the confusion, we have removed the quotation marks in the updated manuscript and added the explanatory sentences: "We note that a purely 1D system cannot realize a filled point gap at a single momentum, as there is only a finite Hilbert space available at any given momentum. Instead, the discontinuity in the 1D flux tube dispersion described here capitalizes on a topologically nontrivial 2D bulk."

2 In Sec. IV, the authors state that only the parity of the topological invariant (i.e., $ W_{\mathrm{3D}}(E) \text{ mod } 2$) is relevant to the flux response although the integer-valued topological invariant $W_{\mathrm{3D}}$ can be well defined in non-Hermitian systems in three dimensions. Does this mean that the magnetic flux response cannot completely detect the integer-valued topological invariant $W_{\mathrm{3D}}$ and hence that we need to study an additional response to fully probe $W_{\mathrm{3D}}$?

This is indeed correct, the magnetic flux response can only detect $W_{\mathrm{3D}} \text{ mod } 2$, similar to the case of non-Hermitian symmetry indicators in the presence of inversion symmetry [Phys. Rev. B 103, L201114]. Additional responses would be needed to probe the $\mathbb{Z}$ nature of $W_{\mathrm{3D}}$. We have added a clarifying sentence to the manuscript.

3 In Sec. VII, the authors briefly mention meta-material platforms as candidates for the experimental realization of the authors’ theoretical results. I suppose that the “meta-material platforms” here mean, for example, classical photonic systems, mechanical systems, and electrical circuits, as well as quantum ultracold atoms, in which non-Hermiticity can be introduced and controlled in a judicious manner. However, in such meta-material platforms, how can we introduce magnetic fluxes? In contrast to electronic materials, constituents are charge neutral, and hence it is nontrivial to introduce magnetic fluxes.

We thank the Reviewer for pointing this out, in our manuscript we specifically focus on meta-materials based on electrical circuits. Within these, hoppings can be equipped with arbitrary phases by using operational amplifiers. We have included another reference showcasing these circuit elements, to further substantiate the experimental realization section [arXiv:2205.05106]. We have also added a sentence and references on the realization of (effective) magnetic fluxes in photonic, mechanical, and ultra-cold atom systems.

---

## Round 1 · Referee Report · Anonymous (Referee 2) · 2022-12-8

Report

First of all, I want to apologize for taking so long to write this report.

The authors consider the response of 2D and 3D non-Hermitian (NH) systems to fluxes or flux tubes. By performing an analysis of multiple symmetry classes, they show the conditions under which fluxes can lead to the formation of a non-Hermitian skin effect (NHSE). They find several kinds of NHSE, depending on dimension and symmetry class: the flux skin effect, the flux spectral jump, the higher-order flux skin effect, and the flux Majorana mode.

I really enjoyed reading this paper. It is well structured, and the introduction and pedagogical explanation of their approach is exceptionally well written. In my opinion, the classification of flux-induced NHSE fulfills the acceptance criteria of Scipost Physics (https://scipost.org/SciPostPhys/about), namely it "opens a new pathway in an existing or a new research direction, with clear potential for multipronged follow-up work." Therefore, I believe this work should be published in Scipost Physics. Before that, however, I would like to ask the authors to address the following points:

1) I'm confused about the notion of semi-infinite boundary conditions (SIBC) in the case of fluxes. For the NHSE appearing at boundaries, I can visualize what SIBC means, it's when the system has only one boundary and extends infinitely in all other directions. How is SIBC done in the case of fluxes, in which sense is it "semi-infinite" ?

2) The point above becomes especially relevant when discussing the flux Majorana mode or the higher-order flux skin effect. If the SIBC there consists of having a semi-infinite system in the z direction, which remains infinite in x and y, then wouldn't the SIBC spectrum contain both the modes associated to the ends of the flux tubes and the modes associated with the surface NHSE? In footnote [61] and in the discussion on page 7, the authors mention that the finite-size splitting of the surface modes is algebraic, whereas the splitting of flux modes is exponential, so it should be possible to have a higher-order flux skin effect without a surface NHSE. However, the definition of SIBC in Eq. (3) consists of first taking the limit of $L \to \infty$, so that this splitting is formally zero, regardless of its algebraic/exponential nature.

3) This last comment also brings me to the notion of observability of the flux skin effects. I really enjoyed the way in which the authors introduce the pseudospectrum, by saying it is a practical definition, given that all experiments have some error. For a realistic condensed-matter system, however, I'm not sure whether it would be possible to have an experimental error which is larger than the exponential splitting of flux modes, while being smaller than the algebraic splitting of the surface modes, especially since the latter might be something like 1/(Avodadro's number).

4) Also relating to observability, how does the flux spectral jump appear when the system is finite in the z direction? Can it still be observed, despite the surface NHSE, or does the later feature wipe it out?

5) Is the flux spectral jump a "jump" or a crossover? Is there really only one momentum at which the number of flux modes scales with system size, or is there some range of momenta at which this happens, maybe with a k-dependent prefactor?

6) What happens to the higher-order flux skin effect and the flux Majorana mode when the flux tubes are dragged away from the top and bottom surfaces? What I mean is that instead of having a plane of minus-sign hoppings extending over all z coordinates, this plane would extend over (say) half of the z values. This means having a single flux loop in the middle of the system, which would still respect the inversion symmetry. Would the modes disappear or would they exist somewhere in the bulk of the system. This question would help to clarify the extent to which the 2 flux tubes can be seen as a quasi-1D NH system aligned along the flux tube direction (discussion on page 7). It might even be related to the observability of flux Majorana modes on the surface. The same question applies to bringing the 2 flux tubes close to each other: would this remove the skin modes, or would it just reduce the number of "orbitals per unit cell" of the quasi-1D NH system along the tubes?

7) I'm confused about the quantization of flux in the presence of inversion symmetry. I could be wrong, but as far as I understand inversion symmetry by itself is not a sufficient condition to quantize the flux, but both inversion and PBC in x and y are needed simultaneously. This means that if I assume realistic systems to have OBC in all directions, then even if the full crystal is perfectly inversion symmetric, the fluxes would not be forced to be quantized. Is this true? This seems to be a different requirement from HOTIs, for instance, where the system would host protected corner modes (say) as long as the crystal is inversion symmetric. The authors should comment on these potential differences.

8) I got confused about the higher-order flux skin effect, when the authors stated that "localized states appear at alternating ends of the two flux tubes, resulting in one state per flux tube." The two flux tubes should be physically indistinguishable, as discussed at the bottom of page 4. Therefore, my feeling is that there shouldn't be 1 mode localized at the top of flux tube 1 and another mode localized at the bottom of flux tube 2. The authors should clarify how these two statements (1 mode per tube and indistinguishable tubes) relate to each other. Maybe by showing the real-space probability density of these modes? Same for the higher-order Majorana flux modes.

9) In the section on experimental realization, why should the size of vortices be matched with the lattice scale by the use of Moire potentials? What would happen if it's not matched? To what extent is this a requirement?

Finally, I have some very minor comments:

A) Please add paper titles to the reference list, it would make it so much more easy to see which paper is cited.

B) Please add another column to Tab. II and and to Tab. III, showing the EHH classes at both $E_0=0$ and at $E_0 \neq 0$. I got confused several times while going through the mapping between the NH Hamiltonian and the EHH.

C) At the top of page 14, and on page 15 [before Eq. (B33)], are those $O$s the only symmetry-allowed perturbations? If the purpose is to show that the modes cannot gap out, then all possible $O$s should be considered.

D) I found a few typos:
There's a typo in Eq. (B2), the brackets are empty.
After Eq. (B11), I think the "not" might be a typo.
Just above Eq. (B30), I think "edge Hamiltonian" is a typo, since the equation is already the combined one, not the small $h$.
Below Eq. (B31), I think "1D" -> "2D".
At several places in the appendix, the 2D system are said to have flux "tubes". I think maybe using vortices, or flux cores is better, while keeping the tube for 3D.

  • validity: high
  • significance: high
  • originality: high
  • clarity: high
  • formatting: excellent
  • grammar: excellent

Author:  Marco Michael Denner  on 2023-01-09  [id 3221]

(in reply to Report 2 on 2022-12-08)

We thank the Reviewer for taking the time to read our manuscript and for providing us with constructive and relevant comments, which we address below:

First of all, I want to apologize for taking so long to write this report. The authors consider the response of 2D and 3D non-Hermitian (NH) systems to fluxes or flux tubes. By performing an analysis of multiple symmetry classes, they show the conditions under which fluxes can lead to the formation of a non-Hermitian skin effect (NHSE). They find several kinds of NHSE, depending on dimension and symmetry class: the flux skin effect, the flux spectral jump, the higher-order flux skin effect, and the flux Majorana mode. I really enjoyed reading this paper. It is well structured, and the introduction and pedagogical explanation of their approach is exceptionally well written. In my opinion, the classification of flux-induced NHSE fulfills the acceptance criteria of Scipost Physics (https://scipost.org/SciPostPhys/about), namely it ``opens a new pathway in an existing or a new research direction, with clear potential for multipronged follow-up work." Therefore, I believe this work should be published in Scipost Physics. Before that, however, I would like to ask the authors to address the following points:

We thank the Reviewer for the appreciation of our work and the positive assessment that our results are relevant and interesting.

1) I'm confused about the notion of semi-infinite boundary conditions (SIBC) in the case of fluxes. For the NHSE appearing at boundaries, I can visualize what SIBC means, it's when the system has only one boundary and extends infinitely in all other directions. How is SIBC done in the case of fluxes, in which sense is it "semi-infinite"?

We thank the Reviewer for notifying us of this possibly confusing point. In the case of flux defects, PBC in all directions always requires an even number of flux cores/tubes to be present. The minimal nontrivial geometry therefore hosts two fluxes, separated along the $x$-direction in our conventions. To clarify our notion of SIBC, we have added the following sentence to the manuscript: "SIBC corresponds to the idealized limit where only one flux (= only one "boundary") is present, while the second flux is infinitely far away. In our conventions, a system with SIBC then has a single flux core/tube at $x=0$, is infinitely extended in the $x$-direction, and is finite with PBC/OBC in all remaining directions (but potentially still thermodynamically large)."

2) The point above becomes especially relevant when discussing the flux Majorana mode or the higher-order flux skin effect. If the SIBC there consists of having a semi-infinite system in the $z$-direction, which remains infinite in $x$ and $y$, then wouldn't the SIBC spectrum contain both the modes associated to the ends of the flux tubes and the modes associated with the surface NHSE? In footnote [61] and in the discussion on page 7, the authors mention that the finite-size splitting of the surface modes is algebraic, whereas the splitting of flux modes is exponential, so it should be possible to have a higher-order flux skin effect without a surface NHSE. However, the definition of SIBC in Eq. (3) consists of first taking the limit of $L\rightarrow \infty$, so that this splitting is formally zero, regardless of its algebraic/exponential nature.

As explained in our answer above and the corresponding addition to the manuscript, SIBC for flux insertions is fundamentally different from the SIBC for sample terminations. In particular, our SIBC implementation corresponds to a single flux defect that is infinitely separated from another defect (or a boundary). In the direction along the flux tubes, we may use standard PBC/OBC (while still considering a thermodynamically system). Since the extended Hermitian Hamiltonian of a system with higher-order flux skin effect or flux Majorana mode only has zeromodes at one end of each flux tube, this SIBC convention is sufficient to cleanly separate -- at least in our theoretical formalism -- the NH flux response from a surface skin effect. However, the Reviewer is right in that this distinction may not be resolvable in experiment (see next Reviewer question).

3) This last comment also brings me to the notion of observability of the flux skin effects. I really enjoyed the way in which the authors introduce the pseudospectrum, by saying it is a practical definition, given that all experiments have some error. For a realistic condensed-matter system, however, I'm not sure whether it would be possible to have an experimental error which is larger than the exponential splitting of flux modes, while being smaller than the algebraic splitting of the surface modes, especially since the latter might be something like 1/(Avodadro's number).

We agree with the Reviewer that such a distinction might be experimentally challenging with respect to the overall (not real-space resolved) spectrum. However, we note that even if surface modes do contribute to the pseudospectrum due to large experimental errors, they are still distributed across the entire surface. On the other hand, flux modes appear at the flux tubes only and therefore lead to an experimentally observable peak in the local density of states. We have added a comment to this effect to Sec. V.

4) Also relating to observability, how does the flux spectral jump appear when the system is finite in the $z$-direction? Can it still be observed, despite the surface NHSE, or does the later feature wipe it out?

We have added the following clarifying sentence in the updated Sec. IV: "For OBC in the $z-$direction, where $k_\parallel$ is not conserved, the NH flux spectral jump still implies an accumulation of states at the flux tube. There may potentially also be surface states contributed by a surface skin effect. Next to their different real-space localization, the two effects can be distinguished in terms of the scaling with system size: whereas the surface skin effect is expected to localize $\mathcal{O}(L_{\parallel} = L_z)$ modes, the flux spectral jump localizes $\mathcal{O}(L_{\perp})$ modes."

5) Is the flux spectral jump a "jump" or a crossover? Is there really only one momentum at which the number of flux modes scales with system size, or is there some range of momenta at which this happens, maybe with a k-dependent prefactor?

We thank the Reviewer for bringing up this point. The non-Hermitian flux spectral jump derives from the flux response of the corresponding extended Hermitian Hamiltonian, which hosts flux localized modes crossing zero energy only at a single time-reversal invariant momentum $k = 0, \pi$. Importantly, no symmetry allowed perturbation is able to move these flux localized states away from zero energy at $k = 0, \pi$. Correspondingly, the connected non-Hermitian system shows a flux response only at these momenta, with flux localized states disappearing for $k \neq 0, \pi$. We have added a sentence to this effect to the updated manuscript.

6) What happens to the higher-order flux skin effect and the flux Majorana mode when the flux tubes are dragged away from the top and bottom surfaces? What I mean is that instead of having a plane of minus-sign hoppings extending over all $z$-coordinates, this plane would extend over (say) half of the $z$-values. This means having a single flux loop in the middle of the system, which would still respect the inversion symmetry. Would the modes disappear or would they exist somewhere in the bulk of the system. This question would help to clarify the extent to which the 2 flux tubes can be seen as a quasi-1D NH system aligned along the flux tube direction (discussion on page 7). It might even be related to the observability of flux Majorana modes on the surface. The same question applies to bringing the 2 flux tubes close to each other: would this remove the skin modes, or would it just reduce the number of "orbitals per unit cell" of the quasi-1D NH system along the tubes?

Reducing the extent of the flux tubes in the $z-$direction does indeed not remove the presence of the higher-order flux skin effect, as inversion symmetry is still preserved. Our focus on flux lines that pass through the entire system is owed to the fact that such a situation is experimentally more relevant. However, it might be possible to effectively terminate flux lines at least in meta-material platforms. We have therefore added a sentence to the manuscript stating that in such a situation the flux response becomes cleanly separated from the surface states. To reply to the Reviewer's other question, bringing two flux tubes close to one another will result in a phase transition (along the flux tubes) that removes all flux modes in the limit of zero distance. We now state this explicitly in Sec.V.

7) I'm confused about the quantization of flux in the presence of inversion symmetry. I could be wrong, but as far as I understand inversion symmetry by itself is not a sufficient condition to quantize the flux, but both inversion and PBC in x and y are needed simultaneously. This means that if I assume realistic systems to have OBC in all directions, then even if the full crystal is perfectly inversion symmetric, the fluxes would not be forced to be quantized. Is this true? This seems to be a different requirement from HOTIs, for instance, where the system would host protected corner modes (say) as long as the crystal is inversion symmetric. The authors should comment on these potential differences.

This is correct -- in experiment, the flux must be fine-tuned to $\phi=\pi$ for our theory to be applicable. We cannot make a universal statement about the case where $\phi\neq 0,\pi$. This is in fact similar to the case of flux modes in Hermitian HOTIs [Nat. Comm. 13, 5791] (but, as noted by the Reviewer, it is different from the case of HOTI corner states). The only general statement is that, for systems with nontrivial flux response and terminated with OBC, modes will be pumped from the boundary to the flux core/tube as $\phi$ is varied from $\phi=0$ to $\phi=\pi$. We have added this elaboration to Sec.V.

8) I got confused about the higher-order flux skin effect, when the authors stated that "localized states appear at alternating ends of the two flux tubes, resulting in one state per flux tube." The two flux tubes should be physically indistinguishable, as discussed at the bottom of page 4. Therefore, my feeling is that there shouldn't be 1 mode localized at the top of flux tube 1 and another mode localized at the bottom of flux tube 2. The authors should clarify how these two statements (1 mode per tube and indistinguishable tubes) relate to each other. Maybe by showing the real-space probability density of these modes? Same for the higher-order Majorana flux modes.

We thank the Reviewer for notifying us of this possibly confusing point. The statement about the localization of flux modes mentioned by the Reviewer refers to the extended Hermitian Hamiltonian, derived in Ref. 14. We now point this out explicitly in the text. Due to the presence of inversion symmetry, a localization of one mode at the same end of each flux tube is not possible. Modes on each end of the two flux tubes would be compatible with inversion symmetry, however, these modes can be annihilated by joining them in an inversion symmetric way on the surface. The only non-trivial scenario is having one mode per flux tube at alternating ends. Such a situation should be compared with the two corner states that localize at opposite corners in a Hermitian higher-order topological insulator protected by inversion symmetry. In the non-Hermitian higher-order flux skin effect, flux skin modes are localized at the same end of each of the two flux tubes. Their real-space density is plotted in Fig.5c and the one for the higher-order Majorana flux modes in Fig.6d,h.

9) In the section on experimental realization, why should the size of vortices be matched with the lattice scale by the use of Moire potentials? What would happen if it's not matched? To what extent is this a requirement?

We thank the Reviewer for this comment. Stretching the flux vortex over several unit cells induces fluxes not quantized to $\phi = \pi$ in individual unit cells, for which no universal statement applies. However, in general we expect flux localized states to not disappear immediately with increasing vortex size, but rather to spread out over the vortex region. We have added a statement to this effect.

A) Please add paper titles to the reference list, it would make it so much more easy to see which paper is cited.

We thank the Reviewer for this suggestion, which we implemented in the revised version of our manuscript.

B) Please add another column to Tab. II and and to Tab. III, showing the EHH classes at both $E = 0$ and at $E\neq 0$. I got confused several times while going through the mapping between the NH Hamiltonian and the EHH.

We thank the Reviewer for this suggestion, which we implemented in the revised version of our manuscript.

C) At the top of page 14, and on page 15 [before Eq. (B33)], are those Os the only symmetry-allowed perturbations? If the purpose is to show that the modes cannot gap out, then all possible Os should be considered.

We indeed consider all symmetry-allowed perturbations to ensure that modes cannot be gapped out. We added a clarification to the revised version of our manuscript.

D) I found a few typos: There's a typo in Eq. (B2), the brackets are empty. After Eq. (B11), I think the "not" might be a typo. Just above Eq. (B30), I think "edge Hamiltonian" is a typo, since the equation is already the combined one, not the small h. Below Eq. (B31), I think "1D" $\rightarrow$ "2D". At several places in the appendix, the 2D system are said to have flux "tubes". I think maybe using vortices, or flux cores is better, while keeping the tube for 3D.

We thank the Reviewer for taking the time to spot and notify us of these typos, which are corrected in the revised version of the manuscript.

---

## Round 2 · Referee Report · Anonymous (Referee 1) · 2023-1-17

Report

I would appreciate the response and the corresponding revision of the manuscript. The authors have satisfactorily addressed my concerns and appropriately revised the manuscript. Now, I would like to unreservedly recommend publication of this manuscript in SciPost Physics.

---

## Round 2 · Referee Report · Anonymous (Referee 2) · 2023-1-23

Report

The authors have answered all of the questions I had raised during the first round of review. In my opinion, this paper should be published in SciPost Physics as is.

---

## Round 2 · Author Response

We thank the Reviewers again for taking the time to closely look at our manuscript and providing us with important feedback to improve our work in this new version.

---

## Round 2 · List of Changes

• Added clarification of the NH dispersion in the flux spectral jump in Sec. I
  • Added explanation of the SIBC setup considered for flux responses in Sec. II C
  • Added clarification of the spectral jump occuring at a single momentum in Sec. IV
  • Added statement discussing the observability of the flux spectral jump under OBC in Sec. IV
  • Added clarification that flux defects do not probe the Z nature of W3D in Sec. IV
  • Added clarification that states appear at alternating ends of flux tubes for the EHH in Sec. V
  • Added statement about the quantization of fluxes under OBC in the presence of inversion symmetry in footnote of Sec. V
  • Added distinction of surface modes and flux response in the pseudospectrum in footnote of Sec. V
  • Added clarification of the phase transition occuring when flux tubes are brought into proximity in footnote of Sec. V
  • Added statement about flux tubes terminating inside samples in Sec. V
  • Added clarification about the vortex size in NH superconductors in Sec. VII
  • Added reference arXiv:2205.05106 in Sec. VII
  • Added papers for effective magnetic fluxes in other metamaterials in Sec. VII
  • Added paper titles to the reference list
  • Added clarification that O considers all symmetry allowed perturbations in the relevant cases
  • Corrected typos
  • Added EHH classes away from E0 = 0 to Tab. II and III

---

## Editorial Decision

published